# THE GEOMETRY OF PHASE TRANSITIONS IN DIFFUSION MODELS: TUBULAR NEIGHBOURHOODS AND SINGULARITIES

## ABSTRACT

Diffusion models undergo phase transitions during the generative process where data features suddenly emerge in the final stages. The current study aims to elucidate this critical phenomenon from the geometrical perspective. We employ the concept of "injectivity radius", a quantity that characterises the structure of the data manifold. Through theoretical and empirical evidence, we demonstrate that phase transitions in the generative process of diffusion models are closely related to the injectivity radius. Our findings offer a novel perspective on phase transitions in diffusion models, with potential implications for improving performance and sampling efficiency.

## 1 INTRODUCTION

Generative models (Bond-Taylor et al., 2022) address the fundamental challenge of approximating and sampling from complex probability distributions. Diffusion models (Sohl-Dickstein et al., 2015; Ho et al., 2020; Song et al., 2021), a prominent class of generative models, incorporate two primary processes: a forward (diffusion) process, wherein data points are perturbed by incrementally adding noise to the data, mapping a complex distribution into an analytically tractable prior distribution, and a backward (reverse diffusion) process, where noise is denoised back into a sample from the data distribution by reversing the noise perturbation. The reverse process involves estimating the "score vector", the gradient of the log-density of the perturbed data distribution.

Recent findings report that diffusion models exhibit critical phenomena, the abrupt emergence of distinctive features during the generative process (Ho et al., 2020; Meng et al., 2022; Choi et al., 2022; Zheng et al., 2023; Raya & Ambrogioni, 2023; Georgiev et al., 2023; Sclocchi et al., 2024; Biroli et al., 2024; Li & Chen, 2024), a critical phenomenon we refer to as a "phase transition". Elucidating phase transitions is expected to help distinguishing between irrelevant information (noise) and relevant information, or memorisation and generalisation. This distinction offers valuable insights into optimising the sampling process and developing better conditional diffusion models, such as for tasks involving language-conditioned image generation. While experimental evidences of such phenomena have been provided in various studies, theoretical frameworks still remain limited and under development. Raya & Ambrogioni (2023) have defined local energy and examined its stability; however, their approach is limited to simple data structures, such as hyperspheres, for which the potential is known. We extend this inquiry by exploring tubular neighbourhoods, applicable to data manifolds with more complex geometries, to better understand phase transitions in diverse contexts.

Building on this foundation, we propose a novel geometric interpretation of phase transitions in diffusion models, grounded in the behaviour of the score vectors. As demonstrated in prior research (Stanczuk et al., 2024), the score vectors at the final time step of the generative process are orthogonal to the tangent plane of the data distribution. This implies that score vectors map noisy data points to their nearest points on the noise-free data distribution at the final stage of the generative process. However, these points may not always be unique and depend on the data geometry. Moreover, the uncertainty of such generative trajectories is expected to increase as they move farther from the data manifold. To address this interpretation, we employ the concept of the "injectivity radius" — the supremum distance within which the nearest point on the data distribution is always uniquely determined. We define the region within the injectivity radius as the tubular neighbourhood (Fig. 1).

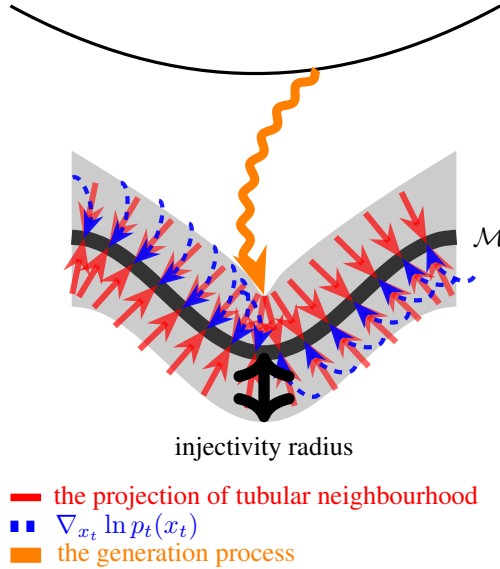

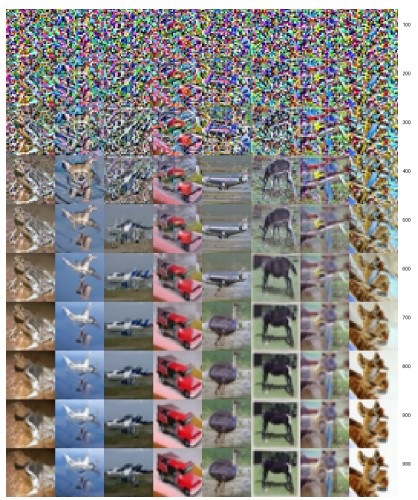

Figure 1: **Conceptual diagram of our perspective**. The orange path represents the generative process from Gaussian noise to the data manifold $\mathcal{M}$. A singularity occurs at the endpoint of this path. The grey region represents the tubular neighbourhood of the data manifold $\mathcal{M}$. We hypothesise that transitions of particles within the grey region play a crucial role in the generative process.

Figure 2: **Example of phase transitions**: CIFAR-10 late initialisation generation. The critical phenomena known as phase transitions or symmetry breaking where the distinctive data features emerge at the certain point of generative process.

We hypothesise that the boundary between the tubular neighbourhood and the region beyond it plays a crucial role in the generative dynamics and is intimately connected to phase transitions.

To test our hypothesis, we conduct experiments using synthetic data and demonstrate that, under conditions of constant curvature, the hypothesis holds true. In contrast, in scenarios where the curvature of the data manifold is non-constant, singularities corresponding to varying curvatures can emerge, leading to the possibility of multiple phase transitions. Moreover, we show that the concept of the tubular neighbourhood corresponds to the final phase transition in the generative process. Finally, we experimentally demonstrate that by embedding the original data distribution into a hypersurface, the theory of the tubular neighbourhood can be leveraged to achieve more efficient sampling. Our code can be found at https://anonymous.4open.science/r/project-anonymous-2024/.

## CONTRIBUTIONS

- We present a novel geometrical perspective of diffusion models to understand critical phenomena, offering a new framework for interpreting the emergence of significant features during the generative process.
- For a given data manifold, we propose an algorithm to estimate the injectivity radius of the tubular neighbourhoods (Section 3). This provides a practical tool for quantifying the geometric structure of data manifolds.
- We analyse the diffusion dynamics through the theory of tubular neighbourhoods and empirically demonstrate that phase transitions occur around these regions (Sections 4 and 5). This combined theoretical and experimental approach strengthens our geometric interpretation of diffusion models and offers a potential method for optimising sampling efficiency by identifying critical points in the generative process.

## 2    PRELIMINARIES

In this section, we briefly introduce some basic mathematical concepts related to the paper.

## 2.1 DIFFUSION MODELS

In Song et al. (2021), score-matching Hyvärinen (2005) and diffusion-based generative models (Sohl-Dickstein et al., 2015; Ho et al., 2020) have been unified into a single continuous-time score-based framework where the diffusion is driven by a stochastic differential equation (SDE) or Langevin dynamics. In this context, $x_t \in \mathbb{R}^d$ represents the data at time $t$, which evolves through time $t \in [0, T]$. This framework relies on Anderson's Theorem (Anderson, 1982), which states that under certain Lipschitz conditions on the drift coefficient $f : \mathbb{R}^d \times \mathbb{R}^d \to \mathbb{R}^d$ and on the diffusion coefficient $g : \mathbb{R}^d \times \mathbb{R} \to \mathbb{R}^d \times \mathbb{R}^d$ and an integrability condition on the target distribution $p_0(x_0)$, a forward diffusion process governed by the SDE

$$dx_t = f_t(x_t)dt + g_t(x_t)dw_t \tag{1}$$

has a reverse diffusion process governed by the SDE

$$dx_t = -\left[ f_t(x_t) - \frac{g_t(x_t)^2}{2} \nabla_{x_t} \ln p_t(x_t) \right] dt + g_t(x_t)dw_t, \tag{2}$$

where $w_t$ is a standard Wiener process in reverse time. We could derive that probability distribution $p_t(x)$ of SDE satisfies the Fokker-Planck equation

$$\frac{\partial}{\partial t} p_t(x_t) = -\nabla_{x_t} \cdot (p_t(x_t)f_t(x_t)) + \frac{1}{2}\Delta_{x_t} \left[ g_t(x_t)^2 p_t(x_t) \right]. \tag{3}$$

Diffusion models are trained by approximating the score function $\nabla_{x_t} \ln p_t(x_t)$ with a neural network $s_\theta(x_t, t)$ parameterised by $\theta$.

## 2.2 FROM THE MANIFOLD HYPOTHESIS TO TUBULAR NEIGHBOURHOODS

Data often concentrates around a lower-dimensional manifold, a concept known as the manifold hypothesis (Fefferman et al., 2013; Loaiza-Ganem et al., 2024). We work in this paper based on this hypothesis. For simplicity, we will assume all data manifolds are compact and embedded in the Euclidean space $\mathbb{R}^d$. In principle, any Riemannian manifolds can be isometrically embedded into some Euclidean space (the Nash embedding theorem).

A tubular neighbourhood of a manifold is roughly speaking a set of points near the manifold and every point of the set has a unique projection onto it (see Appendix C.3 for the formal definition). It is theoretically known that every manifold embedded in $\mathbb{R}^d$ has a tubular neighbourhood. In fact if we take a sufficiently small neighbourhood of a manifold, we may find a tubular neighbourhood. On the other hand, it is easy to imagine that we cannot take a too large neighbourhood as a tubular neighbourhood. See also Appendix A for previous studies which inspired our perspective.

# 3 INJECTIVITY RADIUS OF A DATA MANIFOLD

In this section, we present how to estimate the supremum of possible radii of tubular neighbourhoods — the *injectivity radius* — of a given data manifold. Based on the theoretical argument in below, we establish the algorithm for the estimation (see Algorithm 1 in Appendix F). Throughout this section, let $\mathcal{M}$ denote an $n$-dimensional manifold (data manifold) in the Euclidean space $\mathbb{R}^d$. For the terminologies of Manifold Theory, see Appendices C.2 and C.3.

We refer to (Litherland et al., 1999) for some notions and the case where $(n, d) = (1, 3)$, i.e., the manifold $\mathcal{M}$ is a *knot*. The first crucial claim of this section is that many theoretical facts proven in their paper work for general dimensions as well. The second claim is that the quantities appearing in their paper can be estimated from a given data cloud and its data manifold. For simplicity, we will explain the former briefly and focus on the latter.

## 3.1 ENDPOINT MAPS AND TUBULAR NEIGHBOURHOODS

We explain how to realise a tubular neighbourhood of a manifold embedded in the Euclidean space.

**Definition 3.1.** The $\epsilon$-*neighbourhood* of $\mathcal{M}$ in $\mathbb{R}^d$ is the set

$$\mathcal{M}(\epsilon) = \bigcup_{\boldsymbol{x} \in \mathcal{M}} \{ \boldsymbol{y} \in \mathbb{R}^d \mid \|\boldsymbol{y} - \boldsymbol{x}\| < \epsilon \}.$$

**Definition 3.2.** The *normal bundle* to $\mathcal{M}$ in $\mathbb{R}^d$ is the set

$$N\mathcal{M} = \{(\boldsymbol{x}, \boldsymbol{v}) \in \mathbb{R}^d \times \mathbb{R}^d \mid \boldsymbol{x} \in \mathcal{M}, \boldsymbol{v} \perp T_{\boldsymbol{x}}\mathcal{M}\},$$

where $T_{\boldsymbol{x}}\mathcal{M}$ denotes the tangent space to $\mathcal{M}$ at $\boldsymbol{x}$.

Notice that the set $N\mathcal{M}$ forms a $d$-dimensional manifold. (The dimensions in the direction to $\mathcal{M}$ and its normal are $n$ and $d - n$, respectively.)

**Definition 3.3.** Let $E_0 \colon \mathbb{R}^d \times \mathbb{R}^d \to \mathbb{R}^d$, $(\boldsymbol{x}, \boldsymbol{v}) \mapsto \boldsymbol{x} + \boldsymbol{v}$ be the addition map. We call its restriction

$$E = E_0|_{N\mathcal{M}} \colon N\mathcal{M} \to \mathbb{R}^d, \quad (\boldsymbol{x}, \boldsymbol{v}) \mapsto \boldsymbol{x} + \boldsymbol{v}$$

the *endpoint map* (or the *exponential map*).

**Proposition 3.4** (cf. Theorem C.11). *Let $\epsilon > 0$ and consider the subset*

$$N\mathcal{M}_\epsilon = \{(\boldsymbol{x}, \boldsymbol{v}) \in N\mathcal{M} \mid \|\boldsymbol{v}\| < \epsilon\} \subset N\mathcal{M}.$$

*Then the image $E(N\mathcal{M}_\epsilon)$ coincides with the $\epsilon$-neighbourhood $\mathcal{M}(\epsilon)$ of $\mathcal{M}$ in $\mathbb{R}^d$. Furthermore, this image forms a tubular neighbourhood of $\mathcal{M}$ if and only if the map $E|_{N\mathcal{M}_\epsilon}$ is an embedding.*

### 3.2 INJECTIVITY RADIUS AND ITS ESTIMATION

**Definition 3.5.** The *injectivity radius* $R(\mathcal{M})$ of $\mathcal{M}$ is the supremum of numbers $\epsilon > 0$ such that the $\epsilon$-neighbourhood of $\mathcal{M}$ in $\mathbb{R}^d$ is also a tubular neighbourhood. If such $\epsilon$ does not exist, define $R(\mathcal{M}) = 0$. We also define the following two quantities.

(1) The *first injectivity radius* $R_1(\mathcal{M})$ of $\mathcal{M}$ is the infimum of the set

$$\{\|\boldsymbol{v}\| \mid (\boldsymbol{x}, \boldsymbol{v}) \in N\mathcal{M} \text{ is a critical point of the map } E \text{ for some point } \boldsymbol{x} \in \mathcal{M} \}.$$

(2) The *second injectivity radius* $R_2(\mathcal{M})$ of $\mathcal{M}$ is the infimum of the set

$$\left\{ \frac{1}{2}\|\boldsymbol{x}_1 - \boldsymbol{x}_2\| \;\middle|\; \begin{array}{l} \boldsymbol{x}_1, \boldsymbol{x}_2 \in \mathcal{M}, \boldsymbol{x}_1 \neq \boldsymbol{x}_2, \\ \boldsymbol{x}_1 - \boldsymbol{x}_2 \perp T_{\boldsymbol{x}_1}\mathcal{M} \text{ and } \boldsymbol{x}_1 - \boldsymbol{x}_2 \perp T_{\boldsymbol{x}_2}\mathcal{M} \end{array} \right\}.$$

Roughly saying, $R_1(\mathcal{M})$ is the radius that the endpoint map fails to be regular at some point; $R_2(\mathcal{M})$ is the radius at which two separated tubes come into touch each other (see Figure 3). Thanks to the following assertion, these quantities precisely measure the injectivity radius, i.e., where the first singularity for the $\epsilon$-neighbourhoods occurs.

**Theorem 3.6** (§2 of Litherland et al. (1999)).
*It holds that $R(\mathcal{M}) = \min\{R_1(\mathcal{M}), R_2(\mathcal{M})\}$.*

In this paper, the estimation of $R_2(\mathcal{M})$ is performed following the definition. See Appendix D.3 for some ideas which may make the estimation easier. Therefore we here argue how to estimate $R_1(\mathcal{M})$. This quantity is closely related to the curvature of $\mathcal{M}$ (cf. Fefferman (2020)). Also, it is simple if we consider the case that $n = 1$ — the manifold $\mathcal{M}$ is a curve in $\mathbb{R}^d$ (see Appendix D.2). In general case, it seems to be difficult. However we show the following (see also Theorem C.7).

**Theorem 3.7.** *Assume that the manifold $\mathcal{M} \subset \mathbb{R}^d$ is expressed by $\mathcal{M} = \boldsymbol{F}^{-1}(\boldsymbol{0}) = \{\boldsymbol{x} \in \mathbb{R}^d \mid \boldsymbol{F}(\boldsymbol{x}) = \boldsymbol{0}\}$, where $\boldsymbol{F} \colon \mathbb{R}^d \to \mathbb{R}^{d-n}$ is a differentiable map of which $\boldsymbol{0} \in \mathbb{R}^k$ is a regular value. In addition, assume that we have vector fields $\boldsymbol{t}_1, \boldsymbol{t}_2, \ldots, \boldsymbol{t}_N$ ($n \leq N$) defined near $\mathcal{M}$ such that for every $\boldsymbol{x} \in \mathcal{M}$ the vectors $\boldsymbol{t}_1(\boldsymbol{x}), \boldsymbol{t}_2(\boldsymbol{x}), \ldots, \boldsymbol{t}_N(\boldsymbol{x})$ are tangent to $\mathcal{M}$ and span the tangent space $T_{\boldsymbol{x}}\mathcal{M}$. Then the first injectivity radius $R_1(\mathcal{M})$ coincides with the infimum of the Euclidean*

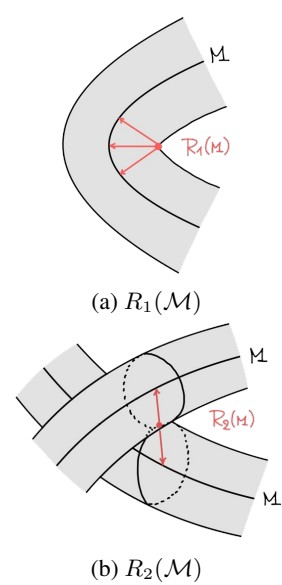

(a) $R_1(\mathcal{M})$

(b) $R_2(\mathcal{M})$

Figure 3: Sketches of the first and second injectivity radii

*norms $\|\boldsymbol{v}\|$ of vectors $\boldsymbol{v} \in \mathbb{R}^d$ such that $\boldsymbol{v} \perp T_{\boldsymbol{x}}\mathcal{M}$ and the $d \times (d + N - n)$-matrix*

$$L_{\mathcal{M}}(\boldsymbol{x}, \boldsymbol{v}) = \left[ \frac{\partial \boldsymbol{F}}{\partial \boldsymbol{x}}(\boldsymbol{x})^T \quad \left( \frac{\partial \varphi_1}{\partial \boldsymbol{x}}(\boldsymbol{x}, \boldsymbol{v}) - \frac{\partial \varphi_1}{\partial \boldsymbol{v}}(\boldsymbol{x}, \boldsymbol{v}) \right)^T \quad \cdots \quad \left( \frac{\partial \varphi_N}{\partial \boldsymbol{x}}(\boldsymbol{x}, \boldsymbol{v}) - \frac{\partial \varphi_N}{\partial \boldsymbol{v}}(\boldsymbol{x}, \boldsymbol{v}) \right)^T \right]$$
(4)

*is degenerate for some point $\boldsymbol{x} \in \mathcal{M}$, where*

$$\varphi_i \colon \mathbb{R}^d \times \mathbb{R}^d \to \mathbb{R}, \quad \varphi_i(\boldsymbol{x}, \boldsymbol{v}) = \langle \boldsymbol{t}_i(\boldsymbol{x}), \boldsymbol{v} \rangle$$

*for $i = 1, 2, \cdots, N$.*

This assertion is proven by an application of the Method of Lagrange Multiplier. See Appendix D.1 for its precise proof. We here note some remarks.

**Remark 3.8.** The condition that the matrix $L(\boldsymbol{x}, \boldsymbol{v})$ degenerates at $(\boldsymbol{x}, \boldsymbol{v}) \in N\mathcal{M}$ is equivalent to that the determinant of the $d \times d$-minor

$$\left[ \frac{\partial \boldsymbol{F}}{\partial \boldsymbol{x}}(\boldsymbol{x})^T \quad \left( \frac{\partial \varphi_{i_1}}{\partial \boldsymbol{x}}(\boldsymbol{x}, \boldsymbol{v}) - \frac{\partial \varphi_{i_1}}{\partial \boldsymbol{v}}(\boldsymbol{x}, \boldsymbol{v}) \right)^T \quad \cdots \quad \left( \frac{\partial \varphi_{i_n}}{\partial \boldsymbol{x}}(\boldsymbol{x}, \boldsymbol{v}) - \frac{\partial \varphi_{i_n}}{\partial \boldsymbol{v}}(\boldsymbol{x}, \boldsymbol{v}) \right)^T \right]$$
(5)

of $L(\boldsymbol{x}, \boldsymbol{v})$ vanishes for every $n$-tuple $(i_1, \ldots, i_n)$ satisfying that $1 \le i_1 < \cdots < i_n \le N$. Indeed, the matrix $\frac{\partial \boldsymbol{F}}{\partial \boldsymbol{x}}(\boldsymbol{x})$ is of full-rank for every point $\boldsymbol{x} \in \mathcal{M} = \boldsymbol{F}^{-1}(\boldsymbol{0})$.

**Remark 3.9.** It is crucial to find vector fields $\boldsymbol{t}_i$ satisfying the above condition. For example, (small extensions of) the gradient vector fields $\boldsymbol{t}_i = \text{grad } x_i$ $(i = 1, \ldots, d)$ satisfies the condition, where $x_i \colon \mathcal{M} \to \mathbb{R}$ denotes the projection to the $i$-th axis in $\mathbb{R}^d$. In general, we have to take the number $N$ greater than $n$.

### 3.3 EXAMPLE (UNIT CIRCLE $S^1$)

Let us verify Theorem 3.7 through the most typical manifold — the unit circle $S^1$. Define a function $F \colon \mathbb{R}^2 \to \mathbb{R}$ by

$$F(x, y) = x^2 + y^2 - 1.$$

Then we have $S^1 = F^{-1}(0)$. One of the normal vector field on $S^1$ is given as $\text{grad}(F) = (\frac{\partial F}{\partial x}, \frac{\partial F}{\partial y}) = (2x, 2y)$, so $(-y, x)$ is a tangent vector field which spans the tangent space to $S^1$ at each point $(x, y) \in S^1$. Applying Theorem 3.7, the first injectivity radius $R_1(S^1)$ is calculated as follows. For a point $(x, y) \in S^1$, the matrix

$$L_{S^1}((x, y), (v_1, v_2)) = \begin{bmatrix} 2x & v_2 + y \\ 2y & -v_1 - x \end{bmatrix}$$

is degenerate (i.e., its determinant is zero) if and only if $(v_1, v_2) = (-x, -y)$. Thus, we obtain

$$R_1(S^1) = \sqrt{(-x)^2 + (-y)^2} = 1.$$

By definition, $R_2(S^1)$ is also equal to 1, so the injectivity radius $R(S^1)$ is equal to 1.

This result is utilised in Section 5.1 for the experimental validation.

### 3.4 A PILOT NUMERICAL EXPERIMENT TO VALIDATE THE PROPOSED ALGORITHM

We perform a pilot experiment to verify the algorithm. The detailed setting and the results are present in the Appendix F.1. The estimated $R$ for $S^1$ is $0.999 \pm 0.006$.

## 4 TUBULAR NEIGHBOURHOODS AND DIFFUSION DYNAMICS

In this section, we investigate the relation between tubular neighbourhoods and diffusion dynamics.

## 4.1 THE PROPORTION OF PARTICLES WITHIN THE TUBULAR NEIGHBOURHOOD

Let $\epsilon > 0$. Let $\mathcal{M}(\epsilon)$ be the $\epsilon$-neighbourhood of a compact oriented manifold $\mathcal{M}$ in the Euclidean space $\mathbb{R}^d$ as defined in Definition 3.1. Suppose $p_t(x)$ is a smooth solution to the Fokker-Planck equation (3) with an initial condition $p_0(x) = \delta_{\mathcal{M}}(x)$ here $\delta_{\mathcal{M}}(x)$ is Dirac's density function with its support $\mathcal{M}$. We define a function $\Gamma_{\mathcal{M}(\epsilon)}(t)$ as follows:

$$\Gamma_{\mathcal{M}(\epsilon)}(t) := \int_{\mathcal{M}(\epsilon)} p_t(x)dx. \tag{6}$$

**Remark 4.1.** The readers may understand this function represents the proportion of particles within the tubular neighbourhood(see also Section 5.1 for the specific cases in numerical experiments).

**Proposition 4.2.** *Assume* $\beta(t) : \mathbb{R}_{\geq 0} \to \mathbb{R}$ *is a smooth function and* $f_t(x) = \frac{1}{2}\beta(t)f(x)$, $g_t(x) = \sqrt{\beta(t)}$ *in (3) (* $f(x)$ *is some smooth vector field). We have:*

$\lim_{t \to 0} \frac{\partial}{\partial t}\Gamma_{\mathcal{M}(\epsilon)}(t) = 0$ *and* $\lim_{t \to \infty} \frac{\partial}{\partial t}\Gamma_{\mathcal{M}(\epsilon)}(t) = 0$. *Thus there exists at least one* $t_c$ *in* $(0, +\infty)$ *such that* $\frac{\partial^2}{\partial t^2}\Gamma_{\mathcal{M}(\epsilon)}(t_c) = 0$. *Moreover if* $\beta(t) > 0$ *and*

$$(\nabla_x \ln p_t(x) - f(x)) \cdot \boldsymbol{n} < 0 \tag{7}$$

*for any* $x \in \partial\mathcal{M}(\epsilon)$ *and any* $t \in \mathbb{R}_{>0}$ *then* $\Gamma_{\mathcal{M}(\epsilon)}(t)$ *is strictly monotonically decreasing. Here* $\boldsymbol{n}$ *is an unit outward pointing normal vector field along* $\partial\mathcal{M}(\epsilon)$.

**Remark 4.3.** In other words, the first term of the SDE (2) at the boundary of the tubular neighbourhood is closely related to the behaviour of $\Gamma_{M(\epsilon)}(t)$. We can write $\Gamma_{M(\epsilon)}(t)$ in terms of free energies on the boundary of the tubular neighbourhood. Refer to Appendix H for a comprehensive analysis and additional details.

## 4.2 THE SCORE VECTOR FIELDS AND TUBULAR NEIGHBOURHOODS

One expresses marginal distribution $p_t(x)$ of Variance Preserving (VP-SDE) (DDPM) as follows:

$$p_t(x) = \int_{\mathcal{M}} N(y|\theta_t x, (1 - \theta_t^2)I)p_0(y)dy, \tag{8}$$

where $\theta_t = e^{-\frac{1}{2}\int_0^t \beta(\tau)d\tau}$ and $p_0(y)$ is the distribution at time $t = 0$. In this section we investigate how three quantities(dimension, injectivity radius, time step) affect the behaviour of the score vector fields $\nabla_x \ln p_t(x)$. Let us first consider the case of spheres.

**Proposition 4.4.** *Suppose* $\mathcal{M} = S^n$ *is a* $n$-sphere of radius $R$ in $\mathbb{R}^d$. *Let* $\epsilon$ *be as* $R > \epsilon > 0$. *Let* $\boldsymbol{n}$ *be a unit outward pointing normal vector to the boundary of* $\epsilon$-neighbourhood $\partial\mathcal{M}(\epsilon)$. *Assume*

$$\frac{\epsilon + (1 - \theta_t)(R - \epsilon)}{\sqrt{1 - \theta_t^2}} \geq \sqrt{d}, \tag{9}$$

$x \in \partial\mathcal{M}(\epsilon)$ *and* $p_0(y)$ *is constant* $C$ *greater than* 0 *on* $\mathcal{M}$. *Then:*

$$\nabla_x \ln p_t(x) \cdot \boldsymbol{n} \leq 0.$$

**Example 4.5.** Let $\mathcal{M} = S^1$ in $\mathbb{R}^2$ and $|x| = 0.99$ (i.e. $\epsilon = 0.99$). Compute (9) and we understand that $\nabla_x \ln p_t(x)$ points toward $S^1$ if $\theta_t > 0.712$. Similar thing can be observed for $S^2$ in $\mathbb{R}^3$. Therefore this explains the Figure 21 and Figure 22.

**Remark 4.6.** This is a kind of generalisation of Theorem D.1 in (Stanczuk et al., 2024).

**Proposition 4.7.** *Suppose* $\mathcal{M}$ *is a compact oriented manifold embedded in* $\mathbb{R}^d$. *Let* $\epsilon_0$ *be an injectivity radius. Let* $\epsilon_0 > \epsilon > 0$. *Let* $\boldsymbol{n}$ *be a unit outward pointing normal vector to the boundary of* $\epsilon$-neighbourhood $\partial\mathcal{M}(\epsilon)$. *Assume*

$$\frac{\epsilon + |x|(1 - \theta_t)}{\sqrt{1 - \theta_t^2}} \geq \sqrt{d}, \tag{10}$$

$x \in \partial\mathcal{M}(\epsilon)$ *and* $p_0(y)$ *is constant* $C$ *greater than* 0 *on* $\mathcal{M}$. *Finally assume a line segment with* $x$ *and the origin as its vertices does not intersect* $\mathcal{M}$. *Then:*

$$\nabla_x \ln p_t(x) \cdot \boldsymbol{n} \leq 0.$$

**Remark 4.8.** This is simplified statement, see Section I.2 for details. The smaller the injectivity radius slower time of the turning of the score vector field becomes. This explains phenomena in Section 5.2.

## 5 EXPERIMENTS

**Remark 5.1.** Throughout this section, given a manifold $\mathcal{M}$ embedded in $\mathbb{R}^d$, *the tubular neighbourhood of $\mathcal{M}$* means that with the injectivity radius $R(\mathcal{M})$, for short.

In this section, we empirically demonstrate the presence of phase transitions at the boundary of the tubular neighbourhood during the generative process of diffusion models. In particular, we analyse the proportion of particles outside the tubular neighbourhood at each time step using the standard DDPM setup ($T = 1000$) and investigate the corresponding changes in the Wasserstein distance between the training data distribution and the generated distribution for varying initial times $T$. The Wasserstein distance is evaluated using the late initialisation scheme (Raya & Ambrogioni, 2023), where the generation process begins at different initial time steps. During inference, we adopted values of $T$ ranging from 1 to 1000, with the initial state set to $\mathbf{x}_T \sim \mathcal{N}(\mathbf{0}, I)$. Our experiments show that during late initialisation, the Wasserstein distance undergoes a sharp shift at certain time step, indicating the onset of phase transitions as particles enter the tubular neighbourhood. The number and timing of these transitions vary based on the curvature of the underlying manifold. For hyperspheres, one phase transition occurs, aligned with a rapid decrease in the proportion of particles outside the tubular neighbourhood and a sharp rise in the Wasserstein distance. For ellipses and tori, the timing of transitions varies due to regions with higher curvature. In the following subsections, we make detail experiments on various geometries, including hyperspheres (see 5.1), ellipses, tori (see 5.2), and also disjoint arcs (see 5.3), to provide a comprehensive understanding of how tubular neighbourhoods affect the generation process. Additionally, we demonstrate that embedding real-world datasets into a hypersphere improves sampling efficiency(see 5.4). The detailed experimental setup, including DDPM parameters and configurations, is provided in Appendix J.

### 5.1 RELATIONSHIP BETWEEN TUBULAR NEIGHBOURHOOD AND PHASE TRANSITION IN UNIT HYPERSPHERE

Based on the assumption that entering the particle within the tubular neighbourhoods have strict relation with occurring the phase transition, we show the several experiments which compared the proportion within the tubular neighbourhood to the Wasserstein distance of the diffusion model. In the experiments, we first count the proportion of the particle outside the tubular neighbourhoods (red line). Here, because we used the unit hypersphere, the injectivity radius in each experiments are 1. Then, we calculate the Wasserstein distance of diffusion model when doing the late initialisation (blue line). This experiments show that the Wasserstein distance of diffusion model rise after some particle entered in the tubular neighbourhoods. Regarding Figure 4, our assumption is strictly true, as the Wasserstein distance increases after a particle enters the tubular neighbourhood. For further experimental details, refer to Sections J.2 and J.3.

Table 1: Wasserstein distances $W$ for different late initialisation times. $\rho_{proportion}$ represents the proportion of particles outside the tubular neighbourhood.

| Dataset $\diagdown$ $\rho_{proportion}$ | 0.1 | 0.5 | 0.9 | 0.95 | 0.99 | 0.999 | 1.0 |
|---|---|---|---|---|---|---|---|
| $S^1$ embedded in $\mathbb{R}^{16}$ | 0.283 | 0.073 | 0.020 | 0.019 | 0.018 | 0.019 | 0.018 |
| $S^2$ embedded in $\mathbb{R}^{16}$ | 1.344 | 0.343 | 0.058 | 0.038 | 0.030 | 0.030 | 0.031 |
| $S^{20}$ embedded in $\mathbb{R}^{24}$ | 3.895 | 1.858 | 0.970 | 0.882 | 0.807 | 0.781 | 0.759 |

### 5.2 RELATIONSHIP BETWEEN TUBULAR NEIGHBOURHOOD AND PHASE TRANSITION IN OTHER CASES

In this section, we demonstrate that while hyperspheres fail to maintain the hypothesis under high-dimensional ambient spaces, this failure is due to the increased distance between the data manifold

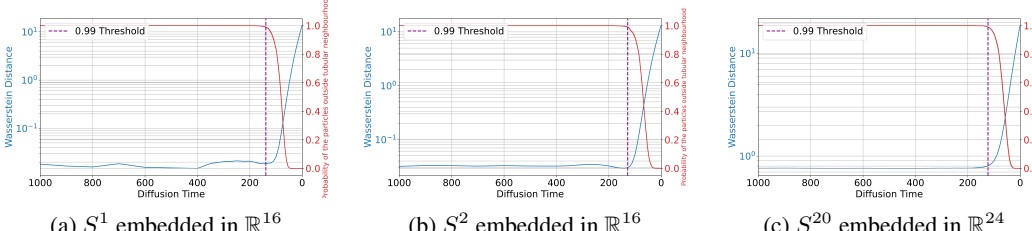

(a) $S^1$ embedded in $\mathbb{R}^{16}$     (b) $S^2$ embedded in $\mathbb{R}^{16}$     (c) $S^{20}$ embedded in $\mathbb{R}^{24}$

Figure 4: The blue line (left axis) depicts the Wasserstein distance between the training data distribution and the generated distribution, measured as a function of the shifted initial diffusion time $T$. The red line (right axis) indicates the proportion of particles outside the tubular neighbourhood at each diffusion time in a standard diffusion model. The purple dashed line marks the diffusion time when 99% of particles exit the tubular neighbourhood. Notably, a sharp increase in the Wasserstein distance is observed as particles enter the tubular neighbourhood.

and the initial Gaussian distribution. Independently, we also show that for tori and ellipsoids, the hypothesis does not hold due to different underlying reasons specific to their geometric properties. As shown in the case of $S^{20}$ embedded in $\mathbb{R}^{48}$ (Figure 5), the hypothesis breaks down when the ambient space is increased. This phenomenon can be attributed to the growing distance between the data manifold and the initial Gaussian distribution as the ambient space dimension increases. Generally, when the ambient space dimension becomes larger, the expected region where the Gaussian distribution is concentrated moves further away from the origin, scaling with the square root of the dimension if the initial state follows a standard normal distribution. In contrast, in our experiments, the data manifold is a unit hypersphere, and its average position remains fixed at a constant distance of 1 from the origin. Consequently, a significant discrepancy emerges between the two distributions. This discrepancy indicates the presence of a phase in the generation process during which the data distribution experiences a substantial average shift. As a result, when sampling from a Gaussian distribution under late initialisation, the difference from the expected distribution at the original time step becomes significantly larger, making accurate reconstruction more challenging (see J.3).

We now focus on two manifolds with "mixed" curvatures in some sense — ellipses and tori. We can verify that the injectivity radius of the ellipse with major axis $2R$ and minor axis $2r$ is given by $r^2/R$, and that of the torus with major radius $R$ and minor radius $r$ is given by $\min\{R - r, r\}$ (see J.4). During the experiments, we observed multiple phases of increase in the Wasserstein distance under late initialisation. Moreover, in standard diffusion models, we confirmed that the time at which particles begin to enter the tubular neighbourhood corresponds to the final sharp increase in the Wasserstein distance. This suggests that the timing of the last spontaneous symmetry breaking can be inferred from the particles within the tubular neighbourhood. Furthermore, these findings imply that the concept of injectivity radius corresponds to a mathematical quantity representing the region of the data manifold with the highest curvature.

Table 2: Wasserstein distances $W$ for different late initialisation times. $\rho_{proportion}$ represents the proportion of particles outside the tubular neighbourhood.

| Dataset $\diagdown$ $\rho_{proportion}$ | 0.1 | 0.5 | 0.9 | 0.95 | 0.99 | 0.999 | 1.0 |
|---|---|---|---|---|---|---|---|
| $S^{20}$ embedded in $\mathbb{R}^{48}$ | 21.754 | 13.380 | 7.964 | 6.978 | 5.366 | 3.900 | 0.752 |
| Ellipse ($R = 3, r = 1$) embedded in $\mathbb{R}^{16}$ | 7.762 | 5.893 | 4.016 | 3.601 | 2.936 | 2.184 | 0.479 |
| Torus ($R = 3, r = 1$) embedded in $\mathbb{R}^{16}$ | 2.597 | 1.888 | 1.433 | 1.335 | 1.167 | 0.872 | 0.272 |

## 5.3 DISJOINT ARCS CASE

To test the assumption that the injectivity radius of a manifold is determined by its region with the highest curvature, we conducted an experiment using two arcs of circles embedded in $\mathbb{R}^{16}$. One arc has a radius of 1, centered at the origin, and the other has a radius of 2 (see Figure 6). The injectivity radii of arcs are 1 and 2, respectively, but the injectivity radius of the manifold itself is 1. Figure 7 shows that the Wasserstein distance rises in two phases. The first rise corresponds to the arc with a

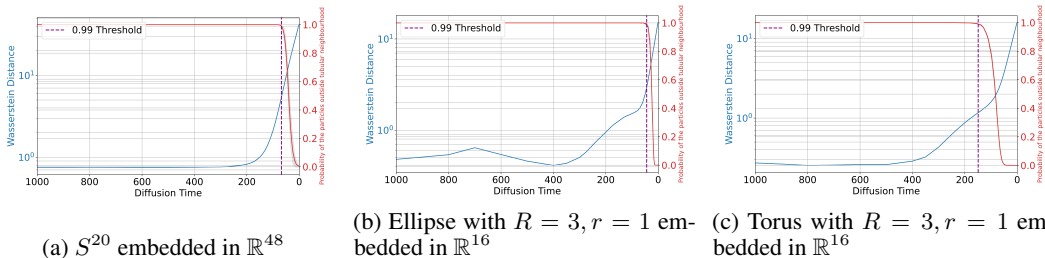

(a) $S^{20}$ embedded in $\mathbb{R}^{48}$

(b) Ellipse with $R = 3, r = 1$ embedded in $\mathbb{R}^{16}$

(c) Torus with $R = 3, r = 1$ embedded in $\mathbb{R}^{16}$

Figure 5: (Blue, left axis) and (Red, right axis) show the Wasserstein distance and the proportion of particles outside the tubular neighborhood, respectively, as described in Figure 4. The purple dashed line marks the 99% threshold. Unlike Figure 4, the Wasserstein distance increases before the particles enter the tubular neighborhood.

radius of 2, and the second rise corresponds to the arc with a radius of 1, which begins as particles start entering the tubular neighbourhood. This experiment confirms that each curvature corresponds to an injectivity radius and a rise in the Wasserstein distance, representing a phase transition. In natural datasets, although multiple curvatures exist, only one phase transition is typically observed.

Table 3: Wasserstein distances $W$ for different late initialisation times. $\rho_{proportion}$ represents the proportion of particles outside the tubular neighbourhood.

| Dataset $\quad \rho_{proportion}$ | 0.1 | 0.5 | 0.9 | 0.95 | 0.99 | 0.999 | 1.0 |
|---|---|---|---|---|---|---|---|
| Disjoint arcs | 0.283 | 0.073 | 0.020 | 0.019 | 0.018 | 0.019 | 0.018 |

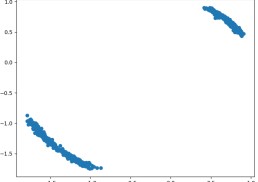

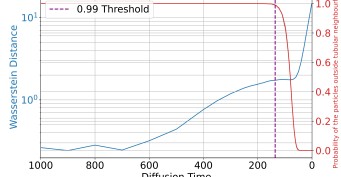

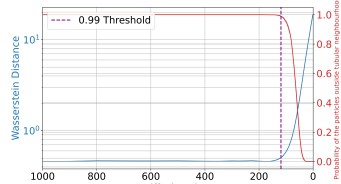

Figure 6: Projection of the 1st and 2nd dimensions of data embedded in $\mathbb{R}^{16}$.

Figure 7: A second rise in the Wasserstein distance occurs, indicating phase transitions.

Figure 8: Fashion MNIST $S^{20}$ embedded in $\mathbb{R}^{24}$.

## 5.4 EXPLORING TUBULAR NEIGHBOURHOODS IN REAL DATASETS

As shown in the section 5.3, the injectivity radius of a given manifold corresponds to the region of the manifold with the highest curvature. This implies that the injectivity radius may not capture the full nature of the phase transition. However, natural datasets like MNIST and Fashion MNIST exhibit diverse curvature, resulting in a very small injectivity radius, which fails to adequately represent the datasets' properties. To address this issue, we embedded the natural dataset into a unit hypersphere and analysed the relationship between the proportion of the particles outside the tubular neighbourhood and the Wasserstein distance when doing late initialisation. In the experiments, we first used Hyperspherical VAE (Davidson et al., 2018) to embed each dataset to some hypersphere. The detailed parameter of the Hyperspherical VAE is given in Appendix J.6. Here, It is important to note that these experiments differ from the experiment of 5.1 in that the distribution on the hypersphere is not uniform. This is because the Hyperspherical VAE used for embedding does not necessarily produce a uniform distribution in the latent space. However, through our experiments, we found that this had no significant impact on the results. Figure 8 shows the Wasserstein distance begin to rise just after the particle begin to enter the tubular neighbourhood. Figure 9 illustrates the reconstruction process at various time steps when sampling from the latent space constrained to a

hypersphere using late initialisation and decoding with a Hyperspherical VAE. It can be observed that the reconstruction accuracy for the Fashion MNIST dataset does not degrade notably, as the Wasserstein distance in the latent space increases by less than 5% during the diffusion time interval from 1000 to 200. Further details can be found in Appendix J.6. These results suggest that the concept of the tubular neighbourhood could be leveraged to perform efficient sampling.

Table 4: Measured Wasserstein distances $W$ for different late initialisation times. The variable $\rho_{proportion}$ represents the proportion of particles outside the tubular neighbourhood, and the corresponding $W$ at the respective time points are shown in the table.

| Dataset $\rho_{proportion}$ | 0.1 | 0.5 | 0.9 | 0.95 | 0.99 | 0.999 | 1.0 |
|---|---|---|---|---|---|---|---|
| MNIST $S^{20}$ embedded in $\mathbb{R}^{24}$ | 2.254 | 1.060 | 0.668 | 0.632 | 0.592 | 0.576 | 0.559 |
| Fashion MNIST $S^{20}$ embedded in $\mathbb{R}^{24}$ | 2.697 | 1.295 | 0.673 | 0.590 | 0.509 | 0.472 | 0.453 |

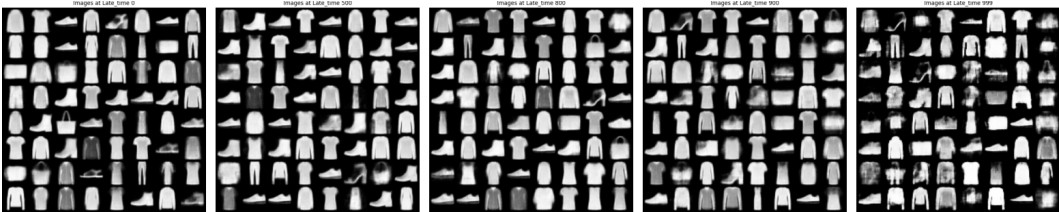

Figure 9: Fashion MNIST images decoded using SVAE after diffusion times of 1000, 500, 200, 100, and 1 (from left to right) in the latent space.

## 6 DISCUSSION AND CONCLUSION

In this study, we employed the concept of the *injectivity radius* to understand the generative process of diffusion models and analysed it theoretically and experimentally from the perspective of the geometric structure of data manifolds. Specifically, we provided a new perspective on a phenomenon where certain features emerge rapidly over a short time interval, referred to as a *phase transition*.

However, the concept of the injectivity radius, which may correspond to the region of the manifold with the largest curvature, only partially explains the phase transition phenomenon. To address this limitation, defining a mathematical quantity that corresponds to the *smallest curvature* of the manifold may remove the assumption of constant curvature, potentially leading to a more general theory applicable to a wider range of datasets. This would allow for more efficient sampling methods that better reflect the geometric properties of the data manifold, thereby enhancing the overall performance of diffusion models. Moreover, from the perspective of *nonequilibrium thermodynamics*, the system's free energy, defined by weighting the energy at each point with the probability distribution function, can provide a more comprehensive macroscopic understanding of phase transitions (for details, see Appendix G). This approach could enable a more accurate discussion of phase transitions as a macroscopic phenomenon, complementing the microscopic geometric analysis.

By exploring the relationship between the data's geometric structure and the score vectors, as discussed in Section 4.2, it may also be possible to design optimal *noise scheduling* strategies that are tailored to the geometry of the data manifold. Clarifying how score vectors interact with the manifold's geometric properties could further optimise the generative process. In addition, our experiments with real-world datasets suggest that modifying the *VAE embedding into hyperspheres* could extend the applicability of the tubular neighbourhood concept to more complex datasets. This modification is expected to not only improve sampling efficiency but also increase the flexibility of diffusion models in handling a wider variety of real-world data. These discussions lie at the intersection of concepts from *differential geometry*, particularly singularities, *statistical physics*, especially phase transitions in non-equilibrium thermodynamics, and *computer science*, specifically diffusion models. This interdisciplinary approach represents an important step forward, and further theoretical development and practical applications in this direction hold promising potential.

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

## A    RELATED WORK

Diffusion models (Sohl-Dickstein et al., 2015; Ho et al., 2020; Song et al., 2021) have emerged as a powerful class of generative models (Bond-Taylor et al., 2022), demonstrating remarkable performance in various domains including text-to-image synthesis (Nichol & Dhariwal, 2021; Dhariwal & Nichol, 2021; Ding et al., 2021; Ramesh et al., 2022; Nichol et al., 2022; Rombach et al., 2022; Saharia et al., 2022; Yu et al., 2022; Ho et al., 2022a), text-to-speech (Chen et al., 2021; Kong et al., 2021; Liu et al., 2023), video generation (Ho et al., 2022b; Singer et al., 2023; Xing et al., 2023), natural language processing (Li et al., 2022; Lou et al., 2023; Li et al., 2023b), robot manipulations (Janner et al., 2022; Zhu et al., 2023), inverse problems (Daras et al., 2024), and protein interactions modelling (Abramson et al., 2024).

**Motivations and related works.** Our work is motivated by several recent theoretical advancements (Yeğin & Amasyalı, 2024) and practical challenges (Chen et al., 2024; Yang et al., 2024):

- **Optimisation of Diffusion Time**: Some empirical studies report existence of an optimal diffusion time that enhances model efficiency (Franzese et al., 2023).
- **Critical Phenomena and Statistical Thermodynamics of Diffusion Models**: There are some empirical and theoretical studies report heterogeneity/non-uniformity, critical phenomena during generation (Ho et al., 2020; Meng et al., 2022; Choi et al., 2022; Zheng et al., 2023; Raya & Ambrogioni, 2023; Georgiev et al., 2023; Sclocchi et al., 2024; Biroli et al., 2024; Li & Chen, 2024; Yu & Huang, 2024; Kadkhodaie et al., 2024; Zhang et al., 2024). Raya & Ambrogioni (2023) reveals a spontaneous symmetry breaking in diffusion models, dividing the generative dynamics into two phases: a linear steady-state around a central fixed point and an attractor dynamics towards the data manifold. They linked the fixed points of the Fokker-Planck equations to moments of spontaneous symmetry breaking in the Hessian of the potential functions and demonstrated an end-to-end asymptotic analysis in a simple discrete distribution supported on two points and some other toy examples. The authors also propose a Gaussian late initialisation scheme which improves model performance, generation efficiency, and increases sample diversity. The concurrent work (Li & Chen, 2024) introduces a theoretical framework to understand phase transitions (they coined the term "critical windows" to describe the narrow time intervals in the generation during which specific features of the final image sample emerge, such as image class or background colour). The authors propose a formal non-asymptotic framework to study these windows, focusing on data from a mixture of strongly log-concave densities. They show that these windows can be provably bounded in terms of certain measures of inter- and intra-group separation. Biroli et al. (2024) employs statistical physics to identify three distinct dynamical regimes: initial noise, "speciation" transition, and "collapse" transition. Sclocchi et al. (2024) examines the hierarchical structure of data in diffusion models and identifies phase transitions in the generative process with sudden drops in high-level feature reconstruction probability whereas the smooth evolution of low-level feature reconstruction. Georgiev et al. (2023) focuses on data attribution to provide a framework for identifying specific training examples that influence generated images. These previous approaches, ranging from empirical studies to theoretical frameworks, provide valuable insights into phase transitions in generative dynamics; however, many of these methods face the challenge of requiring assumptions about the data. Our method offers new insights into phase transitions derived uniquely from the geometric structure of arbitrary data manifolds. Furthermore, our framework is essentially parallel to prior approaches, and therefore we expect to advance our understanding of phase transitions by deepening the relationship between the findings of previous research and our theoretical framework. A yet another recent work (Ikeda et al., 2024), while not explicitly addressing phase transitions, outlines connection between diffusion models and non-equilibrium thermodynamics, featuring interesting discussions on the relationships between noise scheduling, generation quality, entropy generation rates, and optimal transport. Other intriguing studies from a physics perspective include path integral interpretation of stochastic trajectories (Hirono et al., 2024) and Bayes-optimal denoising interpretation incorporating a spin-glass perspective (Ghio et al., 2023).
- **Geometrical approaches**: There are some geometrical perspectives on diffusion models inspired our work (Chung et al., 2022; Wenliang & Moran, 2023; Chen et al., 2023a; Park et al., 2023; Ghimire et al., 2023; Chen et al., 2023c; Okawa et al., 2023; Oko et al., 2023).
- **Other theories to understand diffusion and generation processes**: A deeper understanding of these processes is essential for advancing theoretical research and practical applications, such

as generation control through prompting and interpolation. Recent studies have delved into the underlying mechanisms of diffusion and generation trajectories to identify optimal intervention points during the generation process, which can help achieve desired data outputs. While flow-matching algorithms have shown promise, in the practical user cases, diffusion models surprisingly sometimes outperform the flow-matching, underscoring the need to understand the factors contributing to this superior performance. There are several works on convergence guarantees for diffusion models (Bortoli et al., 2021; Bortoli, 2022; Block et al., 2020; Chen et al., 2023b; Lee et al., 2022; Liu et al., 2022; Pidstrigach, 2022; Wibisono & Yang, 2022; Chen et al., 2023e; Lee et al., 2023; Li et al., 2023a; Benton et al., 2023a;b; Chen et al., 2023d; Li et al., 2024).

- **Flow matching techniques**: Flow matching algorithms (Lipman et al., 2023; Tong et al., 2024) are yet another prominent techniques in generative modelling. They are closely related to diffusion models as flow matching often leverages diffusion paths for training, in which optimal transport via ordinary differential equations (ODEs) yields straighter trajectories. It is very interesting to consider the influence on the quality and diversity of generated samples or critical dynamics such as spontaneous symmetry breaking. Our method may have the potential to analyse these aspects. Such generative models considering a transport from one distribution to another are expected to continue to develop, and geometric interpretations will further contribute to improving interpretability, efficiency, and control to ensure safety.

## B  SOCIAL IMPACTS

- **Green AI (Environmental Impact)**: Reducing the high energy consumption of diffusion models during both training and generation is crucial. The exponential increase in computational demands due to the growing use of diffusion models in industry poses significant environmental concerns. Optimising these models can lead to more sustainable AI practices, addressing the urgent need for eco-friendly AI technologies. Recent studies emphasise the need for environmental sustainability in AI, focusing on reducing the energy consumption and carbon footprint of AI models Verdecchia et al. (2023).
- **Fairness, AI Safety and Alignment**: Ensuring AI safety and alignment is critical. This includes improving the mechanistic interpretability of diffusion models, optimising control to prevent undesirable behaviours, and mitigating risks such as hallucinations and adversarial attacks. Effective control mechanisms and interpretability can enhance trust and safety in AI applications. Matsumoto et al. (2023) report that the diffusion time is the crucial for mitigating the membership inference attacks (MIAs) on diffusion models (Pang et al., 2023; Pang & Wang, 2023; Duan et al., 2023; Tang et al., 2023; Fu et al., 2023; Dubinski et al., 2024; Kong et al., 2023). Raya & Ambrogioni (2023) and Li & Chen (2024) show that phase transitions help understanding and controlling diversity in generation and Li & Chen (2024) also examines some relationship between phase transitions and MIAs.

## C  MATHEMATICAL SUPPLEMENTARIES

In this appendix, we quickly recall basic mathematical concepts and facts concerned with Linear Algebra and Manifold Theory. See, e.g., Lee (2013) for a detail of Manifold Theory.

### C.1  FORMAL OPERATIONS IN LINEAR ALGEBRA

For the Euclidean space $\mathbb{R}^d$ and its linear subspace $V \subset \mathbb{R}^d$, let $V^\perp$ denote the orthogonal complement of $V$ in $\mathbb{R}^d$.

**Proposition C.1.** *Let $V$ and $W$ be subspaces of $\mathbb{R}^n$. Then the following hold:*

*(1) $V \subset W$ if and only if $V^\perp \supset W^\perp$;*

*(2) $V^\perp \cap W^\perp = (V + W)^\perp$.*

### C.2  DIFFERENTIABLE MANIFOLDS

In this paper, as *manifolds*, we treat only 'submanifolds of the Euclidean space $\mathbb{R}^d$'. So we adapt the following definition.

**Definition C.2.** A subset $\mathcal{M}$ of $\mathbb{R}^d$ is called an $n$-*dimensional manifold*, if for each point $\boldsymbol{x} \in \mathcal{M}$, there is an open neighbourhood $U$ of $\boldsymbol{x}$ in $\mathbb{R}^d$, an open subset of $V$ in $\mathbb{R}^d = \mathbb{R}^n \times \mathbb{R}^{d-n}$, and a diffeomorphism $\phi\colon U \to V$ such that $\phi(\mathcal{M} \cap U) = V \cap (\mathbb{R}^n \times \{\boldsymbol{0}\})$. We call the map $\phi$ a *chart* on $\mathcal{M}$ around $\boldsymbol{x}$.

**Definition C.3.** Let $\mathcal{M} \subset \mathbb{R}^d$ be a manifold and $\boldsymbol{x} \in \mathcal{M}$ be a point. Then the *tangent space* $T_{\boldsymbol{x}}\mathcal{M}$ to $\mathcal{M}$ at $\boldsymbol{x}$ is defined as the set consisting of all velocity vectors of curves on $\mathcal{M}$ through $\boldsymbol{x}$, that is,

$$T_{\boldsymbol{x}}\mathcal{M} = \left\{ \frac{d\gamma}{dt}(0) \;\middle|\; \gamma\colon (-\epsilon, \epsilon) \to \mathcal{M},\ \gamma(0) = \boldsymbol{x} \right\}.$$

Notice that the tangent space forms a linear subspace of $\mathbb{R}^d$.

**Definition C.4.** Let $\mathcal{M} \subset \mathbb{R}^d$ and $\mathcal{M}' \subset \mathbb{R}^{d'}$ be manifolds, and let $F\colon \mathcal{M} \to \mathcal{M}'$ be a differentiable map (i.e., there is an extension $\tilde{F}\colon U \to \mathbb{R}^{d'}$ of $F$ which is a differentiable map on an open set $U$ of $\mathbb{R}^d$). Then the *differential* $dF_{\boldsymbol{x}}$ of $F$ at $\boldsymbol{x}$ is defined as the linear map

$$dF_{\boldsymbol{x}}\colon T_{\boldsymbol{x}}\mathcal{M} \to T_{F(\boldsymbol{x})}\mathcal{M}', \quad dF_{\boldsymbol{x}}\left(\frac{d\gamma}{dt}(0)\right) = \frac{d(F \circ \gamma)}{dt}(0).$$

**Remark C.5.** Take charts $\phi\colon U \to V$ and $\psi\colon U' \to V'$ on $\mathcal{M}$ and $\mathcal{M}'$, respectively. Also let $(x_1, \ldots, x_n)$ and $(y_1, \ldots, y_{n'})$ denote the coordinate on $V \subset \mathbb{R}^n$ and $V' \subset \mathbb{R}^{n'}$, respectively. Then the diffenrential $dF_{\boldsymbol{x}}$ is represented by the Jacobi matrix

$$\frac{\partial(\psi \circ F \circ \phi^{-1})}{\partial \boldsymbol{x}}(\boldsymbol{x}) = \left[ \frac{\partial(\psi \circ F \circ \phi^{-1})_i}{\partial x_j}(\boldsymbol{x}) \right]$$

of the map $\psi \circ F \circ \phi^{-1}\colon V \to V'$ at the point $\boldsymbol{x} \in \mathcal{M}$.

**Definition C.6.** Let $F\colon \mathcal{M} \to \mathcal{M}'$ be a differentiable map between manifolds. A point $\boldsymbol{x} \in \mathcal{M}$ is called a *regular point* (resp. a *critical point*) if the differential $dF_{\boldsymbol{x}}\colon T_{\boldsymbol{x}}\mathcal{M} \to T_{F(\boldsymbol{x})}\mathcal{M}'$ is surjective (resp. not surjective). A point $\boldsymbol{y} \in \mathcal{M}'$ is called a *regular value* (resp. a *critical value*) if every point $\boldsymbol{x} \in \mathcal{M}$ satisfying that $F(\boldsymbol{x}) = \boldsymbol{y}$ is a regular point of $F$ (resp. or not).

The following is essentially a consequence of Implicit Function Theorem.

**Theorem C.7** (cf. Lee (2013)[Corollary 5.14]). *Let* $\boldsymbol{F}\colon \mathbb{R}^d \to \mathbb{R}^{d'}$ *be a differentiable map and* $\boldsymbol{y} \in \mathbb{R}^{d'}$ *a regular value of* $\boldsymbol{F}$. *Then the level set*

$$\boldsymbol{F}^{-1}(\boldsymbol{y}) = \{\boldsymbol{x} \in \mathbb{R}^d \mid \boldsymbol{F}(\boldsymbol{x}) = \boldsymbol{y}\} \subset \mathbb{R}^d$$

*forms a* $(d - d')$-*dimensional manifold.*

**Remark C.8** (explicit description of the tangent spaces to a manifold). Consider the same setup of Theorem C.7 and denote $\boldsymbol{F} = (F_1, \ldots, F_{d'})$. Then the normal to the tangent space $T_{\boldsymbol{x}}\mathcal{M}$ coincides with

$$\left\langle \frac{\partial F_1}{\partial \boldsymbol{x}}(\boldsymbol{x})^T, \cdots, \frac{\partial F_{d'}}{\partial \boldsymbol{x}}(\boldsymbol{x})^T \right\rangle_{\mathbb{R}},$$

which is spanned by the gradient vectors of components of $\boldsymbol{F}$. Therefore the tangent space itself is noting but its orthogonal complement, i.e.,

$$T_{\boldsymbol{x}}\mathcal{M} = \left\langle \frac{\partial F_1}{\partial \boldsymbol{x}}(\boldsymbol{x})^T, \cdots, \frac{\partial F_{d'}}{\partial \boldsymbol{x}}(\boldsymbol{x})^T \right\rangle_{\mathbb{R}}^{\perp}.$$

**Definition C.9.** A differentiable map $F\colon \mathcal{M} \to \mathcal{M}'$ is called an *embedding* if its differential $dF\colon T_{\boldsymbol{x}}\mathcal{M} \to T_{F(\boldsymbol{x})}\mathcal{M}'$ is injective for every point $\boldsymbol{x} \in \mathcal{M}$ and the restriction $F\colon \mathcal{M} \to f(\mathcal{M})$ is a topological homeomorphism (i.e. there is the inverse map $F^{-1}$, and both $F$ and $F^{-1}$ are continuous).

## C.3 Tubular neighbourhoods

Let $\mathcal{M} \subset \mathbb{R}^d$ be a manifold. Recall the normal bundle

$$N\mathcal{M} = \{(\boldsymbol{x}, \boldsymbol{v}) \in \mathbb{R}^d \times \mathbb{R}^d \mid \boldsymbol{x} \in \mathcal{M}, \boldsymbol{v} \perp T_{\boldsymbol{x}}\mathcal{M}\}$$

to $\mathcal{M}$ and the endpoint map

$$E\colon N\mathcal{M} \to \mathbb{R}^d, \quad E(\boldsymbol{x}, \boldsymbol{v}) = \boldsymbol{x} + \boldsymbol{v},$$

which are defined in §3 (Definitions 3.2 and 3.3).

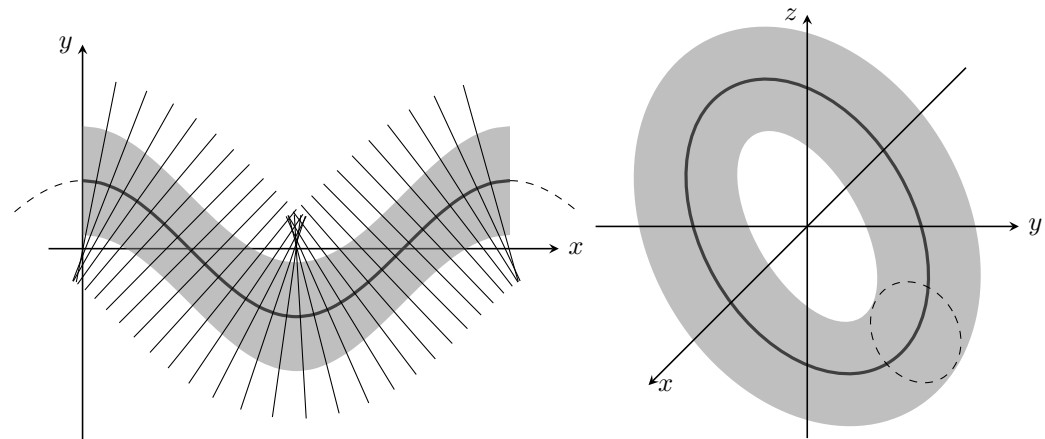

Figure 10: Image under $E$ and tubular neighbourhood of the cosine curve in $\mathbb{R}^2$

Figure 11: Tubular neighbourhood of $S^1$ embedded in $\mathbb{R}^3$

**Definition C.10** (Tubular neighbourhood). A tubular neighbourhood of $\mathcal{M}$ is a neighbourhood of $\mathcal{M}$ in $\mathbb{R}^d$ that is the diffeomorphic image under $E$ of an open subset $V \subset N\mathcal{M}$ of the form

$$V = \{(\boldsymbol{x}, \boldsymbol{v}) \in N\mathcal{M} \mid \|\boldsymbol{v}\| < \delta(\boldsymbol{x})\}$$

for some positive continuous function $\delta \colon \mathcal{M} \to \mathbb{R}$.

**Theorem C.11** (Theorem 6.24 in Lee (2013)). *Every manifold embedded in $\mathbb{R}^d$ has a tubular neighbourhood.*

*Proof.* Let $\mathcal{M}_0$ denote the subset $\{(\boldsymbol{x}, 0) \mid \boldsymbol{x} \in \mathcal{M}\} \subset N\mathcal{M}$. Fix a point $\boldsymbol{x} \in \mathcal{M}$. Since both differentials $dE|_{T_{(\boldsymbol{x},0)}\mathcal{M}_0} \colon T_{(\boldsymbol{x},0)}\mathcal{M}_0 \to T_{\boldsymbol{x}}\mathcal{M}$ and $dE|_{N_{\boldsymbol{x}}\mathcal{M}} \colon N_{\boldsymbol{x}}\mathcal{M} \to N_{\boldsymbol{x}}\mathcal{M}$ are isomorphisms, we have that $dE \colon T_{(\boldsymbol{x},0)}N\mathcal{M} \to \mathbb{R}^d$ is also an isomorphism. By Inverse Function Theorem, the map $E$ is a diffeomorphism on a neighbourhood of $(\boldsymbol{x}, 0) \in N\mathcal{M}$. We can take the neighbourhood to be of the form $V_\delta(\boldsymbol{x}) = \{(\boldsymbol{x}', \boldsymbol{v}') \in N\mathcal{M} \mid \|\boldsymbol{x} - \boldsymbol{x}'\| < \delta, \|\boldsymbol{v}'\| < \delta\}$ for some $\delta > 0$. Let $\rho(\boldsymbol{x})$ denote the supremum of all such $\delta < 1$. We can prove that the function $\rho \colon \mathcal{M} \to \mathbb{R}$ is positive and continuous.

Now consider the open subset $V = \{(\boldsymbol{x}, \boldsymbol{v}) \in N\mathcal{M} \mid \|\boldsymbol{v}\| < \frac{1}{2}\rho(\boldsymbol{x})\}$ of $N\mathcal{M}$. Then the map $E$ is injective on $V$, and hence $E|_V \colon V \to \mathbb{R}^d$ is a smooth embedding. Thus $E(V)$ is a tubular neighbourhood of $\mathcal{M}$. $\qquad\square$

# D  THEORETICAL SUPPLEMENTARIES OF SECTION 3

## D.1  PROOF OF THEOREM 3.7

Under the setup of Theorem 3.7, put $k = d - n$ and we define a map $\boldsymbol{\varphi} \colon \mathbb{R}^d \times \mathbb{R}^d \to \mathbb{R}^{N+k}$ by

$$\boldsymbol{\varphi}(\boldsymbol{x}, \boldsymbol{v}) = (\boldsymbol{F}(\boldsymbol{x}), \varphi_1(\boldsymbol{x}, \boldsymbol{v}), \cdots, \varphi_N(\boldsymbol{x}, \boldsymbol{v})).$$

Notice that the normal bundle $N\mathcal{M}$ to $\mathcal{M}$ is expressed by

$$N\mathcal{M} = \boldsymbol{\varphi}^{-1}(\boldsymbol{0}) = \{(\boldsymbol{x}, \boldsymbol{v}) \in \mathbb{R}^d \times \mathbb{R}^d \mid \boldsymbol{\varphi}(\boldsymbol{x}, \boldsymbol{v}) = \boldsymbol{0}\}.$$

Hence the tangent space $T_{(\boldsymbol{x},\boldsymbol{v})}N\mathcal{M} \subset \mathbb{R}^d \times \mathbb{R}^d$ to $N\mathcal{M}$ at a point $(\boldsymbol{x}, \boldsymbol{v})$ coincides with the orthogonal complement of

$$\left\langle \begin{bmatrix} \frac{\partial F_1}{\partial \boldsymbol{x}}^T \\ \boldsymbol{0} \end{bmatrix}, \cdots, \begin{bmatrix} \frac{\partial F_d}{\partial \boldsymbol{x}}^T \\ \boldsymbol{0} \end{bmatrix}, \begin{bmatrix} \frac{\partial \varphi_1}{\partial \boldsymbol{x}}^T \\ \frac{\partial \varphi_1}{\partial \boldsymbol{v}}^T \end{bmatrix}, \cdots, \begin{bmatrix} \frac{\partial \varphi_N}{\partial \boldsymbol{x}}^T \\ \frac{\partial \varphi_N}{\partial \boldsymbol{v}}^T \end{bmatrix} \right\rangle_{\mathbb{R}}$$

in $\mathbb{R}^d \times \mathbb{R}^d$ (cf. Remark C.8).

We now employ the Method of Lagrange multiplier. That is, we paraphrase the condition that a point $(\boldsymbol{x}, \boldsymbol{v}) \in N\mathcal{M}$ is a critical point of the endpoint map

$$E = E_0|_{N\mathcal{M}} \colon N\mathcal{M} \to \mathbb{R}^d$$

(i.e., the differential $dE_{(\boldsymbol{x},\boldsymbol{v})} \colon T_{(\boldsymbol{x},\boldsymbol{v})} N\mathcal{M} \to \mathbb{R}^d$, which is a linear map, is degenerate) as follows. First, the condition is equivalent to that there exists a non-zero vector of $T_{(\boldsymbol{x},\boldsymbol{v})} N\mathcal{M}$ which vanishes by the differential $(dE_0)_{(\boldsymbol{x},\boldsymbol{v})} \colon \mathbb{R}^d \times \mathbb{R}^d \to \mathbb{R}^d$, i.e.,

$$T_{(\boldsymbol{x},\boldsymbol{v})} N\mathcal{M} \cap \mathrm{Ker}(dE_0)_{(\boldsymbol{x},\boldsymbol{v})} \supsetneq \{\boldsymbol{0}\}.$$

Moreover, we have the following:

$$T_{(\boldsymbol{x},\boldsymbol{v})} N\mathcal{M} \cap \mathrm{Ker}(dE_0)_{(\boldsymbol{x},\boldsymbol{v})} \supsetneq \{\boldsymbol{0}\}$$

$$\Longleftrightarrow \left\langle \begin{bmatrix} \frac{\partial F_1}{\partial \boldsymbol{x}}^T \\ \boldsymbol{0} \end{bmatrix}, \cdots, \begin{bmatrix} \frac{\partial F_k}{\partial \boldsymbol{x}}^T \\ \boldsymbol{0} \end{bmatrix}, \begin{bmatrix} \frac{\partial \varphi_1}{\partial \boldsymbol{x}}^T \\ \frac{\partial \varphi_1}{\partial \boldsymbol{v}}^T \end{bmatrix}, \cdots, \begin{bmatrix} \frac{\partial \varphi_N}{\partial \boldsymbol{x}}^T \\ \frac{\partial \varphi_N}{\partial \boldsymbol{v}}^T \end{bmatrix} \right\rangle_{\mathbb{R}}^{\perp} \cap \left\langle \begin{bmatrix} \boldsymbol{e}_1 \\ \boldsymbol{e}_1 \end{bmatrix}, \cdots, \begin{bmatrix} \boldsymbol{e}_d \\ \boldsymbol{e}_d \end{bmatrix} \right\rangle_{\mathbb{R}}^{\perp} \supsetneq \{\boldsymbol{0}\}$$

$$\Longleftrightarrow \left\langle \begin{bmatrix} \frac{\partial F_1}{\partial \boldsymbol{x}}^T \\ \boldsymbol{0} \end{bmatrix}, \cdots, \begin{bmatrix} \frac{\partial F_k}{\partial \boldsymbol{x}}^T \\ \boldsymbol{0} \end{bmatrix}, \begin{bmatrix} \frac{\partial \varphi_1}{\partial \boldsymbol{x}}^T \\ \frac{\partial \varphi_1}{\partial \boldsymbol{v}}^T \end{bmatrix}, \cdots, \begin{bmatrix} \frac{\partial \varphi_N}{\partial \boldsymbol{x}}^T \\ \frac{\partial \varphi_N}{\partial \boldsymbol{v}}^T \end{bmatrix} \right\rangle_{\mathbb{R}} + \left\langle \begin{bmatrix} \boldsymbol{e}_1 \\ \boldsymbol{e}_1 \end{bmatrix}, \cdots, \begin{bmatrix} \boldsymbol{e}_d \\ \boldsymbol{e}_d \end{bmatrix} \right\rangle_{\mathbb{R}} \subsetneq \mathbb{R}^d \times \mathbb{R}^d,$$

where $\{\boldsymbol{e}_1, \cdots, \boldsymbol{e}_d\}$ denotes the standard basis of $\mathbb{R}^d$. Here we used a property on orthogonal complements (see Appendix C.1).

Finally, it is equivalent to that the matrix

$$\begin{bmatrix} \frac{\partial \boldsymbol{F}}{\partial \boldsymbol{x}}^T & \frac{\partial \varphi_1}{\partial \boldsymbol{x}}^T & \cdots & \frac{\partial \varphi_N}{\partial \boldsymbol{x}}^T & E_d \\ O_{n,d} & \frac{\partial \varphi_1}{\partial \boldsymbol{v}}^T & \cdots & \frac{\partial \varphi_N}{\partial \boldsymbol{v}}^T & E_d \end{bmatrix}$$

is degenerate. Performing elementary row and column operations, and by the definition of $R_1(\mathcal{M})$, the conclusion of Theorem 3.7 follows. $\square$

### D.2 CURVATURE AND THE FIRST INJECTIVITY RADIUS OF A CURVE

Let $\mathcal{M}$ be a curve in $\mathbb{R}^d$, i.e., a one-dimensional manifold embedded in $\mathbb{R}^d$. We see that, in this case, the first injectivity radius $R_1(\mathcal{M})$ is derived from the curvature of $\mathcal{M}$ as follows.

**Definition D.1.** Let $\gamma \colon \mathbb{R} \to \mathbb{R}^d$ be an arc-length parametrization of the curve $\mathcal{M}$, i.e., $\left\| \frac{d\gamma}{ds} \right\| \equiv 1$. Then the curvature $\kappa$ of $\mathcal{M}$ at a point $p = \gamma(s) \in \mathcal{M}$ is defined by the Euclidean norm of the second order derivative $\frac{d^2\gamma}{ds^2}(s)$.

**Proposition D.2.** *Assume that $n = 1$. Let $\gamma \colon \mathbb{R} \to \mathbb{R}^d$ be an arbitrary regular parametrization of the curve $\mathcal{M}$. Then the curvature $\kappa$ of $\mathcal{M}$ is computed by*

$$\kappa(\gamma(u)) = \frac{\sqrt{\|\gamma'(u)\|^2 \|\gamma''(u)\|^2 - \langle \gamma'(u), \gamma''(u) \rangle^2}}{\|\gamma'(u)\|^3}, \tag{11}$$

*where $'$ denotes the differential by $u$.*

Although this is a well-known fact, we show it briefly as follows.

*Proof.* Let $s$ and $u$ denote an arc-length parameter and an arbitrary regular parameter of the curve $\mathcal{M}$. Since it holds that

$$\gamma' = s' \frac{d\gamma}{ds}, \tag{12}$$

we also have that

$$\gamma'' = s'' \cdot \frac{d\gamma}{ds} + (s')^2 \, \kappa \cdot \nu, \tag{13}$$

where $\nu$ denotes the normalization of the vector $\dfrac{d^2\gamma}{ds^2}$. Since two vectors $\dfrac{d\gamma}{ds}$ and $\nu$ form an orthonormal frame of the curve $\mathcal{M}$, it holds that

$$\|\gamma''\|^2 = (s'')^2 + (s')^4 \cdot \kappa^2. \tag{14}$$

Now notice the following: it holds that

$$\|\gamma'\|^2 = (s')^2 \tag{15}$$

by Equation (12), and hence

$$\langle \gamma', \gamma'' \rangle = s' \cdot s''. \tag{16}$$

Applying Equations (15) and (16) to Equation (14), we have the claim. $\qquad\square$

**Theorem D.3.** *Assume that $n = 1$. Let $\kappa$ denote the curvature of $\mathcal{M}$. Then $R_1(\mathcal{M})$ coincides with the infimum of radii of curvature $1/\kappa$.*

*Proof.* See Lemma 1 of Litherland et al. (1999). $\qquad\square$

### D.3 COMMENTS ON THE COMPUTATION OF THE SECOND INJECTIVITY RADIUS

In this paper we used the definition of $R_2(\mathcal{M})$ as-is for the numerical estimation.

We note that one can weaken the condition appearing to the definition of $R_2(\mathcal{M})$ as follows.

**Proposition D.4.** The second injectivity radius $R_2(\mathcal{M})$ coincides with the infimum of the set

$$\left\{ \frac{1}{2}\|\boldsymbol{x}_1 - \boldsymbol{x}_2\| \ \middle| \ \begin{array}{l} \boldsymbol{x}_1, \boldsymbol{x}_2 \in \mathcal{M}, \boldsymbol{x}_1 \neq \boldsymbol{x}_2, \\ \text{and } \boldsymbol{x}_1 - \boldsymbol{x}_2 \perp T_{\boldsymbol{x}_1}\mathcal{M} \end{array} \right\}.$$

*Proof.* See §4 of Litherland et al. (1999). $\qquad\square$

We also have a comment on $R_2(\mathcal{M})$. Numerically, it seems to be possible to compute $R_2(\mathcal{M})$ by using the *persistent homology* of the given data cloud. Indeed, the topology of the $\epsilon$-neighbourhood of the data cloud might change when two tubes touch each other.

## E OTHER EXAMPLES OF INJECTIVITY RADII

We have already seen that Theorem 3.7 works in the case that a data manifold is the unit circle $S^1 \subset \mathbb{R}^2$. In this appendix, we verify the theorem by observing other typical manifolds.

### E.1 TORUS $T^2$

Let $r' > r > 0$, and define a function $F : \mathbb{R}^3 \to \mathbb{R}$ by

$$F(x, y, z) = (\sqrt{x^2 + y^2} - r')^2 + z^2 - r^2.$$

Then we have a torus $T^2 = F^{-1}(0)$ embedded in $\mathbb{R}^3$. We can see that vector fields

$$\boldsymbol{t}_1 = (-y, x, 0), \quad \boldsymbol{t}_2 = (xz, yz, r'\sqrt{x^2 + y^2} - x^2 - y^2)$$

satisfy the assumption in Theorem 3.7. Then the matrix $L_{T^2}((x, y, z), (v_1, v_2, v_3))$ is calculated as follows:

$$L_{T^2}((x, y, z), (v_1, v_2, v_3))$$

$$= \begin{bmatrix} 2(\sqrt{x^2+y^2} - r')\frac{x}{\sqrt{x^2+y^2}} & v_2 + y & zv_1 - 2xv_3 - xz + \frac{r'xv_3}{\sqrt{x^2+y^2}} \\ 2(\sqrt{x^2+y^2} - r')\frac{y}{\sqrt{x^2+y^2}} & -v_1 - x & zv_2 - 2yv_3 - yz + \frac{r'yv_3}{\sqrt{x^2+y^2}} \\ z & 0 & xv_1 + yv_2 + x^2 + y^2 - r'\sqrt{x^2+y^2} \end{bmatrix}.$$

We now parametrise the torus $T^2$ by $(x, y, z) = ((r' + r \cos t) \cos u, (r' + r \cos t) \sin u, \cos t)$ of $T^2 \subset \mathbb{R}^3$. Then the vector $(v_1, v_2, v_3)$ makes $L_{T^2}((x, y, z), (v_1, v_2, v_3))$ degenerate if and only if

$$(v_1, v_2, v_3) = -(r \cos t \cos u, r \cos t \sin u, r \sin t) \quad \text{or}$$

$$(v_1, v_2, v_3) = -\frac{r' + r \cos t}{r \cos t}(r \cos t \cos u, r \cos t \sin u, r \sin t) \quad \left(t \neq \pm\frac{\pi}{2}\right).$$

Hence we obtain $R_1(T^2) = \min\{r, r' - r\}$. Moreover we can see that $R_2(T^2) = \min\{r, r' - r\}$. Thus the injectivity radius is $R(T^2) = \min\{r, r' - r\}$.

### E.2 Unit Sphere $S^2$

Define a function $F \colon \mathbb{R}^3 \to \mathbb{R}$ by

$$F(x, y, z) = x^2 + y^2 + z^2 - 1.$$

Then we have $S^2 = F^{-1}(0)$. Considering the rotation in $\mathbb{R}^3$ around coordinate axes, we see that vector fields

$$\boldsymbol{t}_1 = (-y, x, 0), \quad \boldsymbol{t}_2 = (-z, 0, x), \quad \boldsymbol{t}_3 = (0, -z, y).$$

satisfy the assumption of Theorem 3.7. (Here notice that the number of vector fields which we desire is needed to be greater than 2, by topological reason.) Then the matrix $L_{S^2}((x, y, z), (v_1, v_2, v_3))$ is calculated as follows:

$$L_{S^2}((x, y, z), (v_1, v_2, v_3))$$

$$= \begin{bmatrix} 2x & v_2 + y & v_3 + z & 0 \\ 2y & -v_1 - x & 0 & v_3 + z \\ 2z & 0 & -v_1 - x & -v_2 - y \end{bmatrix}.$$

This matrix is degenerate on $((x, y, z), (v_1, v_2, v_3)) \in NS^2$ if and only if $(v_1, v_2, v_3) = (-x, -y, -z)$. Hence we obtain $R_1(S^2) = \sqrt{(-x)^2 + (-y)^2 + (-z^2)} = 1$. Moreover it is clear that $R_2(S^2) = 1$. Thus the injectivity radius is $R(S^2) = 1$.

### E.3 Unit $n$-Sphere $S^n$

As the final example, we observe the unit $n$-sphere. Define a function $F \colon \mathbb{R}^{n+1} \to \mathbb{R}$ by

$$F(x_1, x_2, \ldots, x_{n+1}) = x_1^2 + x_2^2 + \cdots + x_{n+1}^2 - 1.$$

Then we have $S^n = F^{-1}(0)$. Considering gradient vector fields of the height functions $(x_1, x_2, \ldots, x_{n+1}) \mapsto x_j$ $(j = 1, 2, \ldots, n+1)$, we see that vector fields

$$\boldsymbol{t}_j = (-x_1 x_j, \ldots, -x_{j-1} x_j, 1 - x_j^2, -x_{j+1} x_j, \ldots, -x_{n+1} x_j) \quad (j = 1, 2, \ldots, n+1)$$

satisfy the assumption of Theorem 3.7. Then the matrix $L_{S^n}(\boldsymbol{x}, \boldsymbol{v})$ is calculated as follows:

$$L_{S^n}(\boldsymbol{x}, \boldsymbol{v})$$

$$= \begin{bmatrix} 2x_1 & -2x_1 - \sum_{i \neq 1} x_i v_i + x_1^2 - 1 & & -x_{n+1}v_1 + x_{n+1}x_1 \\ \vdots & -x_1 v_2 + x_1 x_2 & \ddots & \vdots \\ \vdots & \vdots & \ddots & -x_{n+1}v_n + x_{n+1}x_n \\ 2x_{n+1} & -x_1 v_{n+1} + x_1 x_{n+1} & -2x_{n+1}v_{n+1} - \sum_{i \neq n+1} x_i v_i + x_{n+1}^2 - 1 \end{bmatrix},$$

where $\boldsymbol{x} = (x_1, x_2, \cdots, x_{n+1})$, $\boldsymbol{v} = (v_1, v_2, \cdots, v_{n+1})$. Now notice that for a point $\boldsymbol{x} \in S^n$ and a normal vector $\boldsymbol{v}$ to $\boldsymbol{x}$, there exists a scalar $c \in \mathbb{R}$ such that $\boldsymbol{v} = c\boldsymbol{x}$. Using it and performing the elementary row and column operations, the matrix $L_{S^n}(\boldsymbol{x}, \boldsymbol{v})$ is transformed as follows:

$$\begin{bmatrix} x_1 & -c\sum_i x_i^2 - 1 & 0 & \cdots & 0 \\ \vdots & 0 & \ddots & \ddots & \vdots \\ \vdots & \vdots & \ddots & \ddots & 0 \\ x_{n+1} & 0 & \cdots & 0 & -c\sum_i x_i^2 - 1 \end{bmatrix}.$$

Hence the vector $\boldsymbol{v}$ makes the matrix $L_{S^n}(\boldsymbol{x}, \boldsymbol{v})$ degenerate if and only if $c = -1$. Hence we obtain $R_1(S^n) = \|-\boldsymbol{x}\| = 1$. Moreover it is clear that $R_2(S^n) = 1$. Thus the injectivity radius is $R(S^n) = 1$.

---

**Algorithm 1** Algorithm for estimating the injectivity radius (AEIR)

---

**Input:** data $\mathcal{D} \subset \mathbb{R}^d$

**Step 0:** Estimate a map $\boldsymbol{F} = (F_1, \ldots, F_{d-n}) \colon \mathbb{R}^d \to \mathbb{R}^{d-n}$ such that the point $\boldsymbol{0}$ is a regular value of $\boldsymbol{F}$ and the manifold $\boldsymbol{F}^{-1}(\boldsymbol{0}) \subset \mathbb{R}^d$ approximates data $\mathcal{D}$. Put $\mathcal{M} := \boldsymbol{F}^{-1}(\boldsymbol{0})$.

**Step 1:** Estimate vector fields $\boldsymbol{t}_1, \boldsymbol{t}_2, \ldots, \boldsymbol{t}_N$ ($n \leq N$) defined near $\mathcal{M}$ such that for every $\boldsymbol{x} \in \mathcal{M}$ the vectors $\boldsymbol{t}_1(\boldsymbol{x}), \boldsymbol{t}_2(\boldsymbol{x}), \ldots, \boldsymbol{t}_N(\boldsymbol{x})$ span the tangent space $T_{\boldsymbol{x}}\mathcal{M}$.

**Step 2:** Put $g_i \colon \mathbb{R}^d \times \mathbb{R}^d \to \mathbb{R}$, $g_i(\boldsymbol{x}, \boldsymbol{v}) := \langle \boldsymbol{v}, \boldsymbol{t}_i(\boldsymbol{x}) \rangle$ ($i = 1, 2, \ldots, N$). Calculate the matrix

$$
[A_1, \cdots, A_{d-n}, B_1, \cdots, B_N] := \begin{bmatrix} \frac{\partial F_1}{\partial x_1} & \cdots & \frac{\partial F_{d-n}}{\partial x_1} & \frac{\partial \varphi_1}{\partial x_1} - \frac{\partial \varphi_1}{\partial v_1} & \cdots & \frac{\partial \varphi_N}{\partial x_1} - \frac{\partial \varphi_N}{\partial v_1} \\ \vdots & \ddots & \vdots & \vdots & \ddots & \vdots \\ \frac{\partial F_1}{\partial x_d} & \cdots & \frac{\partial F_{d-n}}{\partial x_d} & \frac{\partial \varphi_1}{\partial x_d} - \frac{\partial \varphi_1}{\partial v_d} & \cdots & \frac{\partial \varphi_N}{\partial x_d} - \frac{\partial \varphi_N}{\partial v_d} \end{bmatrix},
$$

where $\boldsymbol{x} = (x_1, \cdots, x_d), \boldsymbol{v} = (v_1, \cdots, v_d)$.

**Step 3:** Collect sufficient amount of samples from the set

$$
\left\{ (\boldsymbol{x}, \boldsymbol{v}) \in \mathbb{R}^d \times \mathbb{R}^d \;\middle|\; \begin{array}{l} \boldsymbol{F}(\boldsymbol{x}) = \boldsymbol{0}, g_i(\boldsymbol{x}, \boldsymbol{v}) = 0 \; (i = 1, 2, \cdots, N), \\ \det[A_1, \ldots, A_{d-n}, B_{i_1}, \ldots, B_{i_n}] = 0 \\ (1 \leq i_1 < \cdots < i_n \leq N) \end{array} \right\},
$$

and estimate $\min \|\boldsymbol{v}\|$ on the set. Put this value $R_1$.

**Step 4:** Collect sufficient amount of samples from the set

$$
\left\{ (\boldsymbol{x}_1, \boldsymbol{x}_2) \in \mathbb{R}^d \times \mathbb{R}^d \;\middle|\; \begin{array}{l} \boldsymbol{F}(\boldsymbol{x}_1) = \boldsymbol{F}(\boldsymbol{x}_2) = \boldsymbol{0}, \boldsymbol{x}_1 \neq \boldsymbol{x}_2, \\ \langle \boldsymbol{x}_1 - \boldsymbol{x}_2, \boldsymbol{t}_i(\boldsymbol{x}_1) \rangle = \langle \boldsymbol{x}_1 - \boldsymbol{x}_2, \boldsymbol{t}_i(\boldsymbol{x}_2) \rangle = 0 \\ (i = 1, 2, \cdots, N) \end{array} \right\},
$$

and estimate $\min \|\boldsymbol{x}_1 - \boldsymbol{x}_2\|$ on the set. Put this value $R_2$.

**Step 5:** Calculate $R := \min\{R_1, R_2\}$.

**Output:** $R$, which estimates $R(\mathcal{M})$.

---

Table 5: Estimated injectivity radii of various manifolds.

| DATA SET | $R_1$ | $R_2$ |
|---|---|---|
| $S^1$ | $1.005 \pm 0.003$ | $0.999 \pm 0.006$ |
| $S^2$ | $1.063 \pm 0.032$ | $0.997 \pm 0.038$ |
| $S^{128}$ | $1.068 \pm 0.023$ | $0.922 \pm 0.056$ |

## F  ALGORITHM FOR ESTIMATING THE INJECTIVITY RADIUS

In this appendix, we show the pseudo-algorithm for estimating the injectivity radius (see Algorithm 1) and some preliminary numerical experiments to verify the proposed algorithm.

### F.1  NUMERICAL EXPERIMENTS TO VALIDATE AIER

For the $S^1$, $S^2$, $S^{128}$ cases, the estimated $R_1$ and $R_2$ using the proposed algorithm are shown in Table 5. We first generate dataset using the exact generative equations and add some Gaussian noise. The $\boldsymbol{F}$ is then approximated using a neural network. The following Step 1 to Step 4 are executed using the neural network approximation $\boldsymbol{F}$. We note that we use the cosine similarity instead of inner products for the discrimination condition defined in the Step 4.

## G  NON EQUILIBRIUM THERMODYNAMICS AND PHASE TRANSITIONS

In Song et al. (2021), score-matching Hyvärinen (2005) and diffusion-based (Sohl-Dickstein et al., 2015; Ho et al., 2020) generative models have been unified into a single continuous-time score-based framework where the diffusion is driven by a stochastic differential equation. This framework relies on Anderson's Theorem (Anderson, 1982), which states that under certain Lipschitz conditions on the drift coefficient $f : \mathbb{R}^d \times \mathbb{R}^d \to \mathbb{R}^d$ and on the diffusion coefficient $g : \mathbb{R}^d \times \mathbb{R} \to \mathbb{R}^d \times \mathbb{R}^d$ and an integrability condition on the target distribution $p_0(x_0)$, a forward diffusion process governed by the SDE

$$dx_t = f_t(x_t)dt + g_t dw_t \tag{17}$$

where $w_t$ is a standard Wiener process. We could derive that probability distribution $p_t(x)$ of the forward SDE 17 satisfies the Fokker-Planck equation:

$$\frac{\partial}{\partial t} p_t(x) = \nabla_x \cdot [p_t(x)\nabla_x u_t(x)] \tag{18}$$

$$= p_t(x)\left(\nabla_x^2 u_t(x) + \nabla_x \ln p_t(x) \cdot \nabla_x u_t(x)\right) \tag{19}$$

where the potential $u_t(x)$ is defined as follows Raya & Ambrogioni (2023):

$$u_t(x) = -\int_{x_0}^{x} f_t(z)dz + \frac{g_t^2}{2}\ln p_t(x) \tag{20}$$

Here, we naturally consider the free energy for non-equilibrium thermodynamics Esposito & den Broeck (2011):

**Definition G.1.**  Non-equilibrium free energy in the system

$$\mathcal{F}_{\text{neq}}(t) := \int_{\mathbb{R}^d} p_t(x)u_t(x)dx. \tag{21}$$

**Theorem G.2.**  The non-equilibrium free energy can be rewritten as follows.

$$\mathcal{F}_{\text{neq}}(t) = \frac{g_t^2}{2}\int_{\mathbb{R}^d} p_t(x)\left[\ln\frac{p_t(x)}{p_{\text{eq}}(x)} + \ln p_{\text{eq}}(x_0)\right]dx. \tag{22}$$

*Proof.*  By the definition of the potential 20:

$$\nabla_x u_t(x) = -f_t(x) + \frac{g_t^2}{2}\nabla_x \ln p_t(x) \tag{23}$$

When the target system is in equilibrium, the solution $p_{\text{eq}}(x)$ of the Fokker-Planck equation 18 that satisfies the following equation exists:

$$\frac{\partial}{\partial t} p_{\text{eq}}(x) = \nabla_x \cdot \left[p_{\text{eq}}(x)\left(-f_t(x) + \frac{g_t^2}{2}\nabla_x \ln p_{\text{eq}}(x)\right)\right] = 0 \tag{24}$$

Therefore, we can rewrite the drift coefficient of the forward SDE 17 using $p_{\text{eq}}(x)$ as follows:

$$f_t(x) = \frac{g_t^2}{2}\nabla_x \ln p_{\text{eq}}(x) \tag{25}$$

From the above, we obtain the following relation.

$$u_t(x) = -\int_{x_0}^{x} f_t(z)dz + \frac{g_t^2}{2}\ln p_t(x) \tag{26}$$

$$= \frac{g_t^2}{2}\left[\ln\frac{p_t(x)}{p_{\text{eq}}(x)} + \ln p_{\text{eq}}(x_0)\right] \tag{27}$$

$$\square$$

On the other hand, the free energy in equilibrium thermodynamics is given by:

**Definition G.3.** Equilibrium free energy in the system

$$\mathcal{F}_{\text{eq}}(t) := \frac{g_t^2}{2} \ln p_{\text{eq}}(x_0) \tag{28}$$

Therefore, the two free energies have the following relationship:

**Theorem G.4.** From the non-negativity of KL-divergence, the following inequality is obtained.

$$\mathcal{F}_{\text{neq}}(t) - \mathcal{F}_{\text{eq}}(t) = \frac{g_t^2}{2} D_{\text{KL}}\left[p_t(x)\|p_{\text{eq}}(x)\right] \geq 0 \tag{29}$$

When the target system is in equilibrium at time $t = t_{\text{eq}}$, i.e., $\frac{\partial}{\partial t}p_t(x)\big|_{t=t_{\text{eq}}} = \frac{\partial}{\partial t}p_{\text{eq}}(x) = 0$, the following equality holds:

$$\mathcal{F}_{\text{neq}}(t_{\text{eq}}) - \mathcal{F}_{\text{eq}}(t_{\text{eq}}) = \frac{g_{t_{\text{eq}}}^2}{2} D_{\text{KL}}\left[p_{\text{eq}}(x)\|p_{\text{eq}}(x)\right] = 0 \tag{30}$$

According to Landau theory of phase transitions, phase transitions in equilibrium thermodynamics are identified when the higher-order derivatives of equilibrium free energy with respect to the order parameters $\lambda_1, \lambda_2, \ldots, \lambda_n$ exhibit discontinuities or divergences. This criterion serves as a fundamental indicator for detecting phase transitions within the framework of equilibrium statistical mechanics:

$$\frac{\partial^n \mathcal{F}_{\text{eq}}}{\partial \lambda_i^n} = 0 \tag{31}$$

On the other hand, it remains unclear whether a simple criterion for critical points, like the one mentioned above, exists for phase transition phenomena in non-equilibrium systems such as the diffusion processes represented by diffusion models.

In recent research Raya & Ambrogioni (2023), it has been demonstrated that the spontaneous symmetry breaking of the potential $u_t(x)$, plays a central role in understanding phase transition phenomena in the diffusion processes represented by diffusion models. Specifically, the spontaneous symmetry breaking of the potential $u_t(x)$ occurs when the first derivative $\nabla_x u_t(x)$ and the second derivative $\nabla_x^2 u_t(x)$ vanishes $\nabla_x u_t(x) = \nabla_x^2 u_t(x) = 0$ at the critical point of the space-time $(x, t) = (x_c, t_c)$, where the fixed point of the Fokker-Planck equation appear.

For instance, we consider a simple one-dimensional example Raya & Ambrogioni (2023) with a dataset consisting of two points $y_{-1} = -1$ and $y_1 = -y_{-1} = 1$ sampled with equal probability. Up to terms that are constant in $x$, the potential is given by the following expression:

$$u_t(x) = \beta(t)\left(-\frac{1}{4}x^2 - \ln\left(e^{-\frac{(x-\theta_t)^2}{2(1-\theta_t^2)}} + e^{-\frac{(x+\theta_t)^2}{2(1-\theta_t^2)}}\right)\right) \tag{32}$$

where $\beta(t) = \beta_{\min} + t(\beta_{\max} - \beta_{\min})$, $\beta_{\max} = 20$, $\beta_{\min} = 0.1$. At the critical point $(x, t) = (0, t_c)$, $t_c = 0.293$, the first derivative $\nabla_x u_t(x)$ and the second derivative $\nabla_x^2 u_t(x)$ vanishes $\nabla_x u_t(x) = \nabla_x^2 u_t(x) = 0$.

**Lemma G.5.** By the definition of the Fokker-Planck equation 18, the Fokker-Planck equation satisfies the following relations at the critical point $(x, t) = (x_c, t_c)$ Raya & Ambrogioni (2023):

$$\frac{\partial}{\partial t}p_t(0)\bigg|_{t=t_c} = p_{t_c}(0)\left[\underbrace{\nabla_x^2 u_{t_c}(0)}_{=0} + \nabla_x \ln p_{t_c}(0) \cdot \underbrace{\nabla_x u_{t_c}(0)}_{=0}\right] = 0 \tag{33}$$

The key insight is that the fixed point of the Fokker-Planck equation $(x_c, t_c) = (0, t_c)$ can be interpreted as spontaneous symmetry breaking in the potential function $u_t(x)$. This phenomenon not only elucidates the emergence of phase transitions but also highlights the role of symmetry breaking as a mechanism that governs such transitions in generative diffusion models.

Here, we discuss the universal properties of diffusion models that hold under more general potentials $u_t(x)$.

**Lemma G.6.** By the intermediate value theorem, there exists $x_c(t)$ $(t \in [0,1])$ such that

$$\int_{\mathbb{R}^d} \frac{\partial}{\partial t} p_t(x) dx = \int_{\mathbb{R}^d} p_t(x) \left[ \nabla_x^2 u_t(x) + \nabla_x \ln p_t(x) \cdot \nabla_x u_t(x) \right] dx \tag{34}$$

$$= \nabla_x^2 u_t(x_c(t)) + \nabla_x \ln p_t(x_c(t)) \cdot \nabla_x u_t(x_c(t)) \tag{35}$$

$$= 0 \tag{36}$$

where $\int_{\mathbb{R}^d} \frac{\partial}{\partial t} p_t(x) dx = \frac{\partial}{\partial t} \int_{\mathbb{R}^d} p_t(x) dx = \frac{\partial}{\partial t} 1 = 0$ and $(x, t) = (x_c(t), t)$.

**Lemma G.7.** By the definition of the Fokker-Planck equation 18:

$$\frac{\partial}{\partial t} p_t(x) = p_t(x) \frac{\partial}{\partial t} \ln p_t(x) \tag{37}$$

$$= p_t(x) \left[ \nabla_x^2 u_t(x) + \nabla_x \ln p_t(x) \cdot \nabla_x u_t(x) \right] \tag{38}$$

We get the following equation:

$$\frac{\partial}{\partial t} \ln p_t(x) = \nabla_x^2 u_t(x) + \nabla_x \ln p_t(x) \cdot \nabla_x u_t(x) \tag{39}$$

**Proposition G.8.** We introduce the Fisher information $I(t)$ of time:

$$I(t) := \int_{\mathbb{R}^d} p_t(x) \left[ \frac{\partial}{\partial t} \ln p_t(x) \right]^2 dx \tag{40}$$

We propose the criterion to identify the critical points $(x, t) = (x_h(t_h), t_h)$ at which phase transitions appear in the diffusion processes:

$$\left. \frac{\partial^2}{\partial t^2} \ln p_t(x_h(t)) \right|_{t=t_h} = 0 \tag{41}$$

In other words, the phase transition in diffusion models occurs at a critical point in space-time $(x_h(t_h), t_h)$, where the Fisher information degenerates.

*Proof.* By the intermediate value theorem, there exists $x_h(t)$ $(t \in [0,1])$ such that

$$I(t) = \int_{\mathbb{R}^d} p_t(x) \left[ \frac{\partial}{\partial t} \ln p_t(x) \right]^2 dx \tag{42}$$

$$= \left[ \frac{\partial}{\partial t} \ln p_t(x_h(t)) \right]^2 \tag{43}$$

According to the Cramér-Rao inequality for an arbitrary stochastic function $\theta_t(x)$, the Fisher information $I(t)$ has a positive lower bound Nicholson et al. (2020); Yoshimura & Ito (2021); Ito (2023):

$$\frac{\left| \frac{\partial}{\partial t} \langle \theta_t(x) \rangle \right|^2}{\operatorname{Var}\left[\theta_t(x)\right]} \leq I(t) \tag{44}$$

We define the extremum of the Fisher information as follows:

$$\left. \frac{\partial}{\partial t} I(t) \right|_{t=t_h} = 2 \left. \frac{\partial}{\partial t} \ln p_t(x_h(t)) \right|_{t=t_h} \left. \frac{\partial^2}{\partial t^2} \ln p_t(x_h(t)) \right|_{t=t_h} = 0 \tag{45}$$

The two conditions that yield the extremum of the Fisher information 45, $\left. \frac{\partial}{\partial t} \ln p_t(x_h(t)) \right|_{t=t_h} = 0$ and $\left. \frac{\partial^2}{\partial t^2} \ln p_t(x_h(t)) \right|_{t=t_h} = 0$, cannot hold simultaneously, as demonstrated in the following discussion.

$$\frac{\partial^2}{\partial t^2} \ln p_t(x_h(t)) = \frac{\frac{\partial^2}{\partial t^2} p_t(x_h(t))}{p_t(x_h(t))} - \left[ \frac{\partial}{\partial t} \ln p_t(x_h(t)) \right]^2 \tag{46}$$

If we assume $\left. \frac{\partial}{\partial t} \ln p_t(x_h(t)) \right|_{t=t_h} = 0$, it would result in $I(t) = 0$, which contradicts the Cramér-Rao inequality that generally imposes a positive lower bound 44. Therefore, the extremum of the Fisher information is determined by $\left. \frac{\partial^2}{\partial t^2} \ln p_t(x_h(t)) \right|_{t=t_h} = 0$. $\square$

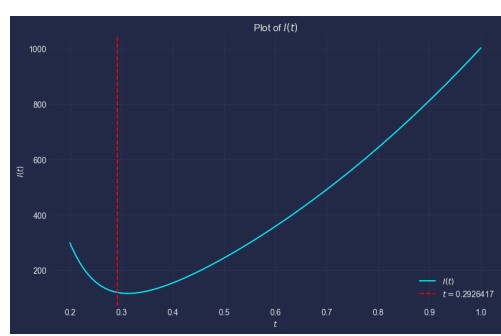
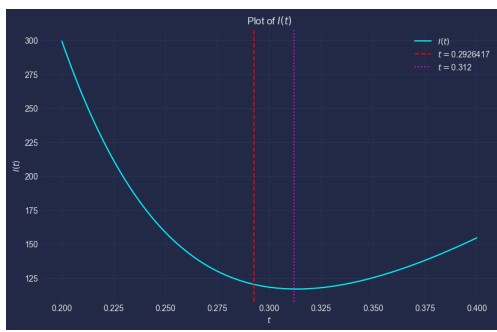

(a) The time dependence of the Fisher information $I(t)$ (Overview)

(b) The time dependence of the Fisher information $I(t)$ (Zoom in)

Figure 12: The time dependence of the Fisher information $I(t)$. The red dashed line represents the time $t_c = 0.293$ at which the potential breaks symmetry in Raya & Ambrogioni (2023). On the other hand, the magenta dotted line indicates the time $t_h = 0.312$ when the Fisher information predicted by our proposed method degenerates. While these two values are close, they do not match exactly.

## H THEORETICAL ANALYSIS ON THE EMPIRICAL RESULTS

### H.1 $\Gamma_{\mathcal{M}(\epsilon)}(t)$

Let $\epsilon > 0$. Let $\mathcal{M}(\epsilon)$ be the $\epsilon$-neighbourhood of a compact oriented manifold $\mathcal{M}$ in the Euclidean space $\mathbb{R}^d$ as defined in Definition 3.1. Suppose $p_t(x)$ is a smooth solution to the Fokker-Planck equation (3) with an initial condition $p_0(x) = \delta_{\mathcal{M}}(x)$ here $\delta_{\mathcal{M}}(x)$ is Dirac's density function with its support $\mathcal{M}$. We define a function $\Gamma_{\mathcal{M}(\epsilon)}(t)$ as follows:

$$\Gamma_{\mathcal{M}(\epsilon)}(t) := \int_{\mathcal{M}(\epsilon)} p_t(x)dx. \tag{47}$$

Roughly speaking, this is understood as a counting function of particles within the $\epsilon$-neighbourhood of $\mathcal{M}$.

**Proposition H.1.** *Assume $\beta(t) : \mathbb{R}_{\geq 0} \to \mathbb{R}$ is a smooth function and $f_t(x) = \frac{1}{2}\beta(t)f(x)$, $g_t(x) = \sqrt{\beta(t)}$ in (3) ( $f(x)$ is some smooth vector field). We have:*

$$\lim_{t \to 0} \frac{\partial}{\partial t}\Gamma_{\mathcal{M}(\epsilon)}(t) = 0$$

*and*

$$\lim_{t \to \infty} \frac{\partial}{\partial t}\Gamma_{\mathcal{M}(\epsilon)}(t) = 0.$$

*Thus there exists at least one $t_c$ in $(0, +\infty)$ such that $\frac{\partial^2}{\partial t^2}\Gamma_{\mathcal{M}(\epsilon)}(t_c) = 0$. Moreover if $\beta(t) > 0$ and*

$$\nabla_x p_t(x) \cdot \boldsymbol{n} - p_t(x)f(x) \cdot \boldsymbol{n} < 0 \tag{48}$$

*for any $x \in \partial\mathcal{M}(\epsilon)$ and any $t \in \mathbb{R}_{>0}$ then $\Gamma_{\mathcal{M}(\epsilon)}(t)$ is strictly monotonically decreasing. Here $\boldsymbol{n}$ a unit outward pointing normal vector field along $\partial M(\epsilon)$. In particular we can express the derivative of the function $\Gamma_{\mathcal{M}(\epsilon)}(t)$ in terms of the free energy $u$ defined in (20):*

$$\frac{\partial}{\partial t}\Gamma_{\mathcal{M}(\epsilon)}(t) = \int_{\partial M(\epsilon)} p_t(x)\nabla_x u_t(x) \cdot \boldsymbol{n}dx.$$

*Proof.* (informal) We may compute for $t > 0$:

$$\frac{\partial}{\partial t}\Gamma_{\mathcal{M}(\epsilon)}(t) = \int_{\mathcal{M}(\epsilon)} \frac{\partial}{\partial t}p_t(x)dx$$

$$= \beta(t)\int_{\mathcal{M}(\epsilon)} \left(\nabla_x \cdot p_t(x)f(x) + \Delta_x p_t(x)\right)dx$$

$$= \beta(t)\left(-\int_{\partial\mathcal{M}(\epsilon)} p_t(x)f(x)\cdot\boldsymbol{n}ds + \int_{\partial\mathcal{M}(\epsilon)} \nabla_x p_t(x)\cdot\boldsymbol{n}ds\right) \quad (49)$$

$$\xrightarrow{t\to 0} 0.$$

The second equality follows since $p_t(x)$ satisfies the Fokker-Planck equation (3). The third equality follows from the divergence theorem where $\boldsymbol{n}$ is the unit outward pointing normal vector field along $\partial\mathcal{M}(\epsilon)$. The last limit follows since $\lim_{t\to 0} p_t(x) = \delta_{\mathcal{M}}(x)$ and in particular $\lim_{t\to 0} p_t(x) = 0$ and $\lim_{t\to 0} \nabla_x p_t(x) = 0$ in $\partial\mathcal{M}(\epsilon)$. To be more precise the convergence of the limit we could make use of the following chain of inequalities:

$$\left|\beta(t)\left(-\int_{\partial\mathcal{M}(\epsilon)} p_t(x)f(x)\cdot nds + \int_{\partial\mathcal{M}(\epsilon)} \nabla_x p_t(x)\cdot\boldsymbol{n}ds\right)\right|$$

$$\leq |\beta(t)|\left(\max_{x\in\partial\mathcal{M}(\epsilon)}\{|f(x)|\}\int_{\partial\mathcal{M}(\epsilon)} |p_t(x)|ds + \int_{\partial\mathcal{M}(\epsilon)} |\nabla_x p_t(x)\cdot\boldsymbol{n}|ds\right)$$

$$\leq |\beta(t)|\left(\max_{x\in\partial\mathcal{M}(\epsilon)}\{|f(x)|\}\sup_{x\in\partial\mathcal{M}(\epsilon)}|p_t(x)|\int_{\partial\mathcal{M}(\epsilon)} 1ds + \sup_{x\in\partial\mathcal{M}(\epsilon)}|\nabla_x p_t(x)|\int_{\partial\mathcal{M}(\epsilon)} 1ds\right)$$

$$= |\beta(t)|\int_{\partial\mathcal{M}(\epsilon)} 1ds\left(\max_{x\in\partial\mathcal{M}(\epsilon)}\{|f(x)|\}\sup_{x\in\partial\mathcal{M}(\epsilon)}|p_t(x)| + \sup_{x\in\partial\mathcal{M}(\epsilon)}|\nabla_x p_t(x)|\right).$$

When $t \to \infty$, $p_t(x_t)$ tends to be stationary i.e., $\lim_{t\to\infty}\frac{\partial}{\partial t}p_t(x_t) = 0$. Therefore

$$\lim_{t\to\infty}\frac{\partial}{\partial t}\Gamma_{\mathcal{M}(\epsilon)}(t) = 0.$$

The existence follows from the mean value theorem. Finally let us show it is strictly monotonically decreasing. The negativity of $\frac{\partial}{\partial t}\Gamma_{\mathcal{M}(\epsilon)}(t)$ follows from (49) and (48). □

## I   THE TIME WHEN THE SCORE VECTOR FIELD REVERSES

In this section we discuss the details of Section 4.

### I.1   VARIANCE PRESERVING SDE AND SCORE VECTOR FIELD

We consider the widely used Variance Preserving (VP-SDE) (DDPM):

$$d\mathbf{Y}_s = -\frac{1}{2}\beta(s)\mathbf{Y}_s ds + \sqrt{\beta(s)}d\widehat{\mathbf{W}}_s \quad (50)$$

with corresponding generative dynamics:

$$d\mathbf{X}_t = \left[\beta(T-t)\nabla_x \log p(\mathbf{X}_t, T-t) + \frac{1}{2}\beta(T-t)\mathbf{X}_t\right]dt + \sqrt{\beta(T-t)}d\mathbf{W}_t. \quad (51)$$

One expresses marginal distribution $p_s(x)$ of Variance Preserving (VP-SDE) (DDPM) as follows:

$$p_s(x) = \int_M N(y|\theta_s x, (1-\theta_s^2)I)p_0(y)dy, \quad (52)$$

where $\theta_s = e^{-\frac{1}{2}\int_0^s \beta(\tau)d\tau}$ and $p_0(y)$ is the distribution at time at 0. The score at point $x$ is given by

$$\nabla_x \ln p_s(x) = \frac{\theta_s}{(1-\theta_s^2)p_s(x)}\int_M (y-\theta_s x)N(y|\theta_s x, (1-\theta_s^2)I)p_0(y)dy. \quad (53)$$

## I.2 ANALYSIS OF THE BEHAVIOUR OF THE SCORE VECTOR FIELD AT THE BOUNDARY OF THE TUBULAR NEIGHBOURHOOD OF A HYPERSPHERE

**Proposition I.1.** *Suppose $\mathcal{M} = S^n$ is a $n$-sphere of radius $R$ in $\mathbb{R}^d$. We predict the following observation: Let $\epsilon$ be as $R > \epsilon > 0$. Let $\boldsymbol{n}$ be a unit outward pointing normal vector to the boundary of $\epsilon$-neighbourhood $\partial \mathcal{M}(\epsilon)$. Assume*

$$\frac{\epsilon + (1 - \theta_s)(R - \epsilon)}{\sqrt{1 - \theta_s^2}} \geq \sqrt{d}, \tag{54}$$

*$x \in \partial \mathcal{M}(\epsilon)$ and $p_0(y)$ is constant $C$ greater than 0 on $\mathcal{M}$. Then:*

$$\nabla_x \ln p_t(x) \cdot \boldsymbol{n} \leq 0.$$

*Proof.* (this proof is yet informal. Although we only perform this proof for the case $d = 2$ and $\mathcal{M}$ is a 1-sphere of radius 1, we hope it can be done in general dimensions). Since $\nabla_x \ln p_t(x) = \frac{\nabla_x p_t(x)}{p_t(x)}$, it is enough to prove $\nabla_x p_t(x) \cdot \boldsymbol{n} \leq 0$. Performing a change of variables $w = \frac{y - \theta_s x}{\sqrt{1 - \theta_s^2}}$ we have:

$$
\begin{aligned}
\nabla_x p_t(x) \cdot \boldsymbol{n} &= C \frac{\theta_s}{(1 - \theta_s^2)} \int_{\mathcal{M}} (y - \theta_s x) N(y; \theta_s x, (1 - \theta_s^2)I) dy \cdot \boldsymbol{n} \\
&= C \int_{\frac{\mathcal{M} - \theta_s x}{\sqrt{1 - \theta_s^2}}} w N(w; 0, I) dw \cdot \boldsymbol{n} \\
&= C \int_{\frac{\mathcal{M} - \theta_s x}{\sqrt{1 - \theta_s^2}}} \frac{w}{|w|} \cdot \boldsymbol{n} |w| N(w; 0, I) dw \\
&= C \int_{N_-} \frac{w}{|w|} \cdot \boldsymbol{n} |w| N(w; 0, I) dw + \int_{N_+} \frac{w}{|w|} \cdot \boldsymbol{n} |w| N(w; 0, I) dw \\
&= C \int_{\mathbb{R}^2} \frac{z}{|z|} \cdot \boldsymbol{n} |z| N(z; 0, I) \delta_{N_-}(z) dz \\
&\qquad\qquad + C \int_{\mathbb{R}^2} \frac{z}{|z|} \cdot \boldsymbol{n} |z| N(z; 0, I) \delta_{N_+}(z) dz, \quad (!)
\end{aligned}
$$

where $\frac{\mathcal{M} - \theta_s x}{\sqrt{1 - \theta_s^2}}$ is the image of the manifold $\mathcal{M}$ by a diffeomorphism $y \mapsto \frac{y - \theta_s x}{\sqrt{1 - \theta_s^2}}$ and $N_-$ (resp. $N_+$) is $\{w \in \frac{\mathcal{M} - \theta_s x}{\sqrt{1 - \theta_s^2}}; w \cdot \boldsymbol{n} < 0 (\text{resp.} > 0)\}$. $dz$ is a volume form of $\mathbb{R}^d$. Let $\theta$ be the angle between $z/|z|$ and $\boldsymbol{n}$. If we use the polar coordinates $(|z_\theta|, \theta) \in (0, \infty] \times [0, 2\pi)$, since $\cos(\theta + \pi) = -\cos(\theta)$, (put $N_z(\theta) := \{(|z|, \theta) \in (0, \infty] \times [0, 2\pi); z \in N \text{ for some } \theta \text{ s.t. } \cos \theta = \frac{z}{|z|} \cdot \boldsymbol{n})\}$) we may estimate (!) as follows:

$$(!) = \int_{\pi/2}^{-\pi/2} \cos \theta \left( \int_0^\infty |z_\theta|^2 N(z_\theta : 0, I) \delta_{N_z(\theta)}(|z_\theta|) d|z| \right) d\theta \tag{55}$$

$$+ \int_{-\pi/2}^{\pi/2} \cos \theta \left( \int_0^\infty |z_\theta|^2 N(z_\theta : 0, I) \delta_{N_z(\theta)}(|z_\theta|) d|z| \right) d\theta$$

$$= \int_{-\pi/2}^{\pi/2} \cos \theta \left( |z_{\theta_+}|^2 N(z_{\theta_+} : 0, I) - |z_{\theta_-}|^2 N(z_{\theta_-} : 0, I) \right) d\theta, \tag{$\int$}$$

where we set $z_{\theta_+} \in N^+$, $z_{\theta_-} \in N^-$ and $z_{\theta_+} = -c_\theta z_{\theta_-}$ for some $c_\theta > 0$. This integral ($\int$) is negative or zero if

$$\left( |z_{\theta_+}|^2 N(z_{\theta_+} : 0, I) - |z_{\theta_-}|^2 N(z_{\theta_-} : 0, I) \right) \leq 0 \tag{56}$$

for any $\theta$. Since $x \in \partial \mathcal{M}(\epsilon)$, $|z_{\theta_+}| \geq |z_{\theta_-}| \geq \frac{\epsilon + (1 - \theta_s)(R - \epsilon)}{\sqrt{1 - \theta_s^2}}$ holds. Since $|z|^2 N(z : 0, I)$ is strictly monotonically decreasing if $|z| \geq \sqrt{2}$, the inequality holds for $|z_{\theta_+}| \geq |z_{\theta_-}| \geq \sqrt{2}$. Thus when $\frac{\epsilon + (1 - \theta_s)(R - \epsilon)}{\sqrt{1 - \theta_s^2}} \geq \sqrt{2}$ the assertion follows. $\qquad\square$

**Example I.2.** Let $\mathcal{M} = S^1$ in $\mathbb{R}^2$ and $|x| = 0.99$. Compute (54) and we understand that $\nabla_x \ln p_t(x)$ points toward $S^1$ if $\theta_s > 0.712$. Similar thing can be observed for $S^2$ in $\mathbb{R}^3$. Therefore Proposition I.1 explain the Figure 21 and Figure 22.

**Remark I.3.** Conjecture 4.4 is a kind of generalisation of Theorem D.1 in Stanczuk et al. (2024). The authors predict we can formulate more general conjecture by using concept of injectivity radii for more general manifolds to illustrate and explain the behaviour of the score vector field of more general diffusion models.

**Proposition I.4.** *Suppose $\mathcal{M}$ is a compact oriented manifold embedded in $\mathbb{R}^d$. We predict the following observation: Let $\epsilon$ be an injectivity radius. Let $\boldsymbol{n}$ be a unit outward pointing normal vector to the boundary of $\epsilon$-neighbourhood $\partial \mathcal{M}(\epsilon)$. Assume*

$$\frac{\epsilon + |x|(1 - \theta_s)}{\sqrt{1 - \theta_s^2}} \geq \sqrt{d}, \tag{57}$$

$x \in \partial \mathcal{M}(\epsilon)$ and $p_0(y)$ is constant $C$ greater than $0$ on $\mathcal{M}$. Finally assume a line segment with $x$ and the origin as its vertices does not intersect $\mathcal{M}$. Assume moreover the following condition.

(i) *For any $y \in \mathcal{M}$ with $(y - \theta_s x) \cdot \boldsymbol{n} > 0$ there exists $y' \in \mathcal{M}$ and some $c > 0$ such that $-c(y - \theta_s x) = (y' - \theta_s x)$.*

(ii) *Assume that for each $y \in \mathcal{M}$ such that $(y - \theta_s x) \cdot \boldsymbol{n} < 0$, there exists $\tilde{y} \in \mathcal{M}$ and $c > 0$ such that $-c(\tilde{y} - \theta_s x) = (y - \theta_s x)$. Then $c \leq 1$.*

(iii) *For any $y \in \mathcal{M}$, $\{c(y - \theta_s x) | c > 0\} \cap \mathcal{M}$ is a finite set.*

*Then:*

$$\nabla_x \ln p_t(x) \cdot \boldsymbol{n} \leq 0.$$

*Proof.* (this proof is yet informal. Although we only perform this proof for the case $d = 2$ and $\mathcal{M}$ is a curve, we hope it can be done in general dimensions). Since $\nabla_x \ln p_t(x) = \frac{\nabla_x p_t(x)}{p_t(x)}$, it is enough to prove $\nabla_x p_t(x) \cdot \boldsymbol{n} \leq 0$. Performing a change of variables $w = \frac{y - \theta_s x}{\sqrt{1 - \theta_s^2}}$ we have:

$$\nabla_x p_t(x) \cdot \boldsymbol{n} = C \frac{\theta_s}{(1 - \theta_s^2)} \int_{\mathcal{M}} (y - \theta_s x) N(y; \theta_s x, (1 - \theta_s^2) I) dy \cdot \boldsymbol{n}$$

$$= C \int_{\frac{\mathcal{M} - \theta_s x}{\sqrt{1 - \theta_s^2}}} w N(w; 0, I) dw \cdot \boldsymbol{n}$$

$$= C \int_{\frac{\mathcal{M} - \theta_s x}{\sqrt{1 - \theta_s^2}}} \frac{w}{|w|} \cdot \boldsymbol{n} |w| N(w; 0, I) dw$$

$$= C \int_{N_-} \frac{w}{|w|} \cdot \boldsymbol{n} |w| N(w; 0, I) dw + \int_{N_+} \frac{w}{|w|} \cdot \boldsymbol{n} |w| N(w; 0, I) dw$$

$$= C \int_{\mathbb{R}^2} \frac{z}{|z|} \cdot \boldsymbol{n} |z| N(z; 0, I) \delta_{N_-}(z) dz$$

$$+ C \int_{\mathbb{R}^2} \frac{z}{|z|} \cdot \boldsymbol{n} |z| N(z; 0, I) \delta_{N_+}(z) dz, \quad (!)$$

where $\frac{\mathcal{M} - \theta_s x}{\sqrt{1 - \theta_s^2}}$ is the image of the manifold $\mathcal{M}$ by a diffeomorphism $y \mapsto \frac{y - \theta_s x}{\sqrt{1 - \theta_s^2}}$ and $N_-$ (resp. $N_+$) is $\{w \in \frac{\mathcal{M} - \theta_s x}{\sqrt{1 - \theta_s^2}}; w \cdot \boldsymbol{n} < 0(\text{resp.} > 0)\}$. $dz$ is a volume form of $\mathbb{R}^d$. Let $\theta$ be the angle between $z/|z|$ and $\boldsymbol{n}$. If we use the polar coordinates $(|z_\theta|, \theta) \in (0, \infty) \times [0, 2\pi)$, since $\cos(\theta + \pi) = -\cos(\theta)$, (put $N_z(\theta) := \{(|z|, \theta) \in (0, \infty] \times [0, 2\pi); z \in N$ for some $\theta$ s.t. $\cos \theta = \frac{z}{|z|} \cdot \boldsymbol{n})\}$) we may estimate (!)

as follows:

$$(!) = \int_{\pi/2}^{-\pi/2} \cos\theta \left( \int_0^\infty |z_\theta|^2 N(z_\theta : 0, I) \delta_{N_z(\theta)}(|z_\theta|) d|z| \right) d\theta + \int_{-\pi/2}^{\pi/2} \cos\theta \left( \int_0^\infty |z_\theta|^2 N(z_\theta : 0, I) \delta_{N_z(\theta)}(|z_\theta|) d|z| \right) d\theta$$

$$= \int_{-\pi/2}^{\pi/2} \cos\theta \left( \sum |z_{\theta_+}|^2 N(z_{\theta_+} : 0, I) - \sum |z_{\theta_-}|^2 N(z_{\theta_-} : 0, I) \right) d\theta$$

$$\leq C' \int_{-\pi/2}^{\pi/2} \cos\theta \left( |z_{\theta_+}|^2 N(z_{\theta_+} : 0, I) - |z_{\theta_-}|^2 N(z_{\theta_-} : 0, I) \right) d\theta, \tag{$\int$}$$

where $z_{\theta_+} \in N^+$, $z_{\theta_-} \in N^-$ and $z_{\theta_+} = -c_\theta z_{\theta_-}$ for some $c_\theta > 0$. If there is no such $z_{\theta_+}$, we set $z_{\theta_+} = 0$. Also we set $|z_{\theta_+}|N(z_{\theta_+} : 0, I) := \max\{|z_{\theta_+}|^2 N(z_{\theta_+} : 0, I)\}$ and $|z_{\theta_-}|N(z_{\theta_-} : 0, I) := \min\{|z_{\theta_-}|^2 N(z_{\theta_-} : 0, I)\}$. Thus by the assumption (ii) we may obtain ($\int$). This integral ($\int$) is negative or zero if

$$\left( |z_{\theta_+}|^2 N(z_{\theta_+} : 0, I) - |z_{\theta_-}|^2 N(z_{\theta_-} : 0, I) \right) \leq 0 \tag{58}$$

for any $\theta$. Since $x \in \partial\mathcal{M}(\epsilon)$ and by the assumption (ii), by the lemma below $|z_{\theta_+}| \geq |z_{\theta_-}| \geq \frac{\epsilon + |x|(1 - \theta_s)}{\sqrt{1 - \theta_s^2}}$ holds. Since $|z|^2 N(z : 0, I)$ is strictly monotonically decreasing if $|z| \geq \sqrt{2}$, the inequality holds for $|z_{\theta_+}| \geq |z_{\theta_-}| \geq \sqrt{2}$. Thus when $\frac{\epsilon + |x|(1 - \theta_s)}{\sqrt{1 - \theta_s^2}} \geq \sqrt{2}$ the result follows. $\square$

**Lemma I.5.** In the situation of the proof above, we have the estimate:

$$|z_{\theta_+}| \geq |z_{\theta_-}| \geq \frac{\epsilon + |x|(1 - \theta_s)}{\sqrt{1 - \theta_s^2}}.$$

*Proof.* The assumeption (ii), $|z_{\theta_+}| \geq |z_{\theta_-}|$ is evident. For any $\tilde{y}$ such that $|\tilde{y} - x| = \epsilon$ we have

$$|y - \theta_s x| \geq |\tilde{y} - \theta_s x|. \tag{59}$$

Since $\max_{\tilde{y} \in \{\tilde{y} | \epsilon = |\tilde{y} - x|\}} |\tilde{y} - \theta_s x| = \epsilon + (1 - \theta_s)|x|$, we have the result. $\square$

**Remark I.6.** The smaller the injectivity radius slower time of the turning of the score vector field becomes.

## J EXPERIMENTAL DETAILS

### J.1 EXPERIMENTS DETAIL FOR DDPM

In previous studies (Raya & Ambrogioni (2023)), the training of diffusion models was performed using DDPM. The number of timesteps is set to 1000, and the noise schedule coefficient $\beta$ linearly increases from $1.0 \times 10^{-4}$ to $2.0 \times 10^{-2}$. A key difference from prior work is that, for denoising, the MLP layers have been replaced with a 1D U-Net. This adjustment is necessary to handle higher-dimensional data, such as 16D, 24D, and 48D, where a more complex model is required.

The model is trained using the mean squared error (MSE) loss function, with AdamW as the optimizer. The learning rate is set to $1 \times 10^{-3}$, and the batch size is 32. For toy data experiments, the training dataset consists of 50,000 points sampled from a uniform distribution. The model is trained without using any advanced samplers like DDIM, relying solely on the standard DDPM reverse process.

### J.2 EXPERIMENTS DETAIL FOR THE ANALYSIS OF TUBULAR NEIGHBOURHOOD IN CIRCLE

In Section 5.1, we conducted experiments using a uniform distribution on the unit circle embedded in a higher-dimensional Euclidean space. The red line plot shows the proportion of particles outside the tubular neighbourhood at each timestep when generation is performed over 1000 timesteps, the same as during training. Here, the injectivity radius that defines the tubular neighbourhood is set to 1. Therefore, being outside the tubular neighbourhood means that a particle's distance from the unit circle exceeds 1. In the generation process, a point is first sampled from Gaussian noise. When the ambient space is sufficiently large, the proportion of particles outside the tubular neighbourhood is 1. As the timesteps progress during the generation process, each data point approaches the data manifold, which in this case is the unit circle. Thus, at the final timestep, all particles are expected to lie within the tubular neighbourhood.

The blue line plot evaluates the accuracy of data generation using the Wasserstein distance when initialisation is delayed during the generation process of the diffusion model. The horizontal axis, Diffusion Time, indicates the number of timesteps performed out of the usual 1000-step generation process. For example, in the case of 800 steps, the initialisation is delayed by 200 steps, with the generation beginning from Gaussian noise and proceeding for the remaining 800 timesteps. Following previous studies, we refer to this as late initialisation.

We can calculate the calculation of the injectivity radius as 1 (see. 3.3).

### J.3 EXPERIMENTS DETAIL FOR THE ANALYSIS OF TUBULAR NEIGHBOURHOOD IN HYPERSPHERE

In Section 5.1, experiments were conducted using a uniform distribution on a unit hypersphere embedded in a higher-dimensional Euclidean space. The experimental setup is the same as in J.2. Given that the injectivity radius of the unit hypersphere is 1(see E.3), being outside the tubular neighbourhood implies that the distance from the unit sphere is greater than or equal to 1.

In Fig. 5 (see Section 5.2), we discussed that the discrepancy between the proportion of particles outside the tubular neighbourhood and the rise in Wasserstein distance can be attributed to the increasing distance between the distributions. To support this hypothesis, we conducted an experiment where we initialized the Gaussian noise using the lateinit scheme, with $\mathbf{x_t} \sim \mathcal{N}(\mathbf{0}, I/\sqrt{d})$, where $d$ is the dimension of the ambient space. Corresponding to the experiment shown in Fig. 5, we performed another experiment on $S^{20}$ with an ambient space of $\mathbb{R}^{48}$. As shown in Fig. 13, we observed that the Wasserstein distance starts increasing as particles begin entering the tubular neighbourhood.

### J.4 EXPERIMENTS DETAIL FOR THE ANALYSIS OF TUBULAR NEIGHBOURHOOD IN ELLIPSE, TORUS

In Section 5.2, experiments were conducted using uniform distributions on an ellipse and a torus, both embedded in a higher-dimensional Euclidean space. The experimental setup is consistent with that described in J.2.

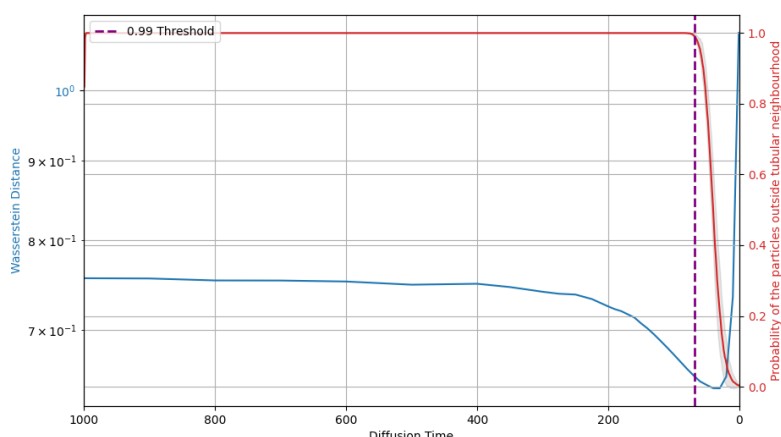

Figure 13: $S^{20}$ embedded in $\mathbb{R}^{48}$

For the ellipse, given the semi-major axis $2a$ and the semi-minor axis $2b$, the injectivity radius is calculated as $\frac{b^2}{a}$. In this experiment, we tested two cases: one with a semi-major axis of 4 and a semi-minor axis of 2, and another with a semi-major axis of 6 and a semi-minor axis of 2. The injectivity radii for these cases are $\frac{1}{2}$ and $\frac{1}{3}$, respectively. We can calculate the injectivity radius of ellipse as follow. Let us verify Theorem 3.7 through ellipse, given the semi-major axis $a$ and the semi-minor axis $b$. Define a function $F \colon \mathbb{R}^2 \to \mathbb{R}$ by

$$F(x, y) = \frac{x^2}{a^2} + \frac{y^2}{b^2} - 1.$$

Then we have $M = F^{-1}(0)$. One of the normal vector field on $M$ is given as $\mathrm{grad}(F) = (\frac{\partial F}{\partial x}, \frac{\partial F}{\partial y}) = (\frac{2x}{a^2}, \frac{2y}{b^2})$, so $(\frac{y}{b^2}, -\frac{x}{a^2})$ is a tangent vector field which spans the tangent space to $M$ at each point $(x, y) \in M$.

Applying Theorem 3.7, the first injectivity radius $R_1(M)$ is calculated as follows. For a point $(x, y) \in M$, the matrix

$$L_M((x, y), (v_1, v_2)) = \begin{bmatrix} \frac{2x}{a^2} & -\frac{v_2}{a^2} - \frac{y}{b^2} \\ \frac{2y}{b^2} & \frac{v_1}{b^2} + \frac{x}{a^2} \end{bmatrix}$$

is degenerate (i.e., its determinant is zero) if and only if $(v_1, v_2) = (-\frac{b^4 x^2 + a^4 y^2}{a^2 b^2 x + \frac{a^4 y^2}{x}}, -\frac{b^4 x^2 + a^4 y^2}{b^4 \frac{x^2}{y} + a^2 b^2 y})$. Let $(x, y) = (a\cos\theta, b\sin\theta)$, then, $(v_1, v_2) = (-\frac{b^2 \cos^2\theta + a^2 \sin^2\theta}{a\cos\theta + a\frac{\sin^2\theta}{\cos\theta}}, -\frac{b^2 \cos^2\theta + a^2 \sin^2\theta}{\frac{b\cos^2\theta}{\sin\theta} + b\sin\theta})$. The $L^2$ norm of $(v_1, v_2)$ is minimized at $\theta = 0$ when $a > b$, and in this case, $R_1(M) = \frac{b^2}{a}$.

For the torus, the injectivity radius is given by $\min(r' - r, r)$, where $r'$ is the major radius and $r$ is the minor radius (see E.1). In this experiment, we used two cases: one with a major radius of 2 and a minor radius of 1, and another with a major radius of 3 and a minor radius of 1. In both cases, the injectivity radius is 1.

These calculations provide the injectivity radii used in our experiments on both the ellipse and the torus, guiding the analysis of the tubular neighborhoods in these geometric settings.

In the experiments presented in Section 5.2, we included figures for an ellipse with a major axis of 6 and a minor axis of 2, as well as a torus with a major radius of 3 and a minor radius of 1. Here, we provide additional figures for an ellipse with a major axis of 4 and a minor axis of 2, and a torus with a major radius of 2 and a minor radius of 1.

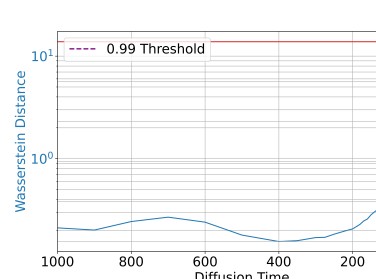

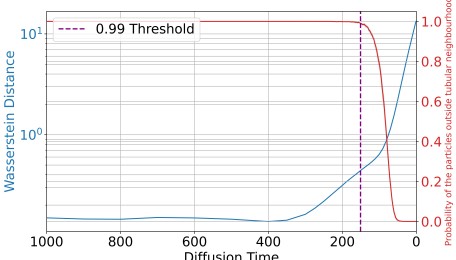

Figure 14: ellipse $R = 2, r = 1$ embedded in $\mathbb{R}^{16}$

Figure 15: torus $R = 2, r = 1$ embedded in $\mathbb{R}^{16}$

| Dataset \ $\rho_{proportion}$ | 0.1 | 0.5 | 0.9 | 0.95 | 0.99 | 0.999 | 1.0 |
|---|---|---|---|---|---|---|---|
| Ellipse $(R = 2, r = 1)$ embedded in $\mathbb{R}^{16}$ | 3.690 | 2.351 | 1.110 | 0.926 | 0.593 | 0.441 | 0.211 |
| Torus $(R = 2, r = 1)$ embedded in $\mathbb{R}^{16}$ | 1.328 | 0.816 | 0.563 | 0.520 | 0.440 | 0.333 | 0.149 |

### J.5 EXPERIMENTS DETAIL FOR THE ANALYSIS OF TUBULAR NEIGHBOURHOOD IN DISJOINTARCS CASES

In Section 5.3, we conducted experiments using a data distribution composed of segments from two circles with different curvatures, both embedded in a higher-dimensional Euclidean space. The dataset was constructed by uniformly sampling 50,000 points from two regions: one segment from a circle with radius 1, covering the angle range from $\pi/6$ to $\pi/3$, and another segment from a circle with radius 2, covering the angle range from $7\pi/6$ to $4\pi/3$.

Next, we consider appropriate values for the injectivity radius. For the submanifold A, the injectivity radius is considered to be 1 (although, strictly speaking, the injectivity radius is undefined at the endpoints where the tangent plane cannot be properly defined, we exclude these points for our analysis). On the other hand, for the submanifold B, since it is a part of a circle with radius 2, the injectivity radius is considered to be 2. Therefore, the injectivity radius for the combined manifold formed by these two segments is determined to be 1.

#### J.5.1 ADDITIONAL EXPERIMENTS

To further investigate the behaviour of the score vector field under different curvature settings, we conducted additional experiments using new datasets. These datasets include:

1. A segment from a circle with radius 3, covering the angle range from $\pi/6$ to $\pi/3$, and a segment from a circle with radius 1, covering the angle range from $7\pi/6$ to $4\pi/3$. See Figure 16 for the experimental result. To further analyse the behaviour of the proportion of particles outside the tubular neighbourhood (depicted as the red solid line in the experimental results), we conducted additional experiments to investigate how this behaviour changes with different values of the neighbourhood radius. Specifically, we considered a neighbourhood radius of $R = 3$ (which is different from the injectivity radius $r = 1$). The proportion of particles outside this larger neighbourhood region is plotted as a red dashed line in Figure 18. This result demonstrates that the behaviour of the red line varies depending on the chosen value for the neighbourhood radius.

2. A segment from a circle with radius $3/2$, covering the angle range from $\pi/6$ to $\pi/3$, and a segment from a circle with radius $1/2$, covering the angle range from $7\pi/6$ to $4\pi/3$. See Figure 17 for the experimental result.

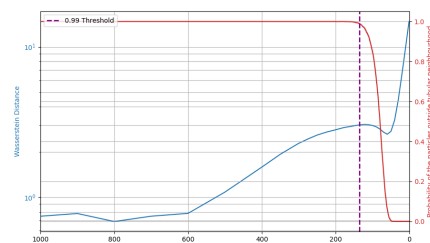

Figure 16: Disjoint arcs case $R = 3, r = 1$ embedded in $\mathbb{R}^{16}$

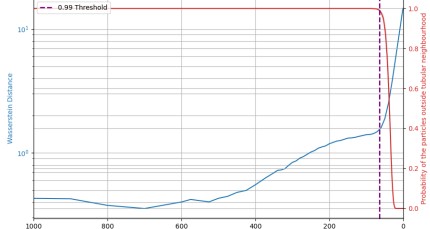

Figure 17: Disjoint arc case $R = 3/2, r = 1/2$ embedded in $\mathbb{R}^{16}$

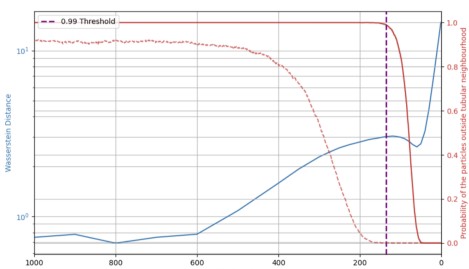

Figure 18: Disjoint arc case $R = 3, r = 1$ embedded in $\mathbb{R}^{16}$

### J.6 EXPERIMENTS DETAIL FOR THE ANALYSIS OF TUBULAR NEIGHBOURHOOD IN NARURAL DATASET

In Section 5.4, we performed experiments by embedding real-world datasets such as MNIST and Fashion MNIST onto a hypersphere, subsequently mapping them into a high-dimensional Euclidean space. For efficient generation and sampling in diffusion models, it is a common practice to reduce the dimensionality into a latent space—similar to the approach in Latent Diffusion—and consider the transitions of the diffusion model within this space. While Latent Diffusion utilizes VQ-VAE for embedding, we employ a Hyperspherical VAE in our methodology.

In our approach, several key modifications were made to the original hyperspherical VAE (sVAE) (Davidson et al., 2018) setup used in prior studies. One significant change was transitioning from binary data representation, where data was handled as binary values, to continuous data representation. As a result, we replaced the Binary Cross-Entropy (BCE) loss function with the Mean Squared Error (MSE) loss function. This modification allows for more accurate modeling and reconstruction of continuous data, particularly when working with datasets such as MNIST and Fashion MNIST.

Furthermore, to address the limitations of the previously used MLP layers for reconstructing natural images, we enhanced the model by adopting CNN-based layers. This adjustment is particularly beneficial for reconstructing images from higher-dimensional latent spaces.

To demonstrate the improvements, we present a comparison between the normal VAE (nVAE) and the hyperspherical VAE (sVAE), focusing on the ELBO (Evidence Lower Bound) and log-likelihood (LL) values for different latent space dimensions (10, 15, and 20). Additionally, we compare the performance of both nVAE and sVAE when using MLP-based and CNN-based architectures. The results are summarized in the following tables:

In Section 5.4, we explored the effectiveness of late initialisation in accelerating image generation by embedding natural images onto a hypersphere and sampling on the hypersphere. It is crucial for practical applications to ensure that points sampled through late initialisation in the latent space can generate realistic images when passed through the decoder. Here, we present generated images

| ELBO (↓) / LL (↓) | nVAE | sVAE |
|---|---|---|
| dim 10 | -23.7 / -22.9 | -25.5 / -24.0 |
| dim 15 | -23.8 / -23.1 | -26.7 / -24.5 |
| dim 20 | -23.8 / -23.1 | -27.6 / -25.0 |

Table 6: Comparison of ELBO and LL for nVAE and sVAE with different latent space dimensions, using MLP-based models on the Fashion MNIST dataset.

| ELBO (↓) / LL (↓) | nVAE (CNN) | sVAE (CNN) |
|---|---|---|
| dim 10 | -22.8 / -22.1 | -24.4 / -22.7 |
| dim 20 | -23.1 / -22.2 | -26.8 / -23.9 |

Table 7: Comparison of ELBO and LL for nVAE and sVAE with different latent space dimensions, using CNN-based models on the Fashion MNIST dataset.

obtained by passing points sampled using late initialisation during the generation process of the diffusion model through the decoder of the hyperspherical VAE, and qualitatively evaluate the results.

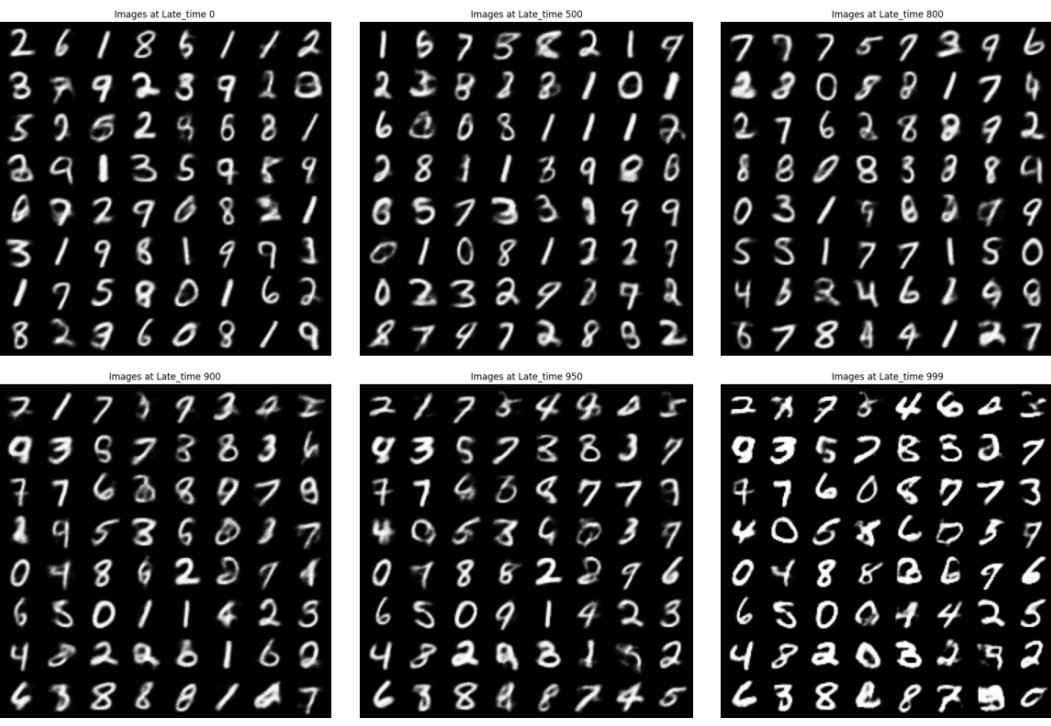

Figure 19: MNIST images decoded using SVAE after diffusion times of 1000, 500, 200, 100, 50, and 1 (arranged from left to right, top to bottom) in the latent space.

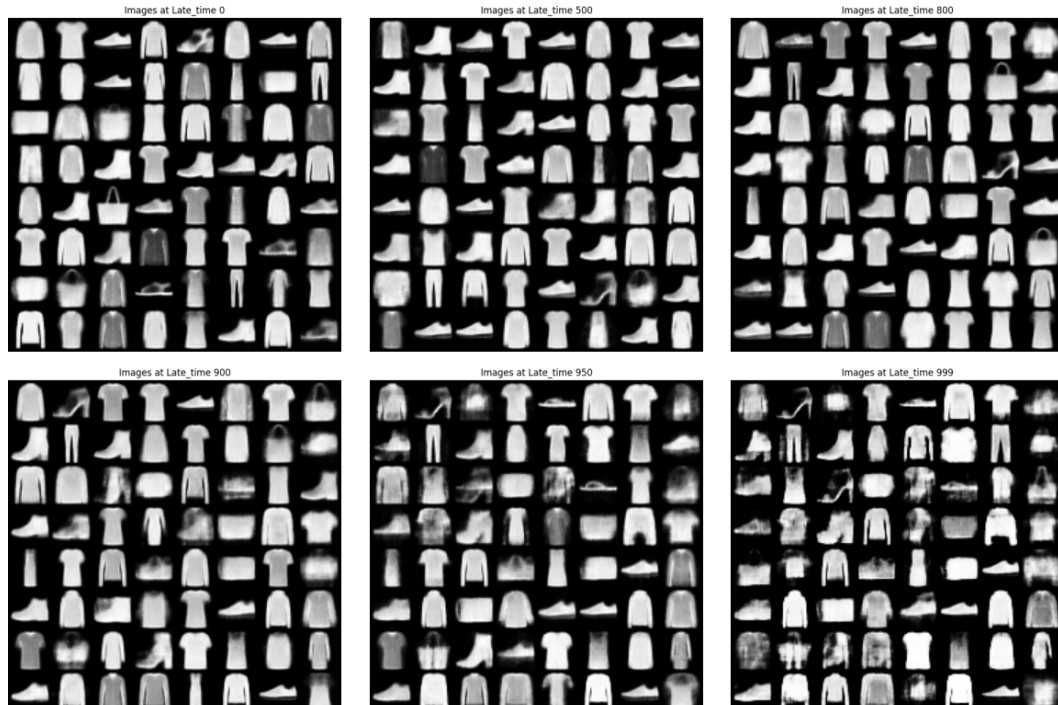

Figure 20: Fashion MNIST images decoded using SVAE after diffusion times of 1000, 500, 200, 100, 50, and 1 (arranged from left to right, top to bottom) in the latent space.

There are limitations regarding embedding onto hyperspheres. Previous studies have shown that as the dimensionality of the embedding onto the hypersphere increases beyond a certain value, the accuracy of the embedding decreases. This phenomenon is related to the fact that the surface area of a hypersphere approaches zero in the high-dimensional limit. Therefore, although we conducted experiments with MNIST and Fashion MNIST, for larger datasets, the accuracy of the embedding would deteriorate to the point where considering a diffusion model would no longer be meaningful.

However, despite these current challenges, there are potential solutions. Previous research focused on embeddings onto unit hyperspheres, but it is possible to consider hyperspheres with a radius of $\sqrt{n}$. When the dimensionality is $n$, the surface area of a hypersphere with a radius of $\sqrt{n}$ increases monotonically, suggesting that the embedding could remain effective even as the dimensionality increases. In this case, efficient generation using diffusion models that leverage the concept of tubular neighborhoods could become meaningful.

## K  ANALYSIS OF SCORE VECTOR

### K.1  SCORE VECTOR FIELD

We present additional experiments detailing the score vectors of DDPM. This section includes two experimental setups concerning the score vector field. Firstly, for the $S^1$ case, the experimental setup includes a grid size of $32 \times 32$ and a trained DDPM with $T = 1000$. The training data is $S^1$, with the red circle at the centre representing $S^1$. See Figure 21 for the corresponding visualization.
Secondly, for the $S^2$ case, the experimental setup includes a grid size of $16 \times 16 \times 16$ and a trained DDPM with $T = 1000$. The training data is $S^2$. Except for the grid size and training data, all other settings remain the same. See Figure 22 for the corresponding visualization.

Thirdly, for the ellipse case described in Section 5.2, we visualized the score vector field for the initial two dimensions of a 16-dimensional latent space. The experimental setup involves training data generated from an ellipse with radii $R = 2$ and $r = 1$. The grid size is $32 \times 32$, and the visualization highlights the behavior of the score vectors in these two dimensions. See Figure 23 for

the corresponding visualization.

Finally, for the disjoint arcs case described in Section 5.3, we visualized the score vector field for the initial two dimensions of a 16-dimensional latent space. The experimental setup involves training data composed of two disjoint arcs, one from a circle with radius $R = 2$ and the other from a circle with radius $r = 1$. The grid size is $32 \times 32$, and the visualization illustrates the interactions between the two arcs. See Figure 24 for the corresponding visualization.

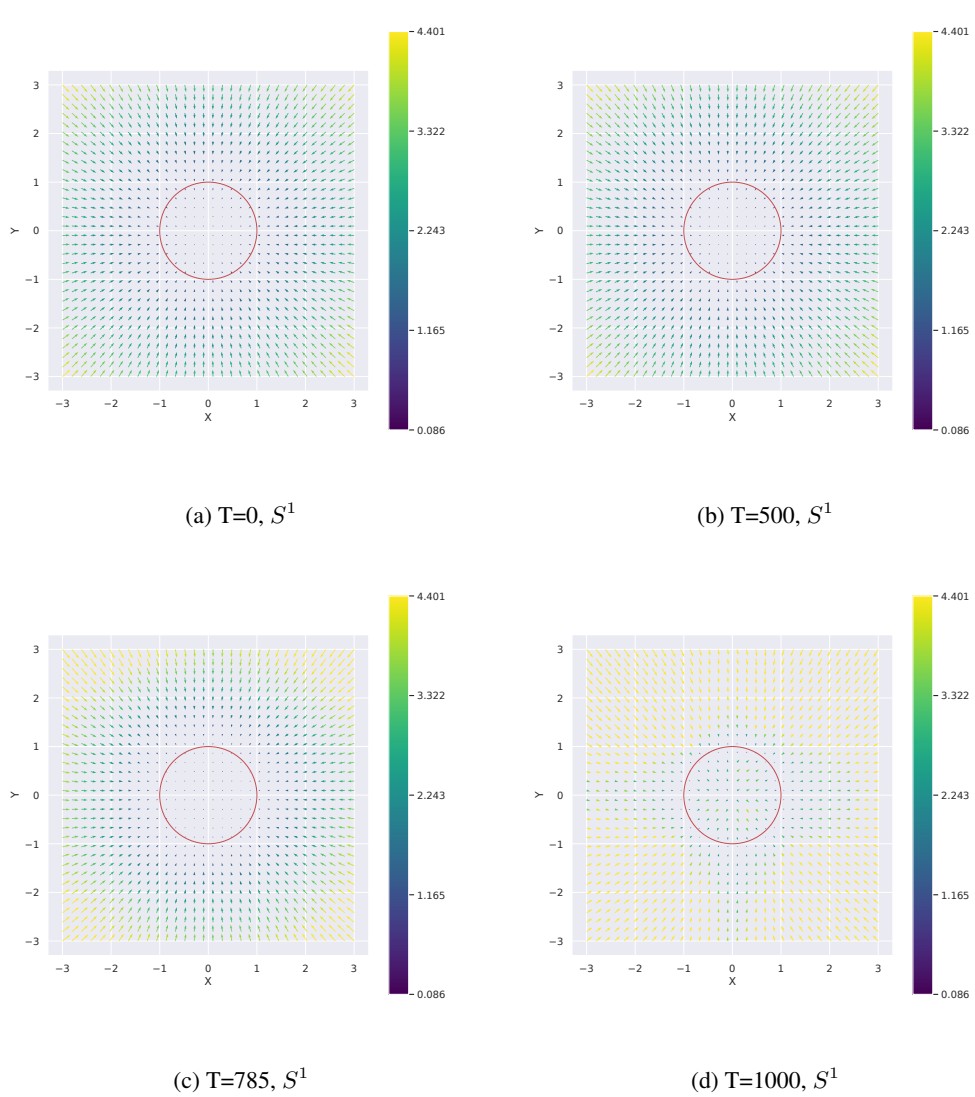

(a) T=0, $S^1$    (b) T=500, $S^1$

(c) T=785, $S^1$    (d) T=1000, $S^1$

Figure 21: Time evolution of score vectors in the backward process of DDPM, $S^1$

## K.2 SQUARE OF THE JACOBIAN $J$ OF THE SCORE VECTOR FIELD

In this section, we extend our analysis to the square of the Jacobian $J$ of the score vector field. We utilise updated experimental setups for both the 2D $S^1$ and the 3D $S^2$ cases. For the 2D $S^1$ case, the grid size is $128 \times 128$ with a trained DDPM using $T = 1000$. The training data remains $S^1$, and we compute and analyse the square of the Jacobian of the score vector field for this setup (Figure 25).

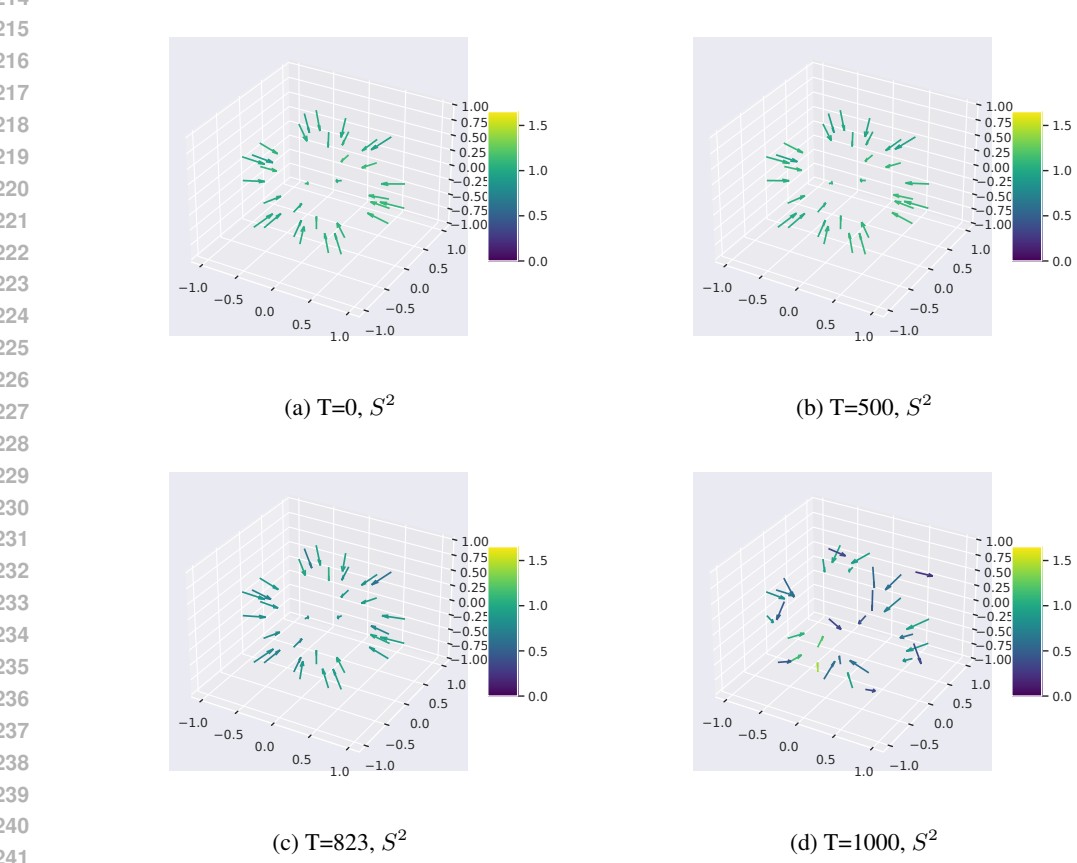

(a) T=0, $S^2$

(b) T=500, $S^2$

(c) T=823, $S^2$

(d) T=1000, $S^2$

Figure 22: Time evolution of score vectors in the backward process of DDPM, $S^2$

Similarly, for the $3D$ $S^2$ case, the grid size is $128 \times 128 \times 128$ with a trained DDPM using $T = 1000$. The training data remains $S^2$, and we compute and analyse the square of the Jacobian of the score vector field for this setup (Figure 26).

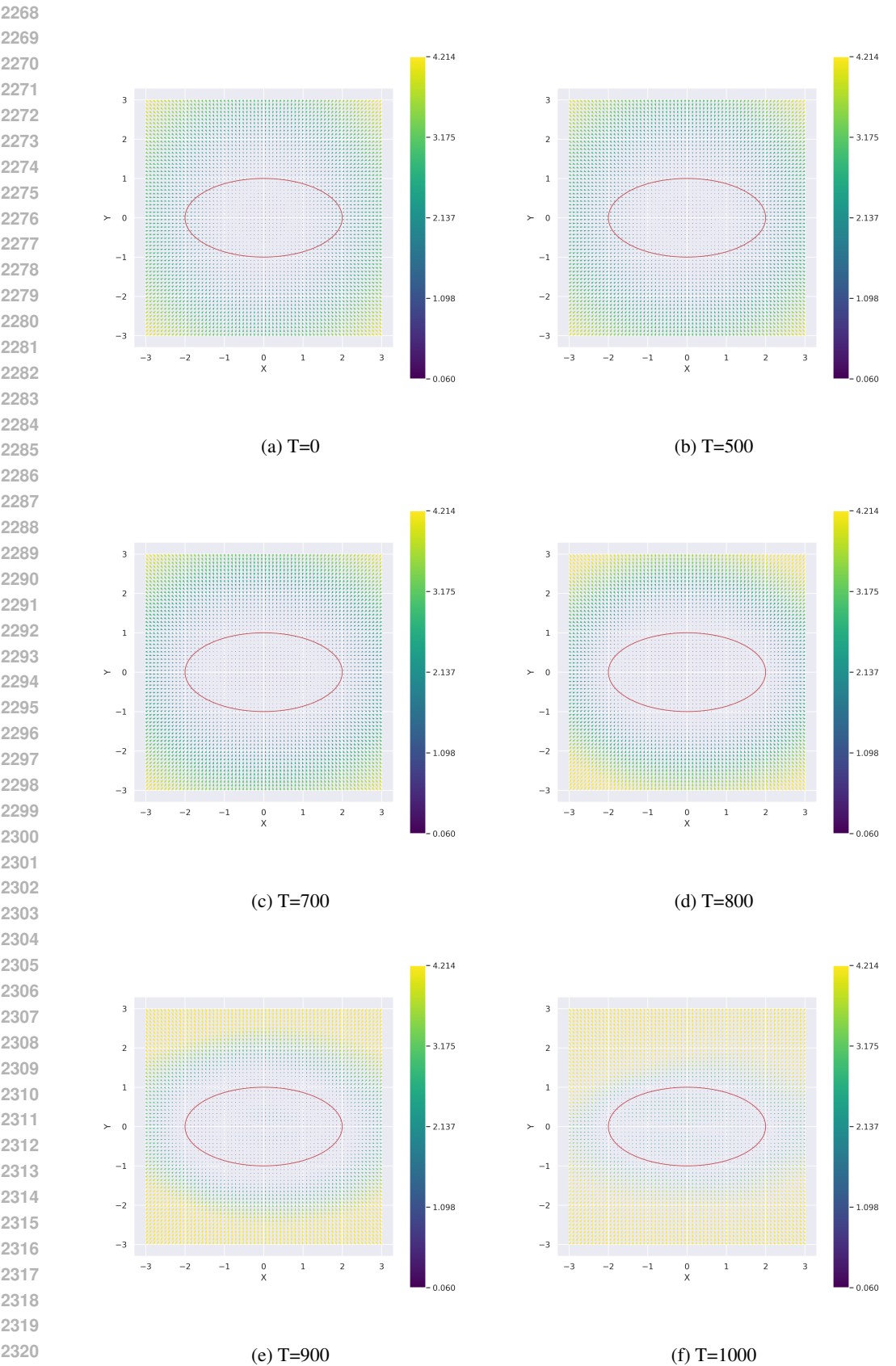

(a) T=0

(b) T=500

(c) T=700

(d) T=800

(e) T=900

(f) T=1000

Figure 23: Time evolution of score vectors in the backward process of DDPM, Ellipse $(R = 2, r = 1)$

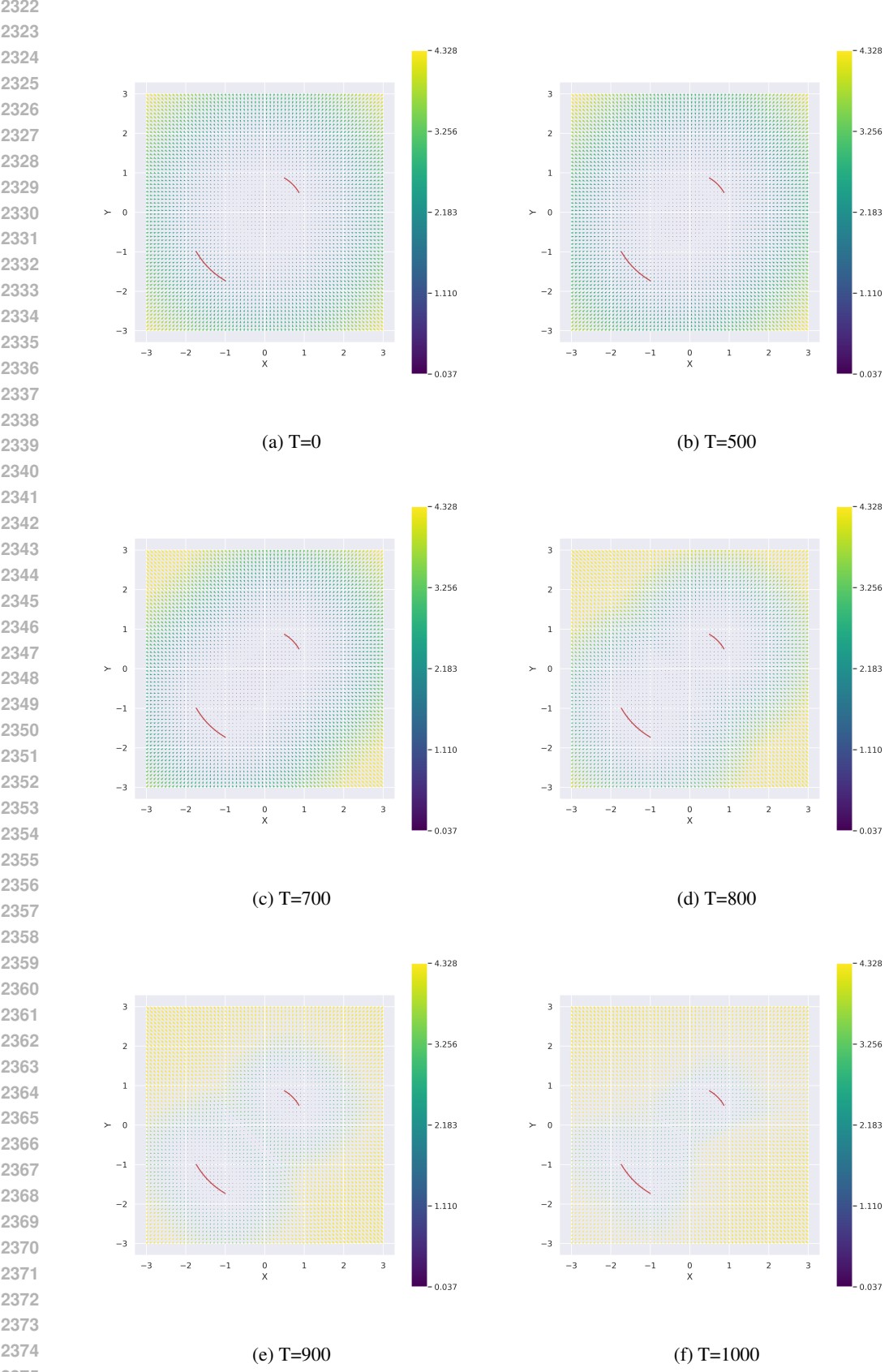

(a) T=0

(b) T=500

(c) T=700

(d) T=800

(e) T=900

(f) T=1000

Figure 24: Time evolution of score vectors in the backward process of DDPM, Disjoint arcs case $(R = 2, r = 1)$

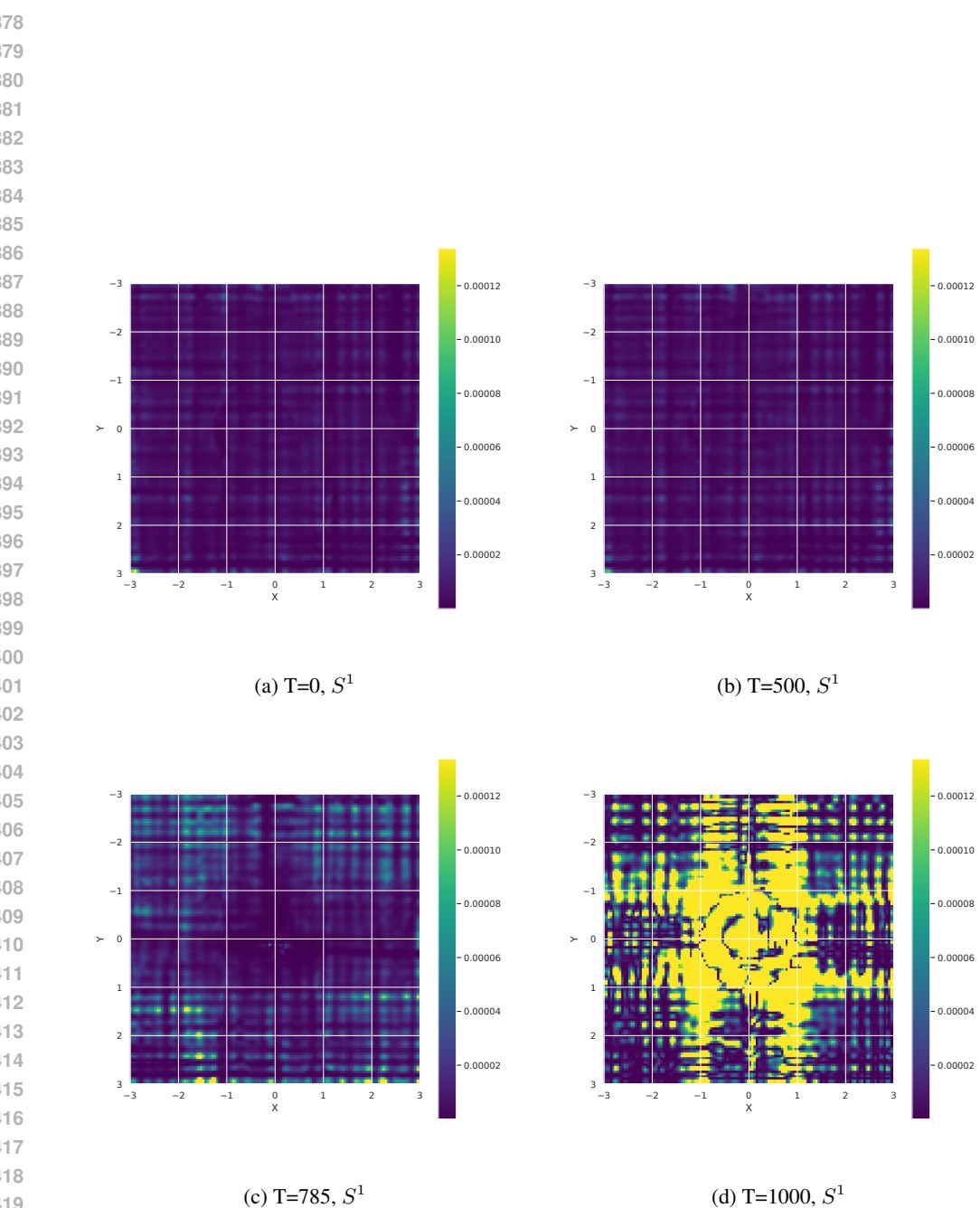

(a) T=0, $S^1$

(b) T=500, $S^1$

(c) T=785, $S^1$

(d) T=1000, $S^1$

Figure 25: Time evolution of the squared Jacobian of score vectors in the backward process of DDPM, $S^1$

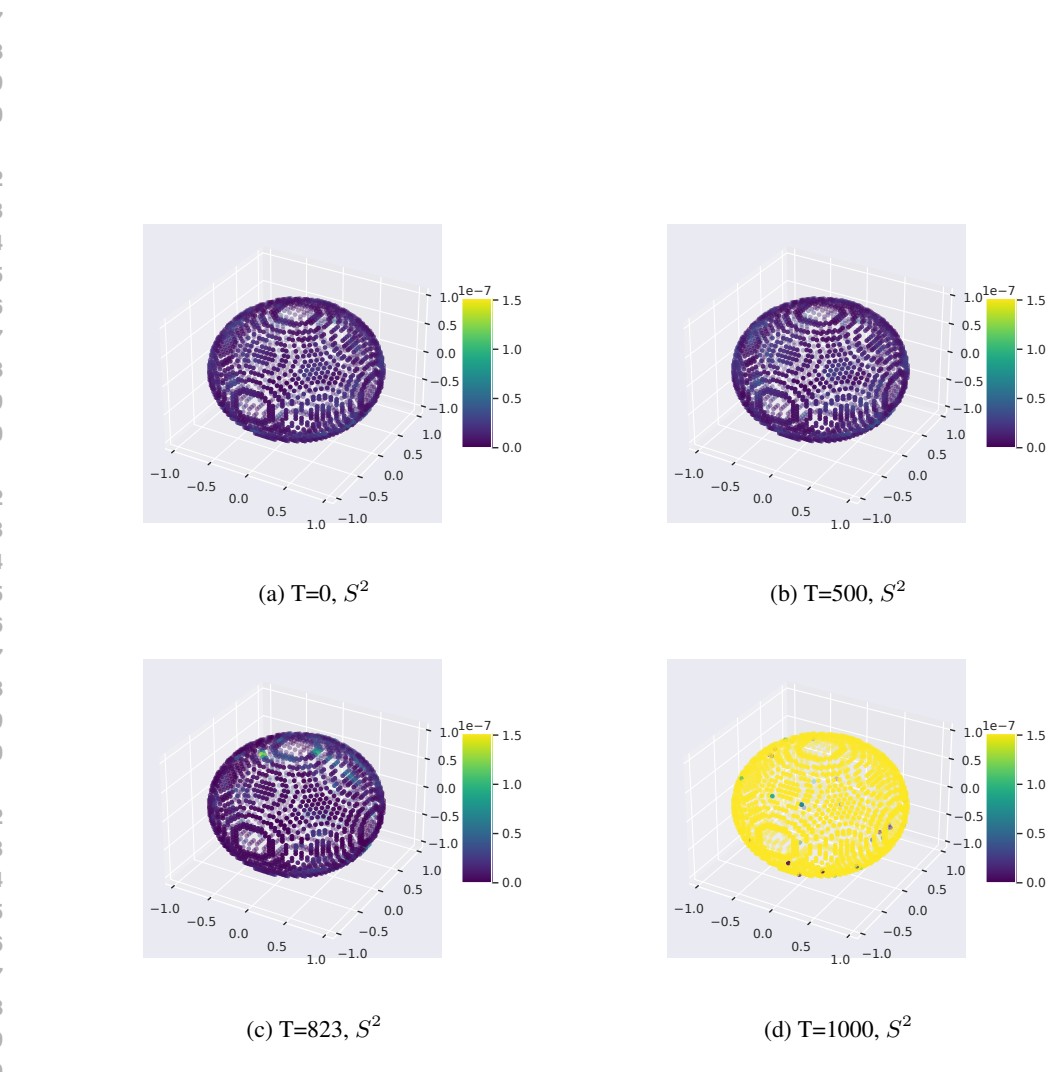

(a) T=0, $S^2$

(b) T=500, $S^2$

(c) T=823, $S^2$

(d) T=1000, $S^2$

Figure 26: Time evolution of the squared Jacobian of score vectors in the backward process of DDPM, $S^2$

