# OpenReview forum: "The Geometry of Phase Transitions in Diffusion Models: Tubular Neighbourhoods and Singularities"
_ICLR.cc/2025/Conference — Submitted to ICLR 2025_

### Official Review · Reviewer_TUbi · 2024-10-29

**Soundness:** 4
**Presentation:** 3
**Contribution:** 3
**Rating:** 8
**Confidence:** 4

**Summary:**

The paper presents a detailed geometric analysis of spontaneous symmetry breaking phenomena in generative diffusion models under the manifold hypothesis. To go beyond the very symmetric cases discussed in (Raya, 2023), the authors study the injectivity radius of the manifold, which is the supremum of possible radii of tubular neighborhoods of the manifold. The main idea is that, if at time t the probability mass of the forward kernel is mostly inside the maximal tubular neighborhood, then  all 'generative decisions' are already been made and the particles are projected by the score with high probability on a single well-defined target point.
Therefore, at least locally, there will be a finite value of t at which the symmetries of the generative process break spontaneously, meaning that, around a given point of the manifold, the distributions of the generative paths collapse on deterministic targets.
The authors show that these local phase transitions depend on the curvature of the manifold, with higher curvature leading to later critical collapse times. This time corresponds to the disappearance of singularities (i.e. points of non-smoothness) on the epsilon-neighborhood of the manifold.

The paper contains both theoretical results and experimental analyses on real datasets. The main result of the theoretical analysis is the characterization of the critical times based on the local curvature. The experiments show the results of late initialization as a way of detecting the first symmetry breaking event as introduced in (Raya, 2023). The authors carefully control the geometry of the target data and the deviation of the distributions in order to control the phenomenon of mean and covariance shift prior to the symmetry breaking event.

**Strengths:**

The paper successfully addresses the main limitation of the theoretical approach used in (Raya, 2023), namely that the analysis of the fixed points is only practical in highly idealized models.
The analysis of the injective radius of the tubular neighborhood offers a very precise and powerful tool to characterize more complex symmetry breaking phenomena that can better capture the richness of real data. These new tools offer an important stepping stone in our theoretical understanding of generative diffusion and its relations with fundamental concepts in statistical mechanics and differential geometry.

The theoretical analysis has the rare property of being both rather rigorous and beautifully intuitive. The figures work extremely well in explaining the main ideas from a visual geometric perspective. On the other hand, the mathematical structure allows for precise calculations to match these intuitions.

The experimental results are not spectacular but they offer a solid validation of the main ideas. I appreciated the breath of the geometric configurations that have been studied and the effort to rigorously control potential confounds.

**Weaknesses:**

I think that the main missing link in the paper is a direct connection with the fixed-point method used in (Raya, 2023). In geometric settings, if the data distribution is uniform when restricted to the manifold, it is possible to define the latent manifold at time t as the set of stable fixed points of the vector field, i.e. the minima of the potential energy function.
Intuitively, the appearance of a singularity in your framework should correspond to the 'collapse' of a portion of the data manifold into a single point in the latent manifold at time t, this is equivalent to the bifurcation of isolated fixed-points corresponding to Curie-Weiss type transitions. For example, it is easy to see that a hyper-sphere collapses into a single latent point when the injective radius is exceeded.
In the general case of unequal curvature, I would surmise that this phenomenon happens locally.
It would therefore be very insightful to study how these geometric singularities correspond to the geometry of the potential energy function, which will connect the theory more directly with the theory of critical phase transitions in physics.


I see two further weak points in the analysis, which however could be addressed in future work.

1) While it is interesting to study the first symmetry breaking event as the main phase transitions, it would be insightful to provide a more detailed analysis on the multiple critical times that arise in manifolds with non constant curvature. Under your model, the first critical time corresponds to the disappearance of the singularities around the points with the highest curvature. This is the first visible symmetry breaking event and it therefore demarcates the first main deviation from approximate Gaussianity in the marginal distribution, as visible from the late initialization curves.
While this effect is very visible and very important in the generative process, it is not necessarily more conceptually important than the further critical times corresponding to regions with lower curvature. In fact, at least locally, these correspond to other genuine critical transitions where important 'generative decisions' are made to spontaneous symmetry breaking events.

While the theory can be easily applied to the analysis of these subsequent transitions, I would have appreciated direct experimental verification of these events under controlled conditions. Note that this would required a different approach than that used in the late initialization experiments.

2) It would have been very nice to have a characterization of the phase transitions in this geometric setting in the language of statistical physics, since the connection with (Raya, 2023) suggests a deep link. However, I assume that the lack of details on the physics side comes from the background of the authors and it can be remedied by future work on the subject.

**Questions:**

1) Could you provide a theoretical connection between your geometric method and the fixed-point analysis presented in (Raya, 2023)?

2) Could you expand your experiments to provide evidence of separate transition times in manifolds with defferent local curvatures?

---

> ### Author Response · Authors · 2024-11-15
> **Official Comment by Authors [Weakness, Q1-2]**
>
> Thank you for your thoughtful and detailed review. We appreciate your deep understanding and positive feedback on our theoretical framework and experimental results, as well as your constructive suggestions. Below, we address your main points and questions.
>
> ---
>
> **Weakness:**
> As you pointed out, the appearance of a singularity in our framework corresponds to the collapse of a part of the data manifold to a single point at time $t$, which is analogous to the bifurcation of an isolated fixed point in a Curie-Weiss type transition. At the same time, our framework deals with phase transitions occurring in non-equilibrium phenomena in the form of diffusion processes, and we believe that it is necessary to extend the existing phase transition theories that have been developed to explain phase transitions in equilibrium systems.
>
> Guided by reviewer TUbi's comments, we recognized the need to develop a more complete and rigorous theory of phase transitions in physics and the occurrence of phase transitions in diffusion processes. We look forward to continued discussions, including possible collaboration.
>
> ---
>
> **Connection to Fixed-Point Analysis (Weakness, Q1):**
> We agree that establishing a stronger theoretical connection between our geometric method and the fixed-point analysis in Raya and Ambrogioni (2023) would be highly insightful. Our approach indirectly incorporates the idea of fixed points by analyzing the behavior of $\nabla u$ at the boundary of the tubular neighborhood (Proposition 4.2). Specifically, the change in the proportion of particles within the tubular neighborhood, $\Gamma(t)$, corresponds to the integral of $p_t(x) \nabla u_t(x)$ over the boundary, providing an analog to the fixed-point framework in terms of analysis of free energies.
>
> While Raya and Ambrogioni (2023) focus on the origin as a fixed point under ideal conditions, our framework naturally extends to scenarios with varying curvatures by leveraging the injectivity radius. This allows us to capture phase transitions at the boundary where the data manifold approximation breaks down. For instance, in a simple 1D example where the manifold consists of two points $\{1, -1\}$, the boundary of the tubular neighborhood is $\{0, 2, -2\}$. In the hypersphere case, the boundary is a union of the origin and a larger sphere of radius 2.
>
> In our framework, $\Gamma(t)$ aligns with the fixed-point dynamics in some sense. We acknowledge the potential for deeper connections between our approach and the critical transitions in physics and plan to explore this in future work. If you are interested, we would welcome collaboration to further develop this connection.
>
> ---
>
> **Multiple Critical Times and Curvature (Weakness, Q2):**
> Your suggestion to study multiple critical times arising from varying local curvatures is well-taken. In our current work, we demonstrate this phenomenon using an experiment with disjoint arcs of different radii (Section 5.3). However, we acknowledge that this is only a preliminary step. To provide more robust evidence, we plan to conduct additional experiments with varied curvatures and datasets that combine multiple local geometries. This will allow us to better illustrate how separate transition times arise in such settings.
>
> ---
>
> **Experimental Variations and Statistical Physics Framework:**
> We agree that connecting our findings to the language of statistical physics would enhance the conceptual depth of our work. While this was beyond the scope of the current paper, we see this as a promising direction for future research and appreciate your suggestion.
>
> ---
>
> Thank you once again for your very insightful feedback. We will revise the manuscript to address these points and include the proposed experimental extensions.

---

> > ### Comment · Reviewer_TUbi · 2024-11-20
> >
> > Dear authors,
> >
> > Thank you for the reply. I am happy to keep my original score. While I think that a more direct connection with statistical physics would be insightful, I still think that the contribution offers an interesting and useful perspective.

---

> ### Author Response · Authors · 2024-11-25
> **Acknowledgment of Reviewer’s Feedback**
>
> We sincerely appreciate your constructive comments and your support of our submission. We intend to investigate this direction in future research and believe it could further enhance the relevance and impact of our contributions.

---

### Official Review · Reviewer_gsx9 · 2024-10-30

**Soundness:** 2
**Presentation:** 2
**Contribution:** 2
**Rating:** 3
**Confidence:** 3

**Summary:**

This paper investigates the phase transition phenomena in the generative process of diffusion models from a geometric perspective. The authors first define the injectivity radius, characterize it as the minimum of the first and second injectivity radii, and propose a practical algorithm for estimating the first injectivity radius. Subsequently, the paper computes the proportion of probability mass within the tubular neighborhood and the Wasserstein distance using the late initialization scheme. Through various experiments involving toy models and the MNIST dataset, the paper concludes that the phase transition corresponds to a rapid decrease in the proportion of particles outside the tubular neighborhood and a sharp increase in the Wasserstein distance. This study illuminates how geometrical structure interacts with the training and inference process of diffusion models, paving the way for further research to develop specific algorithms accordingly.

**Strengths:**

- The paper presents a novel perspective that establishes a connection between the geometric characteristics of the data distribution and the phase transition phenomena observed in the Wasserstein distance during the generative process of diffusion models.
- The paper presents rigorous mathematical arguments and definitions for all concepts introduced in this context, including the injectivity radius and tubular neighborhood. These arguments provide a solid foundation for the experimental reasoning presented in the paper.
- The paper presents compelling experimental evidence that demonstrates the correlation between the increase in the Wasserstein distance and the proportion of particles within the tubular neighborhood.

**Weaknesses:**

- The paper’s scope remains confined to toy models, and there are several instances where the experimental outcomes deviate from the conjectured behavior.
- The algorithm for computing the injectivity radius and the proportion of particles within the tubular neighborhood lacks a clear and comprehensive description. Algorithm 1 requires revision.
- The Wasserstein distance is not explicitly defined between which two probability distributions it is measured.

**Questions:**

Despite the promising experimental results, I remain uncertain about the direct relationship between the increase in the Wasserstein distance and the proportion of particles within the tubular neighborhood. My primary concern is that:
- The injectivity radius appears to be fixed for a specific data distribution. Consequently, for any chosen radius (which may or may not be the injectivity radius), a corresponding tubular neighborhood can be defined. Furthermore, based on my understanding, the red curves in all the figures exhibit a similar trend, given the characteristics of the forward process. So it is unclear if the injectivity radius defined in the paper is a pivotal factor for this phenomenon to appear.
- In several experiments, the timing of the increase in the Wasserstein distance does not precisely align with the 0.99 threshold. This suggests that the conjecture may be attributed to a coincidental association with the exponential diffusion of singularities in the OU process.

To strengthen the conjecture, additional experiments are necessary to provide corroborating evidence. These experiments could include:
- Statistical and quantitative evaluation of the correlation between the Wasserstein distance and the proportion within the tubular neighborhood.
- Experiments conducted on a family of datasets with varying injectivity radii, where the corresponding curves are observed and their relationships with the injectivity radii are assessed.
- Reparameterization of the inference process to extend the final portion, enabling a more precise observation of the phase transition.

---

> ### Author Response · Authors · 2024-11-15
> **Official Comment by Authors [Weakness1-3, Q1-5]**
>
> Thank you for your detailed feedback and constructive suggestions. Below, we address your concerns point by point.
>
> ---
>
> **Experimental Results and Scope (Weakness 1):**
> We acknowledge that some experimental results, such as Fig. 5, may not seem to perfectly align with the theoretical predictions. However, this represents an important insight worth noting. The experiments in Fig. 5 demonstrate that this behavior, as discussed in Sec 5.2, arises from non-uniform curvature (Fig. 5(b), (c)). For cases like Fig. 5(a), the misalignment stems from the distance between the initial distributions in the forward and reverse processes. Additional results in Appendix J.3 support this analysis, showing alignment when the Gaussian noise variance is reduced, which is why the discrepancy occurs. We will also consider conducting additional experiments with datasets exhibiting varying curvatures to further validate the importance of the injectivity radius, as suggested.
>
> ---
>
> **Algorithm 1 Description (Weakness 2):**
> We apologize for the lack of clarity in Algorithm 1. A detailed explanation is provided in P26. To enhance comprehensibility, we will include intuitive explanations for each step in the appendix. Please let us know if there are specific aspects requiring further clarification.
>
> ---
>
> **Definition of Wasserstein Distance (Weakness 3):**
> The Wasserstein distance in our experiments is computed between the initial distribution of the forward process, $p_0(x)$, and the final distribution of the reverse process, $q_0(x)$, for each delayed initialization time $T$. We will revise the captions in relevant figures to clarify this.
>
> ---
>
> **Correlation Between Proportion and Wasserstein Distance (Q1):**
> It is known that in the final stages of the generation process, score vectors on an $\epsilon$-neighborhood are orthogonal to the tangent plane of the data manifold (refer to Stanczuk et al. (2024)). For vectors within a tubular neighborhood to be orthogonal to the tangent plane of the data manifold, they must be within the injectivity radius (see Theorem D.1 in Stanczuk et al. (2024) or Proposition 4.7 in Section 4). Therefore, particles are effectively drawn into the tubular neighborhood.
>
> In fact, as shown in Fig. 4, the red line drops sharply. The time at which this sharp drop occurs corresponds to the point when particles are being absorbed into the tubular neighborhood. Consequently, if this time period is truncated through late initialization, particles will no longer be influenced by the score vector field as they naturally would. This makes it significantly more difficult for the particles to reach points on the data manifold, leading to an increase in the Wasserstein distance.
>
> We will conduct ablation studies to validate this perspective.
>
> ---
>
> **Timing of Phase Transitions (Q2):**
> In Fig. 5(a), (b), and (c), the timing of the Wasserstein distance increase doesn't seem to be precisely aligned with the 0.99 threshold. For Fig. 5(a), this apparent misalignment is due to the separation between the Gaussian initialization region and the training data distribution. Appendix J.3 demonstrates how reducing the variance of the Gaussian initialization supports our hypothesis. For Fig. 5(b) and (c), the discrepancy arises from varying curvatures in the data distribution, as intended. These observations do not undermine our claim that the injectivity radius plays a significant role in the final stages of the generative process.
>
> ---
>
> **Statistical Visualization of Phase Transitions (Q3):**
> We can provide a summary of the Wasserstein distance values before and after the threshold to better illustrate the phase transitions. Please let us know if this would address your concern.
>
> ---
>
> **Experiments with Different Values of $R$ and $r$ (Q4):**
> We will include additional experiments varying the values of $R$ and $r$ in Section 5.3 to further validate our findings.
>
> ---
>
> **Reparameterized Observation of Diffusion Time (Q5):**
> If you are requesting a closer examination of smaller diffusion times, we can provide an expanded view of the range (e.g., $0$–$200$) in Fig. 5. Please let us know if this addresses your concern.
>
> ---
>
> We appreciate your valuable suggestions and will incorporate them in the revised manuscript to strengthen our contributions.

---

> > ### Comment · Reviewer_gsx9 · 2024-11-24
> >
> > Thank you for your responses. However, as far as I understand, I still feel that the claimed correlation between the geometric characteristics of the data distribution and the phase transition phenomena observed in the Wasserstein distance during the generative process of diffusion models requires further investigation before a conclusion can be drawn. Additionally, the presentation of the paper, including figures, could benefit from improvements. Therefore, I am maintaining my score.

---

> > > ### Author Response · Authors · 2024-11-27
> > > **Additional Comment by Authors Regarding Q1**
> > >
> > > As mentioned in our response to Q1, the tubular neighborhood defined by the injectivity radius holds significant meaning distinct from arbitrary neighborhoods, as it represents the region where the transitions of particles determined by score vectors are uniquely defined.
> > >
> > > To address your concerns, we have conducted additional experiments. Specifically, regarding whether the red line exhibits different behavior, we considered the DISJOINT ARCS CASE setting in Section 5.3 with parameters $(R, r) = (3, 1)$, meaning the injectivity radius is 1. The results of these experiments, including the behavior of the red line (representing the proportion of particles outside the tubular neighborhood defined by the injectivity radius) and the red dashed line (representing the proportion of particles outside a neighborhood region with $R = 3$, which is not the injectivity radius), are shown in Figure18 in Appendix J.5. These results demonstrate that the behavior of the red line varies depending on the chosen value for the neighborhood region.
> > >
> > > Additionally, we have included further experiments on the DISJOINT ARCS CASE in Appendix J.5, as well as additional experiments on the behavior of the score vector fields in Appendix K. We hope these experiments will enhance your understanding of the injectivity radius and its implications.

---

### Official Review · Reviewer_55op · 2024-11-03

**Soundness:** 2
**Presentation:** 2
**Contribution:** 2
**Rating:** 3
**Confidence:** 3

**Summary:**

The paper proposes a new viewpoint that the boundary of the tubular neighborhood of the underlying data manifold. This is an intuitively justifiable claim worth pursuing. However the arguments in the paper are not sufficiently solid.

**Strengths:**

The authors defined tubular neighborhoods and their injective radius, and proposed formulas (Theorem 3.7) to compute these radii. They also proves that the score vector fields at the boundary always point towards the interior of the neighborhood, i.e. diffusion trajectories tend to be attracted into the neighborhood.

**Weaknesses:**

There are a few of flaws in the paper's reasoning in my opinion.
1. The paper doesn't define what a phase transition actually means or propose hypothesis on mechanism behind it. In particular, what makes the tubular neighborhood special among all neighborhoods? The message delivered by what is currently written is rather "trajectories will eventually enter the tubular neighborhood" and "some kind of phase transition occurs when trajectories get closer to the data manifold". However the same can also be said about neighborhoods of different radii. It would be helpful if there could be an argument supporting that transition starts once points enter the tubular neighborhood, or once they are sufficiently close by another unknown standard. For example, in Figure 5, it is not obvious whether the transition points match between the red and blue curves.
2. The paper uses an oversimplified setting of a single injective radius for all points in the manifold. However, because: (i) the validity of manifold hypothesis in full generality is questionable see e.g. "Verifying the union of manifolds hypothesis for image data" (Brown et. al ICLR 2023) and intrinsic dimensions may vary among data points, (2) even assuming the manifold hypothesis, in practice the manifold is usually very wiggly, resulting in high variance in both R_1(M) and R_2(M). What "entering the tubular neighborhood" means might become ambiguous in such a setting.
3. The experiments uses a simple setting of unit spheres with an intrinsic radius of 1. This setting is not very helpful because the intrinsic radius is critically determined at the origin, while in the actual diffusion model experiment, the majority (or at least a large percentage) of points approach the sphere from the outside, they don't really "see" the injective radius in the whole process. The emphasis on the injective radius in this case merits more explanation. Similar problem exists in the experiments on ellipses or tori. For instance, for ellipses, the injective radius is determined by the length of the short axis and the principal curvature at the end of long axis, and again the majority of trajectories will approach the manifold without out seeing these factors at all. For the experiments with MNIST and FMNIST, I think the method of embedding them into S^d  ignores their own intrinsic geometries and is inadequate, precisely by the authors's own analysis at the beginning of Section 5.4. Namely the datasets's intrinsic geometry is what matters here, a spherical embedding doesn't change the property that the intrinsic radius is small (and varying drastically among data points). Embedding it into S^d and pretending to have a fake injective radius of 1 completely disconnects the analysis from the reality for actual diffusion processes near the data distribution.

**Questions:**

1. The experiment results record "Wasserstein distances W for different late initialisation times". You should tell the reader this is the Wasserstein distance between which two distributions. Why is it increasing when t->0? (When assessing diffusion models, one often look at Wasserstein distance between the generated distrbution and the ground truth one, which is suppose to decrease as t->0.)
2. R_1(M) should be closely related to the reciprocal of the largest principal curvature at each point. Reference: Section 4 from "Reconstruction and interpolation of manifolds I: The geometric Whitney problem", Fefferman et al. 2021 .

---

> ### Author Response · Authors · 2024-11-15
> **Official Comment by Authors [Strength, Weakness 1-3]**
>
> Thank you for your detailed review and constructive feedback. Below, we address your concerns and suggestions point by point.
>
> ---
>
> **On the Strength:**
> Thank you for mentioning the strength. However, the results of our paper, more precisely as stated in Proposition 4.7, hold under conditions related to the "injectivity radius" and the time step. Proposition 4.7 establishes a sufficient condition for the score vector to point inward, as specified by inequality (10). It explains how larger injectivity radii $\epsilon_0$ facilitate satisfying this condition, whereas higher ambient dimension $d$ imposes more stringent constraints on the time step. We repeat that this is only a snippet of the assertion. Please read again carefully and investigate what is written in Proposition 4.7.
>
> ---
>
> **Phase Transition Definition and Tubular Neighborhood (Weakness 1):**
> You raise an important point regarding the definition of phase transitions and the special role of tubular neighborhoods. In our work, we view phase transitions as significant changes in the Wasserstein distance, supported by the theoretical framework of tubular neighborhoods. These neighborhoods are special because they are directly tied to the injectivity radius, which reflects the highest curvature regions of the data manifold. The disjoint arcs experiment (Fig. 7) demonstrates this, showing phase transitions corresponding to the highest curvature regions.
>
> Regarding Fig. 5, we acknowledge that in some cases, transitions occur before particles fully enter the tubular neighborhood. This behavior, as discussed in Sec 5.2, arises from non-uniform curvature (Fig. 5(b), (c)). For cases like Fig. 5(a), the misalignment stems from the distance between the initial distributions in the forward and reverse processes. Additional results in Appendix J.3 support this analysis, showing alignment when the Gaussian noise variance is reduced.
>
> ---
>
> **Simplified Experimental Settings (Weakness 2):**
> We agree that our current settings are very simplified. Nevertheless, these choices aim to overcome limitations in prior work (e.g., Raya and Ambrogioni (2023)) and establish a clear baseline. The main problem to solve its limitation is that one does not simply analyze the free energy of the dynamics mathematically. In the case of ellipses, it is already a very difficult mathematical problem. We do think extending the analysis to more complex manifolds is indeed valuable future work. Computing free energy for more complex structures is mathematically challenging, and our approach seeks to address phase transitions using the tubular neighborhood framework as a practical alternative. Furthermore, this approach suggests a connection with the field of differential geometry and can be seen as a giant leap in interdisciplinary research.
>
> ---
>
> **Injectivity Radius in Diffusion Process (Weakness 3):**
>
> We agree that the significance of the injectivity radius requires further explanation:
> - The score vector field plays a critical role in the behavior of diffusion models (Equation (2)).
> - The behavior of the score vector field is determined by the injectivity radius (Proposition 4.7).
>
> Given these facts alone, wouldn’t it be reasonable to say that the injectivity radius is a key factor in diffusion models? It is correct that curvature only accounts for partial behavior in diffusion models. However, the claim that the injectivity radius also captures only partial behavior is incorrect. Instead, it should be understood that the injectivity radius strongly influences the behavior at the final stages of the generation process (Proposition 4.7).
>
> It seems that you are confusing curvature with injectivity radius. Curvature is a local quantity, while the injectivity radius is a global quantity. The guiding principle of statistics is to "always consider global quantities and understand them in global terms." If the statistical methods have a promise for mankind, it is better to consider global quantities rather than local ones.
>
> While we acknowledge that spherical embeddings simplify the intrinsic geometry of datasets like MNIST and FMNIST, our results demonstrate consistent qualitative and quantitative behavior regarding phase transitions at the boundaries defined by the injectivity radius. This supports the applicability of tubular neighborhood theory even under these simplifications.

---

> > ### Author Response · Authors · 2024-11-15
> > **Official Comment by Authors [Q1-Q2]**
> >
> > **Clarifying the Use of Wasserstein Distance (Q1):**
> > As you suggested, we will clarify that the x-axis in our experiments represents the time delay in initializing Gaussian noise. The Wasserstein distance is computed between the generated distribution initialized with delay $t$ and the ground truth distribution. The increase in Wasserstein distance as $t \to 0$ reflects the divergence between these distributions.
> >
> > ---
> >
> > **Injectivity Radius and Curvature (Q2):**
> > We agree that the injectivity radius is closely related to the reciprocal of the largest principal curvature, as highlighted in *"Reconstruction and Interpolation of Manifolds I"* (Fefferman et al., 2021). Thank you for the reference. We will incorporate this citation and emphasize its connection to our work in Appendix D.3.
> >
> > ---
> >
> > We appreciate your valuable feedback and will revise the manuscript to address these points. Thank you again for your constructive comments.

---

> > > ### Author Response · Authors · 2024-11-27
> > > **Additional Comments by Authors Regarding Q2**
> > >
> > > Thank you again for suggesting the article.
> > > We have added the citation: *"Reconstruction and interpolation of manifolds I: The geometric Whitney problem," Fefferman et al., 2021,* after Theorem 3.6 in Section 3, as suggested.
> > > As you correctly pointed out, the quantity $ R_1 $ is closely related to the curvature of a given manifold.
> > > However, the concrete estimation of $ R_1 $ (and the injectivity radius $ R $) is generally challenging.
> > > In our work, we considered manifolds embedded in Euclidean spaces and demonstrated that it is possible to estimate $ R_1 $ in this context (Theorem 3.7).
> > > This approach appears sufficiently general for practical applications and theoretical interpretation.

---

> ### Comment · Reviewer_55op · 2024-11-28
>
> I appreciate the authors' response and get the point that
>
> ```
> The guiding principle of statistics is to "always consider global quantities and understand them in global terms."
> ```
> However,  I still have some reservations about this as the injective radius, as a global quantity, is essentially decided by the most curved area of the manifold.  That neighborhood are priori could occupy only a tiny portion of the manifold while the rest 99% has a much larger local injective radius. This makes the main result only a sufficient condition and not enough to capture the true phase transition happening along generic diffusion trajectories. I would like to keep my score.

---

### Official Review · Reviewer_Rfdu · 2024-11-09

**Soundness:** 2
**Presentation:** 2
**Contribution:** 2
**Rating:** 5
**Confidence:** 3

**Summary:**

Under the data manifold hypothesis paper puts forward a geometrical perspective on diffusion models based on the concept of (maximal) tubular neighborhoods and manifold curvature. They derive a condition characterizing
the maximal radius (or more precisely the first injectivity radius) of the tubular neighborhood which is then exploited in an algorithm that allows its estimation.
They then provide some characterization of the direction of the score function on the boundary of a manifold neighborhood in the case of p_0(x) uniform on the manifold.
They then embark on a set of experiments on diffusion models, first with very simple synthetic datasets with simple geometries and then on some real datasets,
claiming a connection between the injectivity radius and the dynamical phase transition (sudden changes of behavior) of the diffusion process (either reverse or forward).

**Strengths:**

The paper in principles would provide an interesting perspective on the temporal dynamics of the generative diffusion model based on the geometry of the data manifold and the concept of tubular neighborhood. They provide analytical and algorithmics tools to characterize the neighborhood. They perform experiments on good variety of synthetic datasets and investigate real dataset as well.

**Weaknesses:**

It is hard to understand the claims, due to vague or missing explanations, misleading captions and so on.

**Questions:**

1. The main set of experiments relies on performing late initialization, which means if I understand correctly,
run the reverse diffusion dynamics  starting from a time T which is generic and not large as usual, while the configuration
is still Gaussian generated. So the configuration at time T could be higly unlikely according to forward process.
Then they claim the a transition happens in T in some Wasserstein distance which they do not define,
and this dependes on the radius of the tubular neightborhood and the fact that the particle enter the neighborhood.
This connection is not elucidated: what is the link between the radius and T? What does it mean to enter the neighborhood
and how it is related to the late initialization? The Wasserstein distance is among which distributions?
I must say that reading the manuscript without this very fundamental explanations has been a frustrating experience.

1. In this paper, the estimation of R2(M) is performed by def-
inition. -> "following the definition" or similar

1. prop. 3.4 names tubular neighborhood before defining it. Move definition c.10 to main?

1. In section 3 the algorithm exploiting Theorem 3.7 to estimate R1 is not clearly stated.

1. In eq. 9, s is supposed to be t instead?

1. Section 4 should convey more intuition on what the propositions imply.

1. Table 1. Honestly based on the caption "Wasserstein distances W for different late initialisation times. ρproportion represents the proportion of particles outside the tubular neighbourhood." I cannot understand
the content of the table. It doesn't seem consistent. The column names are supposed to be proportions (maybe in percentage?) or initalization times?

1. Section 5: "In this section, we empirically demonstrate the presence of phase transitions at the boundary of the
tubular neighbourhood during the generative process of diffusion models. In particular, we analyse
the proportion of particles outside the tubular neighbourhood at each time step and examine the
corresponding changes in the Wasserstein distance."" -> Wasserstein distance among which distributions?

1. proposition 4.2: f(t,x) -> f_t(x) for consistency with Eq. 3. Same for g
1. Authors start talking about phase transitions at the beginning of Section 5, but at these point things are not clear. Which kind of phase transitions? What is changing? Which is the order parameter?
1. What is the vertical line in Fig. 4. It is not explained in the caption...
1. Captions for Fig. 4 and similar, should explaing that the x axis referes to the the "late initialization" time of the reverse process, with T=1000 corresponding to the usual initialization time.

---

> ### Author Response · Authors · 2024-11-15
> **Official Comment by Authors [Q1-Q6]**
>
> **Q1.** Thank you for your detailed and insightful feedback. We appreciate the opportunity to clarify the key concepts related to late initialization, the tubular neighborhood, and the Wasserstein distance.
>
> In our experiments, we measure the Wasserstein distance between the forward process's initial distribution \(p_0(x)\) and the reverse process's final distribution \(q_0(x)\). This distance evaluates the divergence between the training data and the generated data as a function of \(T\), where \(q_0(x)\) is obtained by initializing the reverse process at different time steps.
>
> We made a hypothesis that the tubular neighborhood's radius plays a pivotal role in understanding critical phenomena. It delineates the region around the data manifold where particles are considered aligned with the data distribution. Late initialization impacts the reverse process by bypassing earlier transitions that would guide particles into this neighborhood, thus increasing the Wasserstein distance. Conversely, transitions occurring when particles are far from the neighborhood have minimal influence on the final distribution and can be ignored without negatively affecting performance.
>
> We acknowledge that these connections were not sufficiently detailed in the manuscript and will revise the text to provide clearer explanations of these fundamental concepts. Thank you again for your critical comments, and we hope this response addresses your concerns.
>
> ---
>
> **Q2.** We agree with your suggestion and will revise the phrasing to "following the definition" or a similar expression to improve clarity and precision. We appreciate your careful review and attention to detail.
>
> ---
>
> **Q3.** Thank you for your suggestion. While we have already included a general description in lines 137--138 and provided the formal definition in Appendix C.10 for brevity in the original manuscript, we agree that moving the formal definition to the preliminaries could improve clarity.
>
> ---
>
> **Q4.** Thank you for your comment. Theorem 3.7 is stated alongside Appendix D.1, and the numerical experiments using the pseudocode algorithm are also presented in Section 3.4 and Appendix F. On top of that, could you kindly clarify which part requires further elaboration? This would help us address your concerns more effectively.
>
> ---
>
> **Q5.** Thank you for catching this typo. You are correct; \(s\) should be \(t\) in Equation (9). We will correct this in the revised manuscript. We appreciate your careful review and attention to detail.
>
> ---
>
> **Q6.** We recognize the importance of providing more intuition behind the propositions in Section 4. The primary role of this section is to offer theoretical support for the experimental results presented in Section 5. We will improve the section accordingly.
>
> To enhance clarity, we will expand on the implications of Propositions 4.2 and 4.7 through additional remarks (Remark 4.3, 4.8). Specifically:
>
> - **Proposition 4.2** demonstrates that the time derivative of \(\Gamma(t)\) vanishes at \(t=0\) and \(t=\infty\), which aligns with the red curve's behavior in the Section 5 experiments. It can be understood that \(\frac{d}{dt}\Gamma^{M(\epsilon)}(t)\) represents an integral of \(\nabla u\) over the boundary. Here, the potential energy \(\nabla u\) plays a key role in the diffusion dynamics (Raya and Ambrogioni, 2023). \(\nabla u\) corresponds to the first term of Equation (2) in our paper.
>
> - **Proposition 4.7** provides a sufficient condition for the score vector to point inward, given by inequality (10). We will clarify how larger injectivity radii \(\epsilon_0\) make this condition easier to satisfy, while high ambient dimensions \(d\) impose stricter requirements on the time step. This is only a small snippet of the assertion. Please read again carefully and investigate what is written in Proposition 4.7.
>
> We will revise Section 4 to better convey these intuitions and highlight their experimental relevance. Thank you for your constructive suggestion.

---

> > ### Author Response · Authors · 2024-11-15
> > **Official Comment by Authors [Q7-Q12]**
> >
> > **Q7.** We appreciate your observation. The column names in Table 1 were intended to represent proportions, but they are currently displayed as percentages. We will correct this in the revised manuscript to ensure consistency and clarity.
> >
> > ---
> >
> > **Q8.** The Wasserstein distance in this context is computed between the initial distribution of the forward process, $p_0(x)$ (the training data distribution), and the final distribution of the reverse process, $q_0(x)$. The latter is obtained after delaying the initialization time in the reverse process to different time steps, as analyzed in our experiments. We will clarify this in the revised manuscript.
> >
> > ---
> >
> > **Q9.** You are correct. For consistency with Equation (3), we will update $f(t,x)$ to $f_t(x)$ and $g(t,x)$ to $g_t(x)$ in Proposition 4.2. Thank you for your careful review.
> >
> > ---
> >
> > **Q10.** Thank you for your question. Following Raya and Ambrogioni (2023), we investigate phase transitions characterized by spontaneous symmetry breaking at the origin of a hypersphere. In our work, the sharp increase in the Wasserstein distance between the training data distribution and the generated distribution serves as an indirect indicator of such phase transitions. This phenomenon is observed experimentally as the initialization time in the reverse process (late initialization) is delayed. We will clarify this point in the revised manuscript.
> >
> > ---
> >
> > **Q11.** The vertical purple line in Figure 4 indicates the time step at which the proportion of particles outside the tubular neighborhood reaches 0.99. We will update the caption to include this explanation. Thank you for pointing this out.
> >
> > ---
> >
> > **Q12.** The x-axis in Figure 4, specifically for the blue curve, corresponds to the diffusion time $T$, where $T=1000$ represents the usual initialization time. $T=t$ indicates a late initialization where Gaussian noise initialization is delayed by $1000-t$ steps. We will update the caption to clarify this. Thank you for the suggestion.

---

### Author Response · Authors · 2024-11-15
**Response to all reviewers: General Appreciation and Comments**

We sincerely thank all reviewers for their thoughtful feedback and constructive comments. Your insightful suggestions have provided valuable perspectives that will significantly enhance the quality and clarity of our work.

Overall, we acknowledge that certain aspects of our writing may have led to potential misinterpretations. We will improve the writing and presentation to enhance the clarity.

Our work builds on the findings of Raya and Ambrogioni (2023), who demonstrated that phase transitions in diffusion models stem from spontaneous symmetry breaking at the hypersphere's origin. Specifically, they explored how delaying the initialization time of the reverse diffusion process, referred to as late initialization, affects generative performance. As the initialization time $T$ increases, they observed a sharp rise in the Wasserstein distance between the generated samples and the training data distributions beyond a critical point. The phenomenon is referred to as phase transition.
While Raya and Ambrogioni (2023) provide valuable insights into highly symmetric scenarios, such as those involving hyperspheres, their study is naturally focused on idealized situations. In more general asymmetric settings, multiple phase transitions may occur (c.f. "Generative diffusion in very large dimensions" Biroli and Mézard (2023), and "Manifolds, Random Matrices and Spectral Gaps: The geometric phases of generative diffusion",Ventura et al, (2024)), making the understanding of phase transition phenomena more complex. To address this challenge, we focused on mathematical quantities derived from the geometric structure of data distributions: Tubular neighborhoods. In particular, we demonstrated that the injectivity radius---the supremum distance within which the nearest point on the data distribution is uniquely determined---plays a crucial role in the generation process.
In response to the reviewers’ feedback, we plan to improve the clarity of the intuitive explanation of our theorem, as well as the description of the numerical experiments and results. Additionally, we will include further explanations and conduct additional experiments to strengthen our findings.

Apologies and correction:
Section 4 primarily focuses on the backward process (i.e., the generative process), and $s$ is a typo that should be replaced with $t$. We want to inform you of this in this overall comment because this typo could make the section even harder to understand. In the meantime, we will correct these typos in the manuscript. We apologise for the errors and any lack of readability.

---

> ### Author Response · Authors · 2024-11-24
> **Response to all reviewers: Changes Made in Response to Review Comments**
>
> We have carefully revised the manuscript based on the feedback provided. Below, we outline the specific changes made to the relevant sections. These revisions primarily address clarity and presentation, while additional experiments are currently underway, as detailed below.
> ### Section 4
> - **Notation Adjustment:**
>   All instances of `s` have been changed to `t` for consistency throughout the section.
> - **Remark 4.3:**
>   Revisions were made to clarify and improve the accuracy of the remark.
> - **Proposition 4.2:**
>   The notation `f(t,x)` was updated to `f_t(x)` for consistency with Eq. (3). A similar adjustment was made for `g`.
> ### Section 5
> - **Figure Captions:**
>   The captions for Figures 4 and 5 have been revised to include detailed explanations of the blue and red lines, as well as descriptions of the axes.
> - **Dataset Removal:**
>   In Table 2, the datasets *Ellipse (R=2, r=1)* and *Torus (R=2, r=1)* have been removed due to space constraints. These datasets and their corresponding results have been moved to **Appendix J.4** for reference.
> - **Notation Change in Tables:**
>   In Tables 2, 3, and 4, the notation for `ρ_proportion` has been updated from percentage format to ratio format for clarity.
> ### Section 6
> - **Paragraph Adjustment:**
>   The final sentence of the first paragraph was commented out due to space limitations.
> ### Appendix F
> - **Algorithm 1 Explanation:**
>   Additional explanations have been added to Algorithm 1 to enhance clarity and improve its understanding. We will continue to refine this explanation to ensure better readability and comprehension.
> ### Ongoing Work
> The changes made in this revision primarily consist of straightforward corrections and improvements to the manuscript. Regarding the additional experiments suggested during the review process, we are actively working on extending the **mixed case analysis in Section 5.3**. We aim to report the results of these extended experiments in a subsequent update.

---

> ### Author Response · Authors · 2024-11-27
> **Response to All Reviewers: Changes Made in Response to Review Comments**
>
> In this update, we have incorporated additional experiments and added a new citation to extend the analysis in response to the reviewers' suggestions:
>
> 1. Following the suggestion from Reviewer 55op, we have added the citation: "Reconstruction and interpolation of manifolds I: The geometric Whitney problem," Fefferman et al., 2021, after Theorem 3.6 in Section 3.
>
> 2. As an extension of the Disjoint Arcs cases in Section 5.3, we conducted experiments with the following parameter settings:
>    - (R, r) = (3, 1), where the injectivity radius is 1.
>    - (R, r) = (3/2, 1/2), where the injectivity radius is 1/2.
>
>   The results of these experiments have been included in Appendix J.5.
>
> 3. For the elliptical case and Disjoint Arcs Case, we visualized the score vector fields at different time steps to provide further insights into their dynamics. These results have been included in Appendix K.
>
> We hope these experiments will provide deeper insights into our study on the injectivity radius and its implications as discussed in the paper.

---

### Meta-Review · Area_Chair_okaf · 2024-12-12

**Metareview:**

The paper hypothesizes the phase transition in diffusion model is associated with a concept coined `` ``injectivity radius'', and when the diffusion path enters the tubular neighborhood of the manifold. While the idea might be promising, the writing of the paper limits its clarity and makes it hard to follow. In addition, the paper in its current form doesn't make a fully convincing case why the tubular neighborhood and injectivity radius is the right explanation for the phase transition phenomenon. Therefore, in its current form it is not ready for publication at ICLR.

**Additional Comments On Reviewer Discussion:**

The paper some significant typos in the initial submission that hinders its readability, which the authors corrected to some extent during the rebuttal. Additional experiments are also added to alleviate some reviewers' concerns, but not to the extent that flip the decision.

---

### Decision · Program_Chairs · 2025-01-22

Reject